# QUANTUM MECHANICAL FRAMEWORK FOR QUANTIZATION-BASED OPTIMIZATION: FROM GRADIENT FLOW TO SCHRÖDINGER EQUATION

## ABSTRACT

This work presents a quantum mechanical framework for analyzing quantization-based optimization algorithms. The sampling process of the quantization-based search is modeled as a gradient-flow dissipative system, leading to a Hamilton–Jacobi–Bellman (HJB) representation. Through a suitable transformation of the objective function, this formulation yields the Schrödinger equation, which reveals that quantum tunneling enables escape from local minima and guarantees access to the global optimum. By establishing the connection to the Fokker–Planck equation, the framework provides a thermodynamic interpretation of global convergence. Such an analysis between the thermodynamic and the quantum dynamic methodology unifies combinatorial and continuous optimization, and extends naturally to machine learning tasks such as image classification. Numerical experiments demonstrate that quantization-based optimization consistently outperforms conventional algorithms across both combinatorial problems and nonconvex continuous functions.

## 1 INTRODUCTION

We consider the optimization problem defined by the objective function $f : \mathbb{R}^d \to \mathbb{R}^+$:

$$\min_{\boldsymbol{x} \in \mathcal{X}} f(\boldsymbol{x}), \tag{1}$$

where $\mathcal{X} \subseteq \mathbb{R}^d$ denotes the domain of the parameter $\boldsymbol{x}$. The objective function defined in equation 1 is often nonconvex and possibly nonsmooth, particularly in combinatorial optimization problems. We further introduce a stochastic function $F(\boldsymbol{x}; \xi)$ with a random variable $\xi \in \mathbb{R}^d$, where the objective function satisfies:

$$f(\boldsymbol{x}) = \mathbb{E}_\xi \left[ F(\boldsymbol{x}; \xi) \right], \tag{2}$$

and the function $F : \mathbb{R}^d \times \mathbb{R}^d \to \mathbb{R}^+$ is nonconvex and nonsmooth. This formulation constitutes a stochastic optimization problem. Problems of the form equation 2 commonly arise in a wide range of applications, including machine learning, control theory, and finance. Consequently, an analysis framework to design algorithms capable of handling both combinatorial and stochastic optimization problems has been a central research topic in this field.

From the perspective of combinatorial optimization, heuristic methods such as thermodynamical approaches (e.g., simulated annealing) (Kirkpatrick et al., 1983; Geman & Hwang, 1986; Zhou & Chen, 2013) and biologically inspired algorithms have long served as representative solvers (Goldberg, 1989; Jiang et al., 2007). With the rise of quantum computing, adiabatic quantum algorithms based on spin-glass models have also emerged (Kadowaki & Nishimori, 1998; Leng & Shi, 2025).

Despite their success, these approaches remain specialized for NP-hard combinatorial problems such as the Travelling Salesman Problem (TSP), and are not readily adaptable to gradient-based learning dynamics in machine learning, except in limited cases such as reinforcement learning. Motivated by these limitations, we propose a quantum mechanical analysis framework for quantization-based optimization from the analysis of the gradient-based dissipative system. Such a system is governed by the dynamics $d\boldsymbol{x}_t = -\nabla f(\boldsymbol{x}_t) \, dt$, where energy is gradually dissipated and the trajectory converges toward a stable equilibrium (e.g., a minimum-energy state), naturally analogous to learning dynamics in artificial intelligence. Even sampling-based integer programming solvers can be regarded as

gradient-flow systems when the sampling process aligns the search directions with the gradient (Geman & Geman, 1984; Rere et al., 2015). Within this perspective, we construct a Lagrangian incorporating algorithmic constraints and derive the Hamilton–Jacobi–Bellman (HJB) equation. Building on the HJB formulation, we derive a partial differential equation for the transition probability density via a suitable transformation of the objective function. This formulation leads to a Schrödinger equation for quantization-based optimization via the Witten–Laplacian. The resulting dynamics correspond to the adiabatic evolution of the eigenvalues of the quantum Hamiltonian, demonstrating that quantum tunneling—induced by quantization of the objective function—serves as the essential mechanism for escaping local minima. We further formulate a thermodynamic equation associated with the Schrödinger representation and derive a discrete state updating rule that serves as a learning equation in machine learning. This analysis also reveals that the quantization step size coincides with the temperature in thermodynamical formulations and corresponds to the spectral gap in quantum adiabatic evolution. Consequently, these results establish the global convergence property of quantization-based optimization.

In summary, this work develops a quantum dynamical analysis of quantization-based optimization, bridging concepts from quantum mechanics, thermodynamics, and machine learning. Specifically, our contributions are as follows:

- Applicability as a general optimizer for nonconvex and nonsmooth objective functions through numerical quantization.
- An enhanced quantum tunneling mechanism that enables escape from local minima.
- Demonstrated robustness against stochastic procedures in optimization, such as sampling and random initialization.
- A unified theoretical connection between quantum mechanics and thermodynamics within a gradient-based iterative learning framework.

Together, these contributions highlight the potential of numerical quantization as a quantum-inspired paradigm for optimization in modern machine learning.

## 1.1 RELATED WORKS

**Non-convex optimization based on quantum mechanics** The similarities between the stochastic properties of quantum mechanics and the statistical principles of thermodynamics motivated early efforts in quantum-inspired computing (QIC), such as the quantum random walk (QRW), (Aharonov et al., 1993; Farhi & Gutmann, 1998). A more extensively studied line of work is quantum-inspired annealing (QIA), (Kadowaki & Nishimori, 1998; Santoro & Tosatti, 2006; Hadfield et al., 2019), which formulates the Hamiltonian of the Schrödinger equation as a quadratic unconstrained binary optimization (QUBO) problem and incorporates a quadratic penalty term (Kadowaki & Nishimori, 1998). The dynamics of QIA for escaping local minima have been shown to be analogous to quantum tunneling (Hamacher, 2006; Muthukrishnan et al., 2016). Building on these ideas, the quantum approximate optimization algorithm (QAOA) was introduced (Hormozi et al., 2017; Zhou et al., 2020; Yao et al., 2022), followed by further advancements leading to the variational quantum eigensolver (VQE), (Peruzzo et al., 2014; Uvarov et al., 2020; Su & Liu, 2024). This family of methods not only addresses QUBO formulations but also extends to quantum-computing-based AI learning algorithms, such as Boltzmann machine variants (Khoshaman et al., 2019; Wang et al., 2025). In parallel, QAOA-inspired approaches have evolved through connections to quantum variational Monte Carlo (VMC), (Carleo & Troyer, 2019; Wang et al., 2023) and quantum diffusion Monte Carlo (DMC) (Sánchez-Baena et al., 2018; Zhang & Chen, 2024).

**Non-convex optimization based on thermodynamics** Simulated Annealing (SA), (Khachaturyan et al., 1979; Kirkpatrick et al., 1983), introduced in the early 1980s, was the first thermodynamically inspired Markov Chain Monte Carlo (MCMC) method for global combinatorial optimization. Its dynamics were later analyzed through statistical thermodynamics (Geman & Hwang, 1986; Locatelli, 1996). This line of work further led to stochastic search algorithms based on weak convergence principles, such as Langevin dynamics, applied to stochastic optimization and integer programming (Xu et al., 2018; Li et al., 2022). More recently, thermodynamics-inspired optimization has motivated diffusion models, which underpin modern generative AI (Song & Ermon, 2019; Ho et al., 2020; Miller et al., 2024; Deng et al., 2024).

## 2 PRELIMINARIES

This section presents the paper's definitions, assumptions, and fundamental formulas. We also briefly introduce the notation used throughout the paper. A complete list of all notations can be found in the appendix of the supplementary material.

### 2.1 DEFINITION AND ASSUMPTION

In signal processing literature, researchers conventionally define the quantization of $f \in \mathbb{R}$ as $f^Q \triangleq \lfloor \frac{f}{\Delta} + \frac{1}{2} \rfloor \Delta$, where $\Delta \in \mathbb{R}^+$ denotes a quantization step size (Gray & Neuhoff, 2006; Jiménez et al., 2007). While the conventional quantization definition focuses solely on scalar values, we generalize this framework to examine how the quantization step size influences objective functions through a stochastic formulation, as described below:

**Definition 1.** For $f \in \mathbb{R}$, we define the quantization of $f$ as follows:

$$f^Q \triangleq \frac{1}{Q_p} \left\lfloor Q_p \cdot \left(f + \frac{1}{2Q_p}\right) \right\rfloor = \frac{1}{Q_p} \left(Q_p \cdot f + \varepsilon^q\right) = f + \varepsilon^q Q_p^{-1}, \quad f^Q \in \mathbb{Q}, \tag{3}$$

where $\lfloor f \rfloor \in \mathbf{Z}$ denotes the floor function, defined as the greatest integer less than or equal to for all $f \in \mathbb{R}$, $Q_p \in \mathbb{Q}^+$ is the quantization parameter, which means resolution of quantization, and $\varepsilon^q$ represents the fraction for quantization such that $\varepsilon^q : \Omega \mapsto \mathbb{R}[-\frac{1}{2}, \frac{1}{2})$.

We redefine the quantization step size $\Delta$ as the reciprocal of the quantization parameter $Q_p$, such that $Q_p^{-1} \triangleq \Delta$. Henceforth, $\Delta$ will no longer represent the quantization step size and denote the Laplacian instead. Furthermore, we treat the quantization parameter $Q_p$ as a parametric function such that $Q_p : \mathbb{R}^+ \to \mathbb{R}^+$, generalizing its application within the optimization algorithm. Specifically, we define the quantization step size $Q_p^{-1}$ as a function of the iteration index $t$ in the algorithm, as follows:

**Definition 2.** The quantization parameter $Q_p$ is a monotone-increasing function of $t \in \mathbb{R}^+$ such that $Q_p(t) = \gamma \cdot b^{\bar{h}(t)}$, where $\gamma \in \mathbb{Q}^{++}$ denotes the fixed constant parameter, $b \in \mathbb{Z}[2, \infty)$ represents the base (typically 2), and $\bar{h} : \mathbb{R}^{++} \mapsto \mathbb{Z}^+$ denotes the power function satisfying $\bar{h}(t) \uparrow \infty$ as $t \to \infty$.

**Definition 3.** For the objective function given by equation 1, we define the level set of $f$ such that

$$S(k) \triangleq \{\boldsymbol{x}_t \in \mathcal{X} : f(\boldsymbol{x}_t) = k\}, \quad k \in \mathbb{R}^+, \tag{4}$$

where $\boldsymbol{x}_t : \mathbb{R}^+ \to \mathbb{R}^d$ denotes the state vector at $t \in \mathbb{R}^+$ associated with the objective function. We also define the sublevel set as $\check{S}(k) \triangleq \{\boldsymbol{x}_t \in \mathcal{X} : f(\boldsymbol{x}_t) \le k\} = \bigcup_{k' \in [\min f, k]} S(k')$, where the union spans all $k \in \mathbb{R}[\min f, f^Q(\boldsymbol{x}_0)]$.

To analyze the proposed algorithm through the lens of thermodynamics and quantum mechanics, we introduce the following operations

**Definition 4.** We define the differential $\sharp$ operator $d_{f,h}^{(0)}$ and its adjoint, the differential $\flat$ operator, $d_{f,h}^{(0)*}$ (Le Peutrec & Nectoux, 2021; Lelièvre & Parpas, 2024), as follows:

$$d_{f,h}^{(0)} = e^{-f/h} \left(h\nabla_{\boldsymbol{x}}\right) e^{f/h}, \quad d_{f,h}^{(0)*} = -e^{f/h} \left(h\,\nabla_{\boldsymbol{x}}\cdot\right) e^{-f/h}, \tag{5}$$

where $h \in \mathbb{R}^+$ denotes a proportionality constant, $\nabla_{\boldsymbol{x}}$ and $\nabla_{\boldsymbol{x}}\cdot$ denote the gradient and divergence operators, respectively. The Witten-Laplacian is then defined as $\Delta_{f,h}^{(0)} = d_{f,h}^{(0)*} d_{f,h}^{(0)}$.

Furthermore, we present the following assumptions for numerical analysis.

**Assumption 1.** We assume the objective function, $f$, defined in equation 1, is Lipschitz continuous with a positive constant $L > 0$; that is,

$$|f(\boldsymbol{y}) - f(\boldsymbol{x})| \le L\|\boldsymbol{y} - \boldsymbol{x}\|, \quad \boldsymbol{y}, \boldsymbol{x} \in \mathcal{X}. \tag{6}$$

**Assumption 2.** The quantization error $Q_p^{-1}\varepsilon^q$ defined in equation 3 is an independent and uniformly distributed random variable satisfying $Q_p^{-1}\varepsilon^q \sim \mathcal{U}(0, \frac{1}{12Q_p^2})$ and $\mathbb{E}_{\varepsilon^q}\varepsilon_i^q\varepsilon_j^q = 0$ for all $i, j \in \mathbb{Z}^+$ and $i \ne j$, where $\mathcal{U}(a, b)$ denotes the uniform distribution with the expectation $a$ and the variance $b$.

**Assumption 3.** For a given search algorithm targeting the minimizer of $f$, we assume the evolution of the state vector follows the differential equation $d\boldsymbol{x}_t = -\nabla_{\boldsymbol{x}}f(\boldsymbol{x}_t)dt$.

---

**Algorithm 1:** Blind Random Search (BRS) with the quantization-based optimization

---

**Input:** Objective function $f(x) \in \mathbb{R}^+$
**Output:** $x_{opt}, \ f(x_{opt})$
**Data:** $x \in \mathbb{R}^n$
**Initialization**
$\tau \leftarrow 0$ and $\bar{h}(0) \leftarrow 0$
Set initial candidate $x_0$ and $x_{opt} \leftarrow x_0$
Compute the initial objective function $f(x_0)$
Set $b = 2$ and $\gamma = b^{-\lfloor \log_b(f(x_0)+1) \rfloor}, \ Q_p \leftarrow \gamma$
$f_{opt}^Q \leftarrow \frac{1}{Q_p} \left\lfloor Q_p \cdot (f + \frac{1}{2Q_p}) \right\rfloor$

**while** *Stopping condition is not satisfied* **do**
   Set $\tau \leftarrow \tau + 1$
   Select $x_\tau$ randomly and compute $f = f(x_\tau)$
   Calculate $f^Q \leftarrow \frac{1}{Q_p} \left\lfloor Q_p \cdot (f + \frac{1}{2Q_p}) \right\rfloor$
   **if** $f^Q \leq f_{opt}^Q$ **then**
      $x_{opt} \leftarrow x_\tau, f_{opt}^Q \leftarrow f^Q$
      $\bar{h}(\tau) \leftarrow \bar{h}(\tau) + 1, Q_p \leftarrow \gamma \cdot b^{\bar{h}(\tau)}$
   **end**
**end**

---

## 2.2 FUNDAMENTAL PROCESS OF THE QUANTIZATION-BASED SEARCH FROM THE PERSPECTIVE OF LEVEL SETS

For the sake of clarity, we define the iteration index as the time step at which Algorithm 1 updates the solution vector $\boldsymbol{x}_{opt}$, rather than the nominal iteration index $\tau \in \mathbb{Z}^+$ used in Algorithm 1. This definition allows us to denote the current sub-optimal state $\boldsymbol{x}_{opt}$ as $\boldsymbol{x}_t$, indexed by the time step $t$. To quantify the size of a level set, we introduce a measure $m \colon \mathcal{T} \to \mathbb{R}^+$ on the topological space $\mathcal{T}$, such that for all measurable subsets $A, B \subseteq \mathcal{T}, A \subseteq B \implies m(A) \leq m(B)$. Additionally, we define $f_t^Q \triangleq f^Q(\boldsymbol{x}_t, t)$ for convenience. In Algorithm 1, we distinguish between two cases: the first case is $f_{t+1}^Q < f_t^Q$ and the second case is $f_{t+1}^Q = f_t^Q$. For the case of $f_{t+1}^Q < f_t^Q$, we observe that $\check{S}(f_{t+1}^Q) \subset \check{S}(f_t^Q)$. To refine the

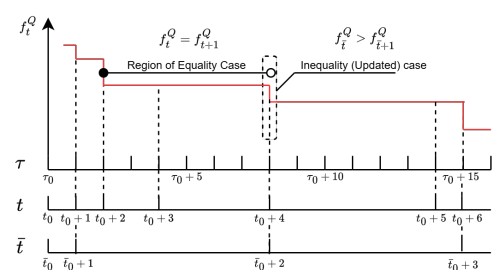

Figure 1: Time indices $\tau$, $t$, and $\bar{t}$. $\tau$ is the basic index, as defined in Algorithm 1. $t$ denotes the time index for $f_{\text{opt}}^Q$, which is updated when $f^Q \leq f_{\text{opt}}^Q$. $\bar{t}$ is updated whenever $f^Q < f_{\text{opt}}^Q$. The red line indicates the trend of $f_\tau^Q$.

analysis, we introduce a secondary time index $\bar{t} \in [t+a, t+b]$ with $a < b$, where $f_{t+a}^Q = f_{t+b}^Q$ implies $f_{\bar{t}+1}^Q < f_{\bar{t}}^Q$ for all $\bar{t}$. The secondary time index excludes intervals where $f_{t+k}^Q = f_t^Q$ for $k \in [a, b)$.

Under these definitions, we can construct the monotone decreasing sequence for $\bar{t}$ such that $\{m(\check{S}(f_{\bar{t}}^Q))\}_{\bar{t}=t_0}$. If we can always find the state $\boldsymbol{x}_{t+1}$ satisfying $f_{t+1}^Q < f_t^Q$, Algorithm 1 converges globally and deterministically, without any assumption of convexity and continuity. However, the inequality search process in Algorithm 1 exhibits significant flaws at any stage. For instance, when $Q_p^{-1}(\bar{t}_s)$ is relatively large, suppose that a feasible candidate $\boldsymbol{x}_{\bar{t}_s+1}$ of $f$ satisfying $f(\boldsymbol{x}_{\bar{t}_s+1}) < f(\boldsymbol{x}_{\bar{t}_s})$ lies within the level set $S(f_{\bar{t}_s}^Q)$. In this case, the algorithm fails to find $\boldsymbol{x}_{\bar{t}_s+1}$ such that $f_{\bar{t}_s+1}^Q < f_{\bar{t}_s}^Q$, since $\hat{f}^Q = f_{\bar{t}_s}^Q$ and it leads to $m\left(\check{S}(f_{\bar{t}_s+1}^Q)\right) = 0$.

To address this deficiency, we analyze the second case, $f_{t+1}^Q = f_t^Q$. In this scenario, since $\boldsymbol{x}_{t+1} \in S(f_t^Q)$, we have $\check{S}(f_{t+1}^Q)|_{Q_p^{-1}(t)} = \check{S}(f_t^Q)$. Consequently, the algorithm can identify a feasible candidate $\boldsymbol{x}_{t+1}$ within $S(f_t^Q)$ provided that the set has a non-zero measure such that $m(\check{S}(f_t^Q)) = m(\check{S}(f_{t+1}^Q)) > 0$ for all $t > 0$. This process implies that the difference of the objective functions between the suboptimal $\boldsymbol{x}_t$ and the updated suboptimal $\boldsymbol{x}_{t+1}$ represents the constraint such that $|f(\boldsymbol{x}_{t+1}) - f(\boldsymbol{x}_t)| < \frac{1}{2}Q_p(t)^{-1}$. Thus, we have the supremum of $f(\boldsymbol{x}_{t+1})$ as $\sup_{\boldsymbol{x}_{t+1} \in \check{S}(f_t^Q)} f(\boldsymbol{x}_{t+1}) = f(\boldsymbol{x}_t) + \frac{1}{2}Q_p^{-1}(t)$, which is proportion to the eigenvalue of the 2-level Hamiltonian for the tunneling effect in the adiabatic evolution.

Existence of the supremum shows that the sequence $\{f_s^Q\}_{s>t}$ generated by Algorithm 1 is not monotonically decreasing, and a conventional analysis is not appropriate to the proof of the convergence. Meanwhile, since the sequence $\{m(\check{S}(f_s^Q))\}_{s=t}^{t_e}$ is monotonically decreasing (possibly non-strictly), we can prove the convergence of the sequence under the perspective of the weak-convergence or the convergence in distribution for large $t > 0$. Statistical evaluation derived from quantum mechanical analysis provides a fundamental equation establishing convergence.

## 3 DYNAMIC ANALYSIS OF THE SEARCH PROCESS

The level set analysis in the previous chapter establishes a foundational framework for understanding the quantization-based optimization governed by Algorithm 1. However, this analysis alone does not fully capture the quantization dynamics underlying the search process and fails to generalize to continuous-domain optimization problems. To analyze the level set dynamics from the perspective of statistics, we introduce an exponential kernel $\Phi$ for the probability density of the objective function, as follows:

$$\Phi : \mathbb{R}^d \times \mathbb{R}^+ \to \mathbb{R}[0, 1], \quad \Phi(\boldsymbol{x}_t, t) = \exp(-Q_p(t) f(\boldsymbol{x}_t)). \tag{7}$$

If we define a normalized variable $Z \triangleq \int_{\mathbb{R}^d} d\boldsymbol{x} \exp(-Q_p f(\boldsymbol{x}))$, we obtain the Gibbs distribution $g(\boldsymbol{x}, t) = Z^{-1}\Phi(\boldsymbol{x}, t)$, $g : \mathbb{R}^d \times \mathbb{R}^+ \to \mathbb{R}[0, 1]$. In equation 7, we define a score function $V(\boldsymbol{x})$ by applying a logarithmic transformation to $g$ for the Hopf-Cole transformation (Léger, 2019):

$$V : \mathbb{R}^d \times \mathbb{R}^+ \to \mathbb{R}^+, \quad V(\boldsymbol{x}_t, t) = -Q_p^{-1}(t) \log g(\boldsymbol{x}_t, t). \tag{8}$$

From equation 7 and equation 8, the function $V$ in equation 8 differs from the original objective function $f$ only by the term $\log Z$. Despite this relationship, we employ $V$ to reformulate the algorithm's dynamics through the Burgers equation framework, a canonical second-order partial differential equation (PDE). This approach is motivated by the structural connection between the Burgers equation, the Fokker-Planck equation (FPE), and the Schrödinger equation, where the transformation in equation 8 serves as a critical tool for analyzing the optimization process.

### 3.1 HAMILTONIAN BASED ANALYSIS

To analyze the equality case, we assume the quantized objective function attains a suboptimal value $f_{t_0}^Q$ at time $t_0$, and the equality case implies $|f_t^Q - f_{t_0}^Q| \le \frac{1}{2}Q_p^{-1}(t_0)$ persists for all $t > t_0$, while $f_{t_e}^Q < f_{t_0}^Q$ holds at an escape time $t_e > t$; thus, we set the time index for equality case as $t \in \mathbb{R}[t_0, t_e]$. We now introduce a key assumption: For $t \in \mathbb{R}[t_0, t_e]$, the equality case induced by quantization allows us to **disregard the precise form of the objective function**. This simplification and the state evolution by Assumption 3 lead to the following cost function according to the property of the gradient-flow dissipative system:

$$\min_{\boldsymbol{x}_t, t \in [t_0, t_e)} |f(\boldsymbol{x}_{t_0}) - f(\boldsymbol{x}_t)| = \int_{t_0}^t \|\nabla_{\boldsymbol{x}} f(\boldsymbol{x}_\tau)\|^2 d\tau, \quad \text{Subject to } |f_{t_0}^Q - f(\boldsymbol{x}_t)| \le \frac{1}{2}Q_p^{-1}(t_0). \tag{9}$$

To minimize the quantized cost function, we establish the Lagrangian for equation 9 for $t$, as follows:

$$L(\boldsymbol{x}_t, \lambda) = \|\nabla_{\boldsymbol{x}} f_t^Q\|^2 + \lambda \left( \frac{1}{4}Q_p^{-2}(t) - (f_t^Q - f(\boldsymbol{x}_t))^2 \right), \quad \forall \boldsymbol{x}_t \in S(f_t^Q) \tag{10}$$

where $\lambda$ denotes the Lagrange multiplier. For $\boldsymbol{x}_t \in S(f_t^Q)$, the score function $V(\boldsymbol{x}_t, t)$ of $f$, as defined in equation 8 under the equality assumption, is given by $V^Q(\boldsymbol{x}_t, t) = k$, where $k$ is a positive constant determined by $f_t^Q$ for $t \in \mathbb{R}[t_0, t_e)$. Thus, the total derivative of the score function is zero, i.e., $dV^Q = 0$, and it implies the following Hamilton–Jacobi–Bellman (HJB) equation (Chen et al., 1995; Wang et al., 2003; Xing & Wang, 2009):

$$\partial_t V^Q(\boldsymbol{x}_t, t) + \nabla_{\boldsymbol{x}} V^Q(\boldsymbol{x}_t, t) \frac{d\boldsymbol{x}_t}{dt} + \min_{\boldsymbol{x}_t \in S(f_t^Q)} L(\boldsymbol{x}_t, \lambda) = 0 \tag{11}$$

To analyze the variation of the sublevel set induced by the quantization-based search algorithm, we construct the Hamiltonian $H(\boldsymbol{x}, t)$, which incorporates the total derivative of the score function (given by equation 7), the Lagrangian (as shown in equation 10), and the state vector evolution $d\boldsymbol{x}_t = -\nabla_{\boldsymbol{x}} f(\boldsymbol{x}_t)dt$. This establishment provides the following HJB equation:

$$\partial_t V^Q(\boldsymbol{x}_t, t) + H(\boldsymbol{x}_t, t) = \partial_t V^Q(\boldsymbol{x}_t, t) - \nabla_{\boldsymbol{x}} V^Q(\boldsymbol{x}_t, t) \cdot \nabla_{\boldsymbol{x}} f(\boldsymbol{x}_t) + \min_{\boldsymbol{x}_t \in S(f_t^Q)} L(\boldsymbol{x}_t, \lambda) = 0. \tag{12}$$

Since the quantized objective functions $f_{t_0}^Q$ and $f_t^Q$ are equivalent at this stage, it is valid to transform the gradient flow dissipative system into a conservative Hamiltonian system.

However, quantizing the objective function yields $\nabla_{\boldsymbol{x}} f_t^Q = 0$, which in turn implies $\nabla_{\boldsymbol{x}} V^Q(\boldsymbol{x}_t, t) = 0$. Thus, equation 12 shows that the minimum of the Lagrangian determines the variation of $V^Q$ with respect to $t$.

**Theorem 3.1.** The derivative of $V^Q$ with respect to $t$ tends to zero, i.e., $\partial_t V^Q(\boldsymbol{x}_t, t) \to 0$, as the quantization step size decreases to zero with increasing $t$, that is, as $Q_p^{-1}(t) \to 0$.

## 3.2 Quantum Mechanical and Thermodynamical Analysis

Theorem 3.1 states that the local minimum condition, $\nabla_{\boldsymbol{x}} f(\boldsymbol{x}_t) = 0$, does not affect the convergence condition $\partial_t V^Q(\boldsymbol{x}_t, t) \to 0$. This result demonstrates that quantization-based optimization is highly robust to local minima. Even if Theorem 3.1 is valid in the equality case, it shows that the algorithm's global convergence depends solely on $Q_p^{-1}(t)$, provided the non-strictly monotonically decreasing property holds. However, Theorem 3.1 does not specify the dynamics governing the search process. To address this, we propose a virtual function that combines with the objective function to satisfy the quantization constraints. This approach builds on the key idea from the previous section: disregarding specific objective functions through a tailored methodology.

**Assumption 4.** Suppose that there exists a virtual objective function $\bar{\psi} : \mathbb{R}^d \times \mathbb{R} \to \mathbb{C}$ induced by quantization, whose amplitude satisfies the constraint in equation 9. We define the transition probability density $\rho : \mathbb{R}^d \times \mathbb{R} \to [0, 1]$:

$$\rho(\boldsymbol{x}_t, t) \triangleq \bar{\psi}(\boldsymbol{x}_t, t) \tfrac{1}{Z} \exp(-f(\boldsymbol{x}_t)/h) \bar{\psi}^*(\boldsymbol{x}_t, t) = \psi(\boldsymbol{x}_t, t) g(\boldsymbol{x}_t) \tag{13}$$

where $\psi$ denotes the probability density function of $\bar{\psi}$ defined as $|\bar{\psi}|^2$, and $h \in \mathbb{R}$ is a scale parameter defined as $h \triangleq Q_p^{-1}(t)$, for $t \in \mathbb{R}[t_0, t_e)$.

In Assumption 4, since the value of the quantized objective function remains constant, the distribution $g : \mathbb{R}^d \times \mathbb{R} \to \mathbb{R}$ depends only on $\boldsymbol{x} \in \mathbb{R}^d$. As a result, the quantization step size can be regarded as a constant within the interval corresponding to the equality case, and we can define a constant scale parameter for $g$ as $h \triangleq Q_p^{-1}(t)$.

As motivated by Assumption 4, we define the following transformed objective function:

$$\bar{f}(\boldsymbol{x}_t, t) = f_t^Q - \tfrac{1}{2} Q_p^{-1}(t) \phi(\boldsymbol{x}_t), \quad \phi : \mathbb{R}^d \mapsto \mathbb{R}[-1, 1], \tag{14}$$

where $\phi$ is a sinusoidal function. Since $\phi$ satisfies the quantization constraint, the quantization error can be viewed as a sinusoidal wave, as illustrated in Figure 2. Accordingly, we replace $\bar{f}$ with $f$ and substitute the quantized function in $g$ with the virtual function, yielding

$$\exp(-\tfrac{1}{h} f_t^Q) = \exp(-\tfrac{1}{h}(f(\boldsymbol{x}_t) + \tfrac{1}{2} Q_p^{-1}(t)\phi(\boldsymbol{x}_t))) = \exp(-\tfrac{1}{h} f(\boldsymbol{x}_t)) \exp(-\tfrac{1}{2} Q_p^{-1}(t)\phi(\boldsymbol{x}_t)) = g(\boldsymbol{x}_t)\psi(\boldsymbol{x}_t, t).$$

Therefore, this formulation enables a non-zero gradient of the quantized objective function.

From the definition of the virtual function, the wave function satisfies the quantization constraints in equation 9, which implies $\lambda = 0$. Under the framework of equation 8 and equation 13, if the score function is defined as $V^Q(\boldsymbol{x}_t, t) = -h \log \rho(\boldsymbol{x}_t, t)$, then the HJB equation in equation 12 can be expressed as follows:

$$\partial_t V(\boldsymbol{x}_t, t) = \nabla_{\boldsymbol{x}} V(\boldsymbol{x}_t, t) \cdot \nabla_{\boldsymbol{x}} f(\boldsymbol{x}_t) - \|\nabla_{\boldsymbol{x}} f(\boldsymbol{x}_t)\|^2, \tag{15}$$

where Assumption 4 ensures that $V^Q(\boldsymbol{x}_t, t) = V(\boldsymbol{x}_t, t)$.

For notational simplicity in this section, we write $g$ for functions of the state $\boldsymbol{x}_t \in \mathbb{R}^d$ instead of $g(\boldsymbol{x}_t)$, omitting the explicit dependence on $\boldsymbol{x}_t$ when clear from context. Similarly, for functions of both the state and an additional parameter, such as $\psi(\boldsymbol{x}_t, t)$, we denote them as $\psi_t$, where the additional parameter is indicated as a subscript. The partial derivative of $V_t$ with respect to $t$ is $\partial_t V_t = -h\rho_t^{-1}\partial_t\rho_t$ and its gradient with respect to $\boldsymbol{x}$ is $\nabla_{\boldsymbol{x}} V_t = -h\rho_t^{-1}\nabla_{\boldsymbol{x}}\rho_t$. Substituting these derivatives into the HJB equation equation 15, we derive the partial differential equation governing $\rho_t$, which characterizes the thermodynamic behavior of the algorithm.

**Theorem 3.2.** Under the definitions and assumptions for the score function and the virtual function, we derive the following thermodynamic evolution for $\rho_t \triangleq \rho(\boldsymbol{x}_t, t)$:

$$\partial_t \rho_t = \nabla_{\boldsymbol{x}} \cdot (\rho_t \nabla_{\boldsymbol{x}} f) + h^{-1}(\|\nabla_{\boldsymbol{x}} f\|^2 - h\Delta_{\boldsymbol{x}} f)\rho_t \tag{16}$$

Furthermore, by rewriting equation 16 using the Witten-Laplacian (Definition 4), we derive the Schrödinger equation, as formalized in the following theorem.

**Theorem 3.3.** Given the thermodynamic evolution described in equation 16, replacing $h$ with $i\hbar/2m$ yields the following Schrödinger equation for $\psi_t$:

$$i\hbar\partial_t \psi_t = -\tfrac{\hbar^2}{2m}\Delta_{\boldsymbol{x}}\psi_t + \tfrac{m}{2}\left(\|\nabla_{\boldsymbol{x}} f\|^2 - \tfrac{i\hbar}{m}\Delta_{\boldsymbol{x}} f\right)\psi_t, \tag{17}$$

where $\hbar$ denotes the reduced Planck's constant and $m$ represents the mass of a particle in quantum mechanics.

By introducing a potential energy $\overline{V} : \mathbb{R}^d \to \mathbb{R}$ as $\overline{V}(\boldsymbol{x}) \triangleq \frac{m}{2}(\|\nabla_{\boldsymbol{x}} f\|^2 - h\Delta_{\boldsymbol{x}} f)$, equation equation 17 takes the standard form of the Schrödinger equation: $i\hbar\partial_t \psi_t = -\frac{\hbar^2}{2m}\Delta_{\boldsymbol{x}} \psi_t + \overline{V}\psi_t$.

**Tunneling Effect and Adiabatic Evolution** : Adiabatic evolution describes the dynamics of a quantum system in which, if the initial state is the ground state of the Hamiltonian, and the Hamiltonian changes sufficiently slowly, the system remains in the instantaneous ground state throughout its evolution. Given the mixing Hamiltonian $H_B(\boldsymbol{x}_t, t)$ as an initial ground state and the problem Hamiltonian $H_P(\boldsymbol{x}_t)$ as an objective function to optimize, we can formulate the adiabatic evolution as follows:

$$\bar{H}(\boldsymbol{x}, t) = (1 - \beta(t)) H_P(\boldsymbol{x}) + \beta(t)H_B(\boldsymbol{x}, t), \quad \beta(t) \in \mathbb{R}[0, 1], \ \beta(t) \downarrow 0, \text{for } t \in [0, T]. \quad (18)$$

In the adiabatic evolution, when the energy gap is sufficiently small, the quantum tunneling effect enables the system's state to transition to a lower energy, facilitating global optimization.

From a number-theoretic perspective, the objective function can be represented in base-$b$ expansion as $f = f_b + \sum_{k=1}^{\infty} f_k b^{-k}$, where $f_k \in \mathbb{Z}[0, b)$. For a given quantization step size $Q_p^{-1}(t) \triangleq b^{-t}$, this expansion yields the following adiabatic evolution equation for quantization-based optimization:

$$f_t^Q = (1 - b^{-t})f(\boldsymbol{x}_t) + b^{-t}f_b, \quad (19)$$

where $f_b$ denotes the ground state, i.e., the value of the objective function at the lowest quantization resolution as determined by quantization.

Therefore, if we demonstrate the quantum tunneling effect of the state updating process in Algorithm 1 by employing the Schrödinger equation addressed in the previous chapter, we can argue that quantization-based optimization is equivalent to Adiabatic evolution. To this end, we analyze the probability of the state existing through an energy barrier $V_0$ at a fixed $t \in \mathbb{R}[t_0, t_e)$ in the equality case, using the time-independent Schrödinger equation:

$$\Delta_{\boldsymbol{x}} \psi_t = 2m\hbar^{-2}(V_0 - f_t^Q)\psi_t, \quad V_0 = f_t^Q + k \cdot Q_p^{-1}(t), \ k \in \mathbb{Z}^+, \quad (20)$$

where the potential energy $\bar{V}$ is defined as $\bar{V} = V_0 - f_t^Q$, since $f_t^Q$ acts as the ground state. For analytical convenience, if we evaluate $\psi_t$ on a one-dimensional eigenspace of the Hamiltonian, the transmission probability $T$ for the state $\boldsymbol{x}_t$ tunneling through the energy barrier $V_0$ with the width $D > 0$ is

$$T \propto \exp\left(-\frac{2}{\hbar}\sqrt{2m(V_0 - f_t^Q)}\right) \cdot D. \quad (21)$$

Consequently, for finite $k$, we observe that $T$ is strictly positive; thus, quantization-based optimization embeds the quantum tunneling effect. This tunneling effect is a theoretical foundation for the QIA dynamics to select the global optimum (Wenzel & Hamacher, 1999; Hérau et al., 2011; Muthukrishnan et al., 2016). By replacing the eigenvalue of the Hamiltonian in equation 18 or the quantized objective function in equation 19 with the standard Hamiltonian self-adjoint operator $\hat{H}$ operating on $\mathbb{H}$, equation 18 forms the basis of quantum computing-based optimization algorithms such as QAOA and VAE (Zhou et al., 2020; Yao et al., 2022; Su & Liu, 2024).

**Derivation of Gradient-based Search Algorithm**: According to Theorem 3.3, equation 16 can be reformulated as the following standard Fokker–Planck equation:

$$\partial_t \rho_t = \nabla_{\boldsymbol{x}} \cdot (\rho_t \nabla_{\boldsymbol{x}} f) + h\Delta_{\boldsymbol{x}} \rho_t \quad (22)$$

By substituting the definition $h = Q_p^{-1}(t)$ into equation 22, we derive the stochastic differential equation (SDE) that governs the evolution of the state vector $\boldsymbol{x}_t$:

$$d\boldsymbol{x}_t = -\nabla_{\boldsymbol{x}} f(\boldsymbol{x}_t)dt + \sqrt{2Q_p^{-1}(t)}d\boldsymbol{W}_t, \quad \boldsymbol{W}_t \in \mathbb{R}^d, \boldsymbol{W}_t \sim \mathcal{N}(\boldsymbol{0}, \boldsymbol{I}_d). \quad (23)$$

Equation equation 23 validates Assumption 3, enabling us to reinterpret the quantization-based search algorithm as a learning process governed by overdamped Langevin dynamics. Since equation 23 is known to be an $O(\sqrt{\eta})$-accurate approximation of the continuous-time SDE (Shi et al., 2023), we obtain the following discrete-time stochastic update rule suitable for general-purpose machine learning applications:

$$\boldsymbol{X}_{\tau_{t+1}} = \boldsymbol{X}_{\tau_t} - \eta \nabla_{\boldsymbol{x}} f(\boldsymbol{X}_{\tau_t}) + \sqrt{2\eta Q_p^{-1}(t)}\, \boldsymbol{\xi}_{\tau_t}, \quad \tau_t \in \mathbb{R}^+. \quad (24)$$

| | Cost of Solution Path | | | Sample Standard Deviation | | | Improvement Ratio | | | |
|---|---|---|---|---|---|---|---|---|---|---|
| Cities | QTZ* | SA | QIA | QTZ* | SA | QIA | QTZ to SA | QTZ to QIA | QTZ to NN | NN |
| 100 | 1691.76 | 1732.16 | 1721.07 | 24.34 | 38.93 | 42.77 | 2.33 | 3.16 | 21.65 | 2159.27 |
| 125 | 1920.42 | 2013.31 | 2054.59 | 37.71 | 57.60 | 37.96 | 4.61 | 6.53 | 16.43 | 2297.87 |
| 150 | 2013.61 | 2218.52 | 2208.50 | 49.44 | 56.93 | 48.41 | 9.24 | 8.82 | 19.38 | 2497.65 |
| 175 | 2176.76 | 2532.26 | 2617.09 | 30.80 | 58.19 | 61.21 | 9.54 | 16.83 | 8.56 | 2380.53 |
| 200 | 2366.72 | 2924.87 | 2988.78 | 28.24 | 90.36 | 62.62 | 18.72 | 20.45 | 14.16 | 2769.73 |

Table 1: Experimental Results for TSP with More Than 100 Cities;

| | Iterations | | | Improvement Ratio | | Solution vs Exact Minimum Ration | | |
|---|---|---|---|---|---|---|---|---|
| Function | QTZ | SA | QIA | SA/QTZ | QIA/QTZ | QTZ | SA | QIA |
| Xin-She Yang N4 | 3144 | 6420 | 17* | 2.04 | * | 54.57% | 54.57% | 35.22% |
| Salomon | 1727 | 1312 | 7092 | 0.76 | 4.11 | 100.00% | 99.99% | 99.99% |
| Drop-Wave | 254 | 907 | 3311 | 3.57 | 13.04 | 100.00% | 99.99% | 99.99% |
| Shaffer N2 | 2073 | 7609 | 9657 | 3.67 | 4.66 | 100.00% | 99.99% | 99.99% |

Table 2: Experimental results for low-dimensional benchmark functions. Iterations denote the number of iterations each algorithm requires to find the minimum of the benchmark functions. Improvement Ratio represents the ratio of iterations required between algorithms, such as SA/QTZ and QIA/QTZ. For the benchmark function "Xin-She Yang N4", an asterisk (*) indicates that QIA failed to find the minimum or the solution found by QTZ (100% = exact optimum).

Here, $\eta \in (0, 1)$ denotes the learning rate, and $\tau_t$ is a discrete-time index defined by $\tau_t \triangleq t\eta$. The random vector $\boldsymbol{\xi}_{\tau_t}$ represents the increment of the Wiener process $\boldsymbol{W}_t$ in equation 23, and follows the distribution $\boldsymbol{\xi}_{\tau_t} \sim \mathcal{N}(\boldsymbol{0}, \boldsymbol{I}_d)$.

The learning equation equation 24, along with its corresponding stochastic differential equation (SDE) equation 23, guarantees global convergence from a thermodynamic perspective, as derived from the Fokker–Planck equation equation 22. Under the assumption that the objective function $f$ satisfies Lipschitz continuity, we can derive the Radon–Nikodym derivative of the time-dependent transition probability density for both $\boldsymbol{X}_t$ and its discretized counterpart $\boldsymbol{X}_{\tau_t}$, associated with equation 23 and equation 24, respectively, and a standard Wiener process. By applying the Radon–Nikodym derivative and invoking Girsanov's theorem, we establish the weak convergence of the transition probability density. This result implies global convergence in the sense of Laplace's method. Detailed proofs of the global convergence of these learning dynamics have been provided by various researchers over the years (Chiang et al., 1987; Locatelli, 1996; Seok & Cho, 2023). Accordingly, we omit the proof in this manuscript.

In contrast to the random vector $\boldsymbol{\xi}_t$ derived from the Wiener process $\boldsymbol{W}_t$, we can formulate an iterative learning equation for quantization-based optimization using other i.i.d. random vectors. One proposed approach involves formulating a learning equation based on the quantization error $\boldsymbol{\varepsilon}_t^q \in \mathbb{R}^d$. This error term generates an independent increment process $\boldsymbol{\varepsilon}_{\tau_t}^q$ with the property $\mathbb{E}_\varepsilon \boldsymbol{\varepsilon}_s^q (\boldsymbol{\varepsilon}_t^q - \boldsymbol{\varepsilon}_s^q) = 0$ for $t > s$, ensuring temporal independence. For a convex function under quantization constraints (Definition 1), the learning equation quantizing the directional derivative $h : \mathbb{R}^d \times \mathbb{R} \to \mathbb{R}^d$ is defined as:

$$\boldsymbol{X}_{\tau_{t+1}}^q = \boldsymbol{X}_{\tau_t}^q + \overline{Q}_p^{-1}(\tau_t) \left\lfloor \overline{Q}_p(\tau_t) \left( \eta h(\boldsymbol{X}_{\tau_t}^q, \tau_t) + \tfrac{1}{2} \right) \right\rfloor = \boldsymbol{X}_{\tau_t}^q + \eta h(\boldsymbol{X}_{\tau_t}^q, \tau_t) + \varepsilon_{\tau_t}^q \overline{Q}_p^{-1}(\tau_t),$$
(25)

where $\overline{Q}_p^{-1} \in \mathbb{R}^+$ represents another quantization step size, typically setting $\overline{Q}_p^{-1} := Q_p^{-\frac{1}{2}}$.

## 4 EXPERIMENTAL RESULTS

### 4.1 COMBINATORIAL AND NON-CONVEX OPTIMIZATION

We conducted numerical experiments on the Traveling Salesman Problem (TSP) to evaluate the performance of the quantization-based search algorithm (QTZ) for combinatorial problems where gradient information is unavailable. We compared QTZ against simulated annealing (SA), a representative thermodynamics-based method, and quantum-inspired annealing (QIA), which is grounded in quantum mechanics. The results show that QTZ outperforms both SA and QIA on TSP instances with 100 or more cities, as shown in Table 1. Specifically, for the 125- and 150-city problems, the sample standard deviation for each algorithm indicates that QTZ performs similarly to QIA. This result indicates that for relatively intractable problems, the search dynamics of QTZ exhibit

dynamics similar to quantum tunneling as observed in QIA. Finally, although we set the quantization schedule in QTZ, the temperature in SA, and the adiabatic evolution schedule in QIA to have similar formulations, SA and QIA fail to find better solutions than the initial path determined by the nearest neighbor scheme for the 175- and 200-city problems. In contrast, QTZ successfully finds shorter paths and maintains a consistent standard deviation across various city sizes (see Table 1).

Next, we evaluated the algorithm on non-convex problems using 10 representative benchmark tests, including standard datasets from CEC 2017 (single-objective real-parameter optimization (Awad et al., 2017)) and CEC 2022 (dynamic optimization problems (Kumar et al., 2021)). The quantization-based search algorithm consistently outperformed conventional gradient-based methods in identifying global optima. For low-dimensional problems, we applied gradient-free algorithms including SA, QIA, and the QTZ. For high-dimensional problems, we combined QTZ with conventional gradient-based search methods, applying this approach in our machine learning experiments. Detailed experimental results, including those omitted due to page constraints, are provided in the Appendix section "Detailed Experimental Information".

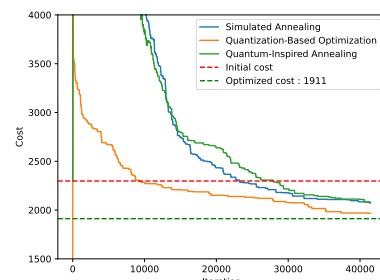

Figure 2: Convergence trends for each algorithm in the 125-city TSP experiment. The quantization-based optimization scheme exhibits a faster convergence property compared to other algorithms.

## 4.2 MACHINE LEARNING

We evaluate the performance of the quantization-based search algorithm on four image datasets (FashionMNIST, CIFAR10, CIFAR100, STL10) (Xiao et al., 2017; Krizhevsky et al., 2009a;b; Coates & Ng, 2011) for machine learning tasks. In Table 3, QSLD refers to applying quantization to the directional derivative in the Adam optimizer, whereas QSLGD denotes quantization applied to a general negative gradient, such as $h = -\nabla_{\boldsymbol{x}} f(\boldsymbol{X}_{\tau_t})$. Experimental results demonstrate that the gradient-based quantization search described in equation 25 achieves 2–3% higher classification accuracy compared to conventional optimizers, including those based on Stochastic Gradient Descent (SGD) and Adaptive Moment Estimation (Adam). As noted previously, detailed experimental results are provided in the Appendix section 'Detailed Information of Experiments'.

## 5 CONCLUSION

In this paper, we propose a numerical quantization-based analysis framework for optimization algorithms, grounded in thermodynamic and quantum mechanical principles. We show that signal quantization applied to the objective function serves as an effective method for escaping local minima and finding global optima, by leveraging quantum tunneling effects. Although quantum tunneling induced by adiabatic evolution is a core mechanism in the quantization-based search algorithm, thermodynamic analysis is still required to rigorously establish global convergence, as the analysis depends on energy level dynamics in real space. Importantly, this study demonstrates only the superposition property in computation induced by signal quantization, without addressing entanglement effects. Further research is needed to extend the framework to quantum computing applications that fully exploit numerical quantization.

| Data Set | FashionMNIST | CIFAR10 | CIFAR100 | STL10 | Data Set | FashionMNIST | CIFAR10 | CIFAR100 | STL10 |
| --- | --- | --- | --- | --- | --- | --- | --- | --- | --- |
| Model | CNN 3 Layers | ResNet-50 (56 Layer Blocks) | | | Model | CNN 3 Layers | ResNet-50 (56 Layer Blocks) | | |
| QSLGD | 89.29 | 73.8 | 37.77 | 50.68 | QSLGD | 0.085426 | 0.009253 | 0.030104 | 0.007205 |
| SGD | 91.47 | 63.31 | 25.90 | 46.92 | SGD | 0.132747 | 0.001042 | 0.005478 | 2.214468 |
| ASGD | 91.42 | 63.46 | 26.43 | 47.90 | ASGD | 0.130992 | 0.001166 | 0.004981 | 2.001648 |
| QSLD | 91.59 | 85.09 | 49.60 | 58.04 | QSLD | 0.059952 | 0.011456 | 0.037855 | 0.005939 |
| ADAM | 87.12 | 82.08 | 46.32 | 57.32 | ADAM | 0.176379 | 0.012421 | 0.038741 | 0.53936 |
| ADAMW | 86.81 | 82.20 | 47.01 | 56.87 | ADAMW | 0.182867 | 0.012551 | 0.038022 | 0.74659 |
| NADAM | 87.55 | 82.46 | 48.56 | 55.93 | NADAM | 0.140066 | 0.014377 | 0.037409 | 1.17814 |
| RADAM | 87.75 | 82.26 | 48.61 | 56.5 | RADAM | 0.146404 | 0.010526 | 0.044913 | 0.763353 |

Table 3: Experimental Results for Image Datasets. Test accuracy (left) and training errors (right).

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

## A    APPENDIX

We provide the notation, proofs of lemmas and theorems, and more detailed information about the experiments in the following sections of the manuscript.

The Python code used for all experiments is available at the following repository: https://github.com/SDE-AI-00/Quantization-based-Optimization

## B    NOTATIONS

- $\mathbb{R}[\alpha, \beta)$   $\{x \in \mathbb{R} | \alpha \leq x < \beta, \ \alpha, \beta \in \mathbb{R}\}$

- $\mathbb{R}^+$   $\{x | x \geq 0, \ x \in \mathbb{R}\}$

- $\mathbb{R}^{++}$   $\{x | x > 0, \ x \in \mathbb{R}\}$

- $\mathbb{Q}^+$   $\{x | x \geq 0, \ x \in \mathbb{Q}\}$

- $\mathbb{Q}^{++}$   $\{x | x > 0, \ x \in \mathbb{Q}\}$

- $\mathbb{Z}^+$   $\{x | x \geq 0, \ x \in \mathbb{Z}\}$

- $\mathbb{Z}^{++}$   $\{x | x > 0, \ x \in \mathbb{Z}\}$, $\mathbb{Z}^{++}$ is equal to $\mathbb{N}$.

- $\lfloor x \rfloor$   $\max\{y \in \mathbb{Z} | y \leq x, \forall x \in \mathbb{R}\}$

- $\lceil x \rceil$   $\min\{y \in \mathbb{Z} | y \geq x, \forall x \in \mathbb{R}\}$

- $\nabla_{\boldsymbol{x}} f(\boldsymbol{x})$   Gradient of the scalar field $f : \mathbb{R}^d \mapsto \mathbb{R}^+$ such that $\nabla_{\boldsymbol{x}} : \mathbb{R} \mapsto \mathbb{R}^d$. For Euclidean space, $\nabla_{\boldsymbol{x}} f(\boldsymbol{x}) = \sum_{i=1}^d \frac{\partial f}{\partial x^i} \boldsymbol{e}_i$, where $\{\boldsymbol{e}_i\}_{i=1}^d = \{\frac{\partial}{\partial x^i}\}_{i=1}^d$ is a local covariant bases

- $\nabla_{\boldsymbol{x}} \cdot \boldsymbol{V}(\boldsymbol{x})$   Divergence of a vector field $\boldsymbol{V} : \mathbb{R}^d \mapsto \mathbb{R}^d$ such that $\nabla_{\boldsymbol{x}} \cdot : \mathbb{R}^d \mapsto \mathbb{R}$. For Euclidean space, $\nabla_{\boldsymbol{x}} \cdot \boldsymbol{V}(\boldsymbol{x}) = \sum_{i=1}^d \frac{\partial f}{\partial x^i}$.

- $\nabla_{\boldsymbol{x}}^2 f(\boldsymbol{x})$   Hessian of the scalar field $f : \mathbb{R}^d \mapsto \mathbb{R}^+$ such that $\nabla_{\boldsymbol{x}}^2 : \mathbb{R} \mapsto \mathbb{R}^{d \times d}$ computed by $\nabla_{\boldsymbol{x}} \nabla_{\boldsymbol{x}} f = \sum_{i,j=1}^d \frac{\partial^2 f}{\partial x^i \partial x^j} \boldsymbol{e}_i \otimes \boldsymbol{e}_j$ for an Euclidean space.

- $\Delta_{\boldsymbol{x}} f(\boldsymbol{x})$   Laplacian of the scalar field $f : \mathbb{R}^d \mapsto \mathbb{R}^+$ such that $\Delta_{\boldsymbol{x}} : \mathbb{R} \mapsto \mathbb{R}$ computed by $\Delta_{\boldsymbol{x}} f(\boldsymbol{x}_t) = \nabla_{\boldsymbol{x}} \cdot \nabla_{\boldsymbol{x}} f = \sum_{i=1}^d \frac{\partial^2 f}{\partial x^{i2}}$ for a Euclidean space.

- $f$ A function that depends on the state vector $\boldsymbol{x}_t$, defined as $f : \mathbb{R}^d \rightarrow \mathbb{R}$. In particular, $f$ denotes a simplified notation introduced after Section 3.2, used in the context of the Fokker–Planck equation or the Schrödinger equation.

- $f_t$ A function that depends on both the state vector and an additional parameter $t$, defined as $f : \mathbb{R}^d \times \mathbb{R} \rightarrow \mathbb{R}$. Specifically, $f_t^Q$ denotes a simplified notation for the quantized objective function introduced after Section 2.2. This notation is defined in the main text.

## C    AUXILIARY DESCRIPTION AND PROOFS OF THEOREMS

Notice that the equation numbers in the statements of the theorems are the same as those in the main manuscript. However, the equation numbers appearing in the proofs of the theorems are independent.

### C.1    AUXILIARY DESCRIPTION FOR CHAPTER 2.2 "FUNDAMENTAL PROCESS OF THE QUANTIZATION-BASED SEARCH FROM THE PERSPECTIVE OF LEVEL SET"

In this section, we more detailed explanations via the following lemmas, elaborating on Section 2.2 of the manuscript.

**Lemma C.1.** For the sequence $\{m(\check{S}(f_{\bar{t}}^Q))\}_{\bar{t}=t_0}^\infty$, where $\bar{t}$ denotes the refined time index, for any $\epsilon > 0$, there exists $\bar{t} > t_0$ such that

$$m(\check{S}(f_{\bar{t}}^Q)) < \epsilon \tag{a1}$$

**Proof of Lemma** The proof follows directly from Algorithm 1 and the definition of the quantized objective function at the refined time index $\bar{t}$, yielding:

$$f_{\bar{t}}^Q - f_{\bar{t}+1}^Q = l \cdot Q_p^{-1}(\bar{t}), \tag{a2}$$

where $l$ is a positive integer, typically $l = 1$. Thus, the sequence $\{m(\check{S}(f_{\bar{t}}^Q))\}_{\bar{t}=t_0}^\infty$ is monotonically decreasing, ensuring convergence to $\hat{m}_f \triangleq \min m(\check{S}(f_{\bar{t}}^Q))$. Consequently, since the measure $m$ decreases proportionally to $Q_p^{-1}(\bar{t})$, the sequence $\{m(\check{S}(f_{\bar{t}}^Q))\}_{\bar{t}=t_0}^\infty$ converges to the optimum. Formally,

$$\forall \epsilon > 0, \ \exists \bar{t} > \bar{t}_0 \ \text{ such that } m(\check{S}(f_{\bar{t}}^Q)) < \epsilon. \tag{a3}$$

$\square$

Meanwhile, the previous lemma describes the global convergence property of the inequality case, and the following lemma describes the boundness property of the equality case, i.e., $f_s^Q = f_t^Q$ for $s \in \mathbb{Z}[t, t_e)$.

**Lemma C.2.** Consider the equality case in which $|f(\boldsymbol{x}_{t+1}) - f(\boldsymbol{x}_t)| < \frac{1}{2}Q_p^{-1}(t)$. Given the quantization parameter $Q_p(t)$ as in Definition 1, whose power index is a linear function of $t$, i.e., $\bar{h}(t) = at+c$, with $a, c \in \mathbb{R}^+$, for $s \in \mathbb{Z}^+[t, t_e)$, the supremum of $f_s^Q$ satisfying $\sup_{s \in \mathbb{Z}^+[t, t_e)} f_s^Q = f_s^Q + \frac{1}{2}Q_p^{-1}(s)$ is bounded as

$$\lim_{s \to \infty} \sup_{\boldsymbol{x}_s \in \check{S}(f_s^Q)} f_s^Q = f_t^Q + \frac{1}{2}Q_p^{-1}(t)\frac{1}{b}\left(\frac{b-2}{b-1}\right), \tag{a4}$$

where $b \in \mathbb{Z}^+$ is the base of $Q_p(t)$ defined in Definition 2, and $t_e$ denotes the escape time such that $f_{t_e}^Q < f_t^Q$. In particular, when $b = 2$, the measure of the set $\{\boldsymbol{x}_s \mid |\sup_{\boldsymbol{x}_s \in \check{S}(f_s^Q)} f_s^Q - f_t^Q| > \epsilon, \ \forall \epsilon > 0\}$ is zero, which means that, as $s$ increases, the state $\boldsymbol{x}_s$ almost surely escapes from the region of the equality case. Therefore, the escape state $\boldsymbol{x}_{t_e}$ can be found in the limit as $s$ increases.

**Proof of Lemma** Let $s = t + k$, where $k \in \mathbb{Z}^+$.

Assuming $\bar{\gamma} \in \mathbb{Q}^+$ and $\bar{b} \in \mathbb{Z}[2, \infty)$, the quantization step size, defined as the reciprocal of the quantization parameter, is given by

$$Q_p^{-1}(t) = \bar{\gamma}^{-1}\bar{b}^{-(a\,t+c)} = \left(\bar{\gamma}\bar{b}^c\right)^{-1}\left(\bar{b}^a\right)^{-t} = \gamma^{-1}b^{-t}, \tag{a5}$$

where $b = \bar{b}^a$ and $\gamma = \bar{\gamma}\bar{b}^c$. It implies that $Q_p^{-1}(t+1) = b^{-1}Q_p^{-1}(t)$.

The quantization-based search algorithm establishes the following relation for the supremum of the quantized objective function for the equality case i.e., $f_s^Q = f_t^Q$ for $s \in \mathbb{Z}[t, t_e)$, as follows:

$$\sup_{\boldsymbol{x}_{t+1} \in \check{S}(f_{t+1}^Q)} f_{t+1}^Q = f_t^Q + \frac{1}{2}Q_p^{-1}(t+1), \ \text{ and } \ \sup_{\boldsymbol{x}_{t+1} \in \check{S}(f_{t+1}^Q)} f_{t+1}^Q = \sup_{\boldsymbol{x}_t \in \check{S}(f_t^Q)} f_t^Q - \frac{1}{2}Q_p^{-1}(t+1). \tag{a6}$$

Therefore, for $k > 0$, we obtain the recursive expressions:

$$\sup_{\boldsymbol{x}_{t+k} \in \check{S}(f_{t+k-1}^Q)} f_{t+k}^Q = \sup_{\boldsymbol{x}_{t+k-1} \in \check{S}(f_{t+k-1}^Q)} f_{t+k-1}^Q - \frac{1}{2}Q_p^{-1}(t+k)$$

$$= \sup_{\boldsymbol{x}_{t+k-2} \in \check{S}(f_{t+k-2}^Q)} f_{t+k-2}^Q - \frac{1}{2}\left(Q_p^{-1}(t+k-1) + Q_p^{-1}(t+k)\right)$$

$$= \sup_{\boldsymbol{x}_{t+k-2} \in \check{S}(f_{t+k-2}^Q)} f_{t+k-2}^Q - \frac{1}{2}Q_p^{-1}(t+k-1)(1+b^{-1}) \tag{a7}$$

$$\cdots$$

$$= \sup_{\boldsymbol{x}_{t+1} \in \check{S}(f_t^Q)} f_{t+1}^Q - \frac{1}{2}Q_p^{-1}(t+2)\sum_{i=0}^{k-2} b^{-i}.$$

Substituting the first term on the right-hand side, we derive

$$
\begin{aligned}
\sup_{\boldsymbol{x}_{t+k}\in\check{S}(f^Q_{t+k-1})} f^Q_{t+k} &= f^Q_t + \frac{1}{2}Q_p^{-1}(t+1) - \frac{1}{2}b^{-1}Q_p^{-1}(t+1)\frac{1-b^{-k+1}}{1-b^{-1}} \\
&= f^Q_t + \frac{1}{2}Q_p^{-1}(t+1)\left(\frac{1-2b^{-1}+b^{-k}}{1-b^{-1}}\right) \\
&= f^Q_t + \frac{1}{2}Q_p^{-1}(t)b^{-1}\left(\frac{1-2b^{-1}+b^{-2}-b^{-2}+b^{-k}}{1-b^{-1}}\right) \\
&= f^Q_t + \frac{1}{2}Q_p^{-1}(t)b^{-1}\left(\frac{(1-b^{-1})^2-b^{-2}(1-b^{-k+2})}{1-b^{-1}}\right) \\
&= f^Q_t + \frac{1}{2}Q_p^{-1}(t)\frac{1}{b}\left((1-b^{-1})-\frac{(1-b^{-k+2})}{b^2(1-b^{-1})}\right) \\
&< f^Q_t + \frac{1}{2}Q_p^{-1}(t)\frac{1}{b}\left(1-\frac{1}{b}+\frac{1}{b(1-b)}\right) \\
&= f^Q_t + \frac{1}{2}Q_p^{-1}(t)\frac{1}{b}\left(1-\frac{1}{b}+\frac{1}{b}+\frac{1}{1-b}\right) \\
&= f^Q_t + \frac{1}{2}Q_p^{-1}(t)\frac{1}{b}\left(\frac{b-2}{b-1}\right).
\end{aligned}
\tag{a8}
$$

Consequently, the limit supremum satisfies:

$$
\lim_{s\to\infty}\sup_{\boldsymbol{x}_s\in\check{S}(f^Q_s)} f^Q_s = f^Q_t + \frac{1}{2}Q_p^{-1}(t)\frac{1}{b}\left(\frac{b-2}{b-1}\right).
\tag{a9}
$$

Furthermore, when $b=2$, equation equation a9 straightforwardly implies:

$$
\lim_{s\to\infty}\sup_{\boldsymbol{x}_s\in\check{S}(f^Q_s)} f^Q_s = f^Q_t.
\tag{a10}
$$

By applying Markov's inequality for the Lebesgue measure $m$, we obtain:

$$
\begin{aligned}
m(\{\boldsymbol{x}_s, s\in\mathbb{Z}^+[t,t_e) \,||\, &\lim_{s\to\infty}\sup_{\boldsymbol{x}_s\in\check{S}(f^Q_s)} f^Q_s - f^Q_t| > \epsilon,\ \forall\epsilon > 0\}) \\
&\leq \frac{1}{\epsilon}\int_{\{\boldsymbol{x}_s,s\in[t,t_e)\}} |\lim_{s\to\infty}\sup_{\boldsymbol{x}_s\in\check{S}(f^Q_s)} f^Q_s - f^Q_t|dm \\
&\leq \frac{1}{\epsilon}\int_{\{\boldsymbol{x}_s,s\in[t,t_e)\}} \lim_{s\to\infty}|\sup_{\boldsymbol{x}_s\in\check{S}(f^Q_s)} f^Q_s - f^Q_t|dm \\
&= \frac{1}{\epsilon}\int_{\{\boldsymbol{x}_s,s\in[t,t_e)\}} \frac{1}{2}Q_p^{-1}(t)\left|\frac{1}{b}\left(\frac{b-2}{b-1}\right)\right|_{b=2}dm = 0
\end{aligned}
\tag{a11}
$$

Therefore, the candidate set generated by Algorithm 1:

$$
\{\boldsymbol{x}_s|f^Q_s \leq f^Q_t\} = \{\boldsymbol{x}_s|f^Q_s < f^Q_t\}\cup\{\boldsymbol{x}_s|f^Q_s = f^Q_t\} = \{\boldsymbol{x}_s|f^Q_s < f^Q_t\}\cup\emptyset.
\tag{a12}
$$

According to equation a12, the state $\boldsymbol{x}_s$ almost surely escapes the set of points satisfying the equality case. $\square$

We summarize the convergence property of Algorithm 1 based on the monotone decreasing sequence derived from the inequality case described in Lemma C.1 and the equality case property given in Lemma C.2, as follows:

**Theorem C.1.** The sequence $\{m(\check{S}(f^Q_s)\}^\infty_{s=t_0}$ is non-strictly monotonically decreasing.

**Proof of Theorem** The proof follows directly Consider the sublevel set $\check{S}(f^Q_s)$ for $s\in\mathbb{Z}[t_0,\infty)$, generated by Algorithm 1, where $s$ denotes the time index associated with candidate updates. We

partition $\check{S}(f_s^Q)$ into cases according to whether $f_s^Q < f_t^Q$ or $f_s^Q = f_t^Q$. By Lemma C.1 and Lemma C.2, the case $f_s^Q < f_t^Q$ yields a monotonically decreasing sequence $\{m(\check{S}(f_s^Q))\}_{s=\bar{t}}$, while the case $f_s^Q = f_t^Q$ gives a constant sequence $\{m(\check{S}(f_s^Q))\}s = t$. By combining both the inequality and equality cases across all time indices, we obtain a unified sequence $\{m(\check{S}(f_t^Q))\}_t$, which is non-strictly monotonically decreasing. $\square$

## C.2 AUXILIARY DESCRIPTION FOR CHAPTER 3.1 "HAMILTONIAN BASED ANALYSIS", AND THE PROOF OF THEOREM 3.1

We provide detailed explanations of equations equation 9 to equation 12 in Section 3.1 using the following lemmas. First, we examine equation equation 9, which presents the fundamental equation for the gradient flow.

**Lemma C.3.** Under the equality case for $t > t_0$ and given the state vector evolution $d\boldsymbol{x}_t = -\nabla_{\boldsymbol{x}} f(\boldsymbol{x}_t) dt$ as in Assumption 3, simplifying the cost function over the region defined by the constraint yields:

$$J(\boldsymbol{x}_t, t) \triangleq f(\boldsymbol{x}_t) - f(\boldsymbol{x}_{t_0}) = -\int_{t_0}^t \|\nabla_{\boldsymbol{x}} f((\boldsymbol{x}_\tau)\|^2 d\tau, \quad \text{Subject to } |f(\boldsymbol{x}_t) - f_{t_0}^Q| \leq \tfrac{1}{2} Q_p^{-1}(t_0), \ t \geq t_0.$$

$$(14)$$

**Proof of Lemma**   The proof is straightforward. Due to the quantized nature of the objective function, the domain may not be simply connected, in contrast to standard assumptions in conventional nonlinear optimization. This implies that the gradient of the objective function may not constitute a conservative vector field, resulting in a gradient-flow dissipative system.

We can thus express the difference $f(\boldsymbol{x}_t) - f(\boldsymbol{x}_{t_0})$ as a path integral:

$$f(\boldsymbol{x}_t) - f(\boldsymbol{x}_{t_0}) = \int_c df = \int_c d\boldsymbol{x}_t \cdot \nabla_{\boldsymbol{x}} f(\boldsymbol{x}_t), \tag{a13}$$

where $c : [t_0, t] \to \mathcal{C} \subset \mathbb{R}$ denotes a parameterized path in the domain.

From the assumption as $d\boldsymbol{x}_t = -\nabla_{\boldsymbol{x}} f(\boldsymbol{x}_t) dt$, we obtain

$$f(\boldsymbol{x}_t) - f(\boldsymbol{x}_{t_0}) = \int_c -d\tau \nabla_{\boldsymbol{x}_\tau} f(\boldsymbol{x}_\tau) \cdot \nabla_{\boldsymbol{x}} f(\boldsymbol{x}_\tau) = -\int_{t_0}^t \|\nabla_{\boldsymbol{x}_\tau} f(\boldsymbol{x}_\tau)\|^2 d\tau. \tag{a14}$$

$\square$

According to Lemma C.3, equation 9 represents the path $c$ that minimizes the integral $\int_c \nabla_{\boldsymbol{x}} f(\boldsymbol{x}_t) \cdot d\boldsymbol{x}_t$ Thus, we rewrite equation a14 as:

$$\min_c |f(\boldsymbol{x}_t) - f(\boldsymbol{x}_{t_0})| = \int_{t_0}^t \|\nabla_{\boldsymbol{x}} f(\boldsymbol{x}_\tau)\|^2 d\tau. \tag{a15}$$

This equation a15 is equal to equation 9 under the quantization constraint.

To derive the HJB equation for quantization-based optimization, we set the Lagrangian of the quantized objective function at the initial point $x_0$, as shown in equation 10, subject to the quantization constraint. Following the definition of the HJB equation, we obtain the form given in equation 11. Under Assumption 3, equation 12 follows immediately. Finally, we analyze the convergence behavior of quantization-based optimization in the regime where the quantization step size is vanishingly small and the time index is sufficiently large.

**Theorem 3.1.** The derivative of $V^Q$ with respect to $t$ tends to zero, i.e., $\partial_t V^Q \to 0$, as the quantization step size decreases to zero with increasing $t$, that is, as $Q_p^{-1}(t) \to 0$.

**Proof of Theorem**   The Lagrangian given by equation 10 is as follows:

$$L(\boldsymbol{x}_t, \lambda) = \|\nabla_x f_t^Q\|^2 + \lambda \left( \tfrac{1}{4} Q_p^{-2}(t) - (f_t^Q - f(\boldsymbol{x}_t))^2 \right), \quad \forall \boldsymbol{x}_t \in S(f_t^Q).$$

Due to the quantization, the norm of the gradient term is zero, i.e., $\|\nabla_x f_t^Q\| = 0$, and the definition of quantization equation 3 leads to $f_t^Q - f(\boldsymbol{x}_t) = \varepsilon^q Q_p^{-1}(t)$.

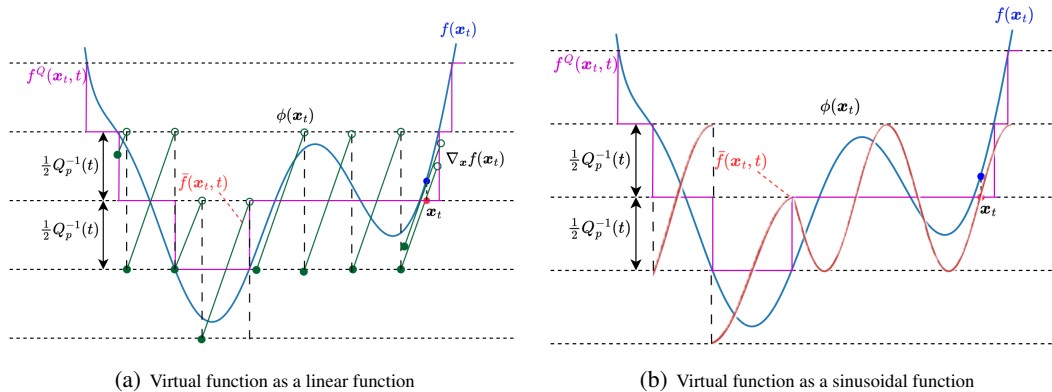

(a) Virtual function as a linear function      (b) Virtual function as a sinusoidal function

Figure 3: Illustrative diagram of virtual functions used for analysis. Left: Linear virtual function — at $\boldsymbol{x}_t$, the gradient of the real objective function matches that of the virtual function. Right: Sinusoidal virtual function — at $\boldsymbol{x}_t$, the gradient of the real objective function again matches that of the virtual function.

Substituting these into equation 10, we obtain

$$L(\boldsymbol{x}_t, \lambda) = \lambda Q_p^{-2}(t)\left(\tfrac{1}{4} - (\varepsilon^q)^2\right). \tag{a16}$$

Taking the expectation of $L(\boldsymbol{x}_t, \lambda)$ with respect to the fraction for the quantization $\varepsilon^q$, we derive

$$\mathbb{E}_{\varepsilon^q} L(\boldsymbol{x}_t, \lambda) = Q_p^{-2}(t)\frac{\lambda}{6}. \tag{a17}$$

Under the assumption $\nabla_{\boldsymbol{x}}\partial_t V^Q(\boldsymbol{x}_t, t) = 0$, the Hamiltonian equation 12 becomes:

$$\mathbb{E}_{\varepsilon^q}\partial_t V^Q(\boldsymbol{x}_t, t) = -\min_{\boldsymbol{x}_t \in S(f_{t_0}^Q)} \mathbb{E}_{\varepsilon^q} L(\boldsymbol{x}, \lambda) = -Q_p^{-2}(t)\frac{\lambda}{6}. \tag{a18}$$

Consequently, as $Q_p^{-1}(t)$ decreases monotonically to zero with increasing $t$, $\mathbb{E}_{\varepsilon^q}\partial_t V^Q$ converges to zero. $\square$

### C.3   Auxiliary Description for Chapter 3.2 "Quantum Mechanical and Thermodynamical Analysis", and The Proof of Theorems

In this section, we provide a detailed explanation of equations equation 15–equation 17, including Theorem 3.2 and Theorem 3.3.

**Lemma C.4.** Under the framework of equation 8 and equation 13, if the score function is defined as $V^Q(\boldsymbol{x}_t, t) = -h\log\rho(\boldsymbol{x}_t, t)$ and the transformed function $\bar{f}(\boldsymbol{x}_t, t)$, given in equation 14, satisfies Assumption 4, then the HJB equation in equation 12 can be expressed as follows:

$$\partial_t V(t, \boldsymbol{x}) = \nabla_{\boldsymbol{x}} V(t, \boldsymbol{x}_t) \cdot \nabla_{\boldsymbol{x}} \bar{f}(\boldsymbol{x}_t, t) - \|\nabla_{\boldsymbol{x}} \bar{f}(\boldsymbol{x}_t, t)\|^2, \tag{a19}$$

where Assumption 4 implies $V^Q(\boldsymbol{x}_t, t) = V(\boldsymbol{x}_t, t)$.

**Proof of Lemma**   From equation 10 and equation 12, we directly define the Hamiltonian for all $\boldsymbol{x}_t \in S(f_t^Q)$ as

$$H(\boldsymbol{x}_t, t) = -\nabla_{\boldsymbol{x}} V^Q(\boldsymbol{x}_t, t) \cdot \nabla_{\boldsymbol{x}} \bar{f}(\boldsymbol{x}_t, t) + \min_{\boldsymbol{x}_t \in S(f_t^Q)} \left\{ \|\nabla_{\boldsymbol{x}} f_t^Q\|^2 + \lambda\left(\tfrac{1}{4}Q_p^{-2}(t) - (f_t^Q - \bar{f}(\boldsymbol{x}_t, t))^2\right) \right\}. \tag{a20}$$

Under Assumption 4, we replace $V^Q(\boldsymbol{x}_t, t)$ and $f_t^Q$ with the $\psi$-based virtual functions $V(\boldsymbol{x}_t, t)$ and $\bar{f}(\boldsymbol{x}_t, t)$. Furthermore, since the $\psi$-based objective function automatically satisfies

$$|f_t^Q - \bar{f}(\boldsymbol{x}_t, t)| \le \tfrac{1}{2}Q_p^{-1}(t) \tag{a21}$$

by Assumption 4, the last term in equation a20 can be eliminated.

Combining these results, we derive equation 15 as

$$\partial_t V(\boldsymbol{x}_t, t) = \nabla_{\boldsymbol{x}} V(\boldsymbol{x}_t, t) \cdot \nabla_{\boldsymbol{x}} \bar{f}(\boldsymbol{x}_t, t) - \|\nabla_{\boldsymbol{x}} \bar{f}(\boldsymbol{x}_t, t)\|^2. \tag{a22}$$

□

In Lemma C4.3, the gradient of the transformed objective function $\bar{f} : \mathbb{R}^d \times \mathbb{R} \mapsto \mathbb{R}^+$ in the HJB equation a22 depends only on the state $\boldsymbol{x}_t$. Since the time parameter $t$ affects the quantization step size $Q_p^{-1}(t)$ and is treated as a constant under Assumption 4, we can replace $\bar{f}$ with a virtual objective function $f : \mathbb{R}^d \mapsto \mathbb{R}^+$ that satisfies the quantization condition.

Although it is possible to introduce a new notation for the virtual function, the only difference between the original and virtual objective functions lies in the satisfaction of the quantization constraint for all $\boldsymbol{x}_t \in S(f_t^Q)$. Therefore, assuming that the original objective function satisfies this constraint, there is no need to distinguish between the original and virtual functions. Accordingly, we use the notation $f$ instead of $\bar{f}$ throughout this section without further distinction.

Following this notation, equation equation 15 can be expressed as:

$$\partial_t V(\boldsymbol{x}_t, t) = \nabla_{\boldsymbol{x}} V(\boldsymbol{x}_t, t) \cdot \nabla_{\boldsymbol{x}} f(\boldsymbol{x}_t) - \|\nabla_{\boldsymbol{x}} f(\boldsymbol{x}_t)\|^2.$$

Based on the HJB equation 15 and brief notations described in the manuscript, we provide the proof of Theorem 3.2 and Theorem 3.3.

**Theorem 3.2.** Under the definitions and assumptions for the score function and the observer function, we derive the following thermodynamic evolution equation for $\rho_t \triangleq \rho(\boldsymbol{x}_t, t)$:

$$\partial_t \rho_t = \nabla_{\boldsymbol{x}} \cdot (\nabla_{\boldsymbol{x}} f \; \rho_t) + h^{-1}(\|\nabla_{\boldsymbol{x}} f\|^2 - h\Delta_{\boldsymbol{x}} f)\rho_t \tag{16}$$

**Proof of Theorem** For brevity, let $V(\boldsymbol{x}_t, t)$ and $f(\boldsymbol{x}_t)$ be denoted as $V_t$ and $f$, respectively.

From earlier results, the relations $\partial_t V_t = -h\rho_t^{-1}\partial_t \rho_t$ and $\nabla_{\boldsymbol{x}} V_t = -h\rho_t^{-1}\nabla_{\boldsymbol{x}} \rho_t$ allow us to substitute $\partial_t V_t$ and $\nabla_{\boldsymbol{x}} V_t$ into equation 15, yielding

$$\partial_t V_t = \nabla_{\boldsymbol{x}} V_t \cdot \nabla_{\boldsymbol{x}} f - \|\nabla_{\boldsymbol{x}} f\|^2 \implies -h\,\rho_t^{-1}\partial_t \rho_t = -h\rho_t^{-1}\nabla_{\boldsymbol{x}}\rho_t \cdot \nabla_{\boldsymbol{x}} f - h^{-1}\|\nabla_{\boldsymbol{x}} f\|^2$$
$$\implies \partial_t \rho_t = \nabla_{\boldsymbol{x}}\rho_t \cdot \nabla_{\boldsymbol{x}} f + h^{-1}\rho_t \|\nabla_{\boldsymbol{x}} f\|^2. \tag{a23}$$

Introducing the Laplacian $\Delta_{\boldsymbol{x}} f \in \mathbb{R}$ and the term $\Delta_{\boldsymbol{x}} f \cdot \rho_t$, we add and subtract $\Delta_{\boldsymbol{x}} f \rho_t$ on the right-hand side of equation a23 to derive

$$\partial_t \rho_t = \nabla_{\boldsymbol{x}}\rho_t \cdot \nabla_{\boldsymbol{x}} f + \Delta_{\boldsymbol{x}} f \cdot \rho_t - \Delta_{\boldsymbol{x}} f \cdot \rho_t + h^{-1}\rho_t \|\nabla_{\boldsymbol{x}} f\|^2$$
$$= \nabla_{\boldsymbol{x}} \cdot (\nabla_{\boldsymbol{x}} f \; \rho_t) - \Delta_{\boldsymbol{x}} f \cdot \rho_t + h^{-1}\rho_t \|\nabla_{\boldsymbol{x}} f\|^2 \tag{a24}$$
$$= \nabla_{\boldsymbol{x}} \cdot (\nabla_{\boldsymbol{x}} f \; \rho_t) + h^{-1}(\|\nabla_{\boldsymbol{x}} f\|^2 - h\,\Delta_{\boldsymbol{x}} f)\rho_t.$$

This completes the proof. □

For convenience, we denote the wave function as $\psi_t \triangleq \psi(\boldsymbol{x}_t, t)$ throughout the following lemma.

**Lemma C.5.** The zero-order Witten-Laplacian $\Delta_{f,h}^{(0)}$ defined in Definition 4 is given by

$$\Delta_{f,h}^{(0)} u = -h^2 \Delta_{\boldsymbol{x}} u + (\|\nabla_{\boldsymbol{x}} f\|^2 - h\Delta_{\boldsymbol{x}} f)u, \tag{a25}$$

where $u : \mathbb{R}^d \times \mathbb{R} \to \mathbb{R}$ or $\mathbb{C}$.

**Proof of Lemma** We compute the first differential $d_{f,h}^{(0)}$ for $u$ as

$$d_{f,h}^{(0)} u = e^{-f/h}(h\nabla_{\boldsymbol{x}})e^{f/h}\,u$$
$$= e^{-f/h}\left(h\nabla_{\boldsymbol{x}}e^{f/h}\,u + h\,e^{f/h}\,\nabla_{\boldsymbol{x}}u\right)$$
$$= e^{-f/h}\left(u\,h\,\frac{1}{h}\nabla_{\boldsymbol{x}}f\,e^{f/h} + he^{f/h}\nabla_{\boldsymbol{x}}u\right)$$
$$= (\nabla_{\boldsymbol{x}}f + h\nabla_{\boldsymbol{x}})\,u.$$

Next, the adjoint operator $d_{f,h}^{(0)*}$ is computed as

$$
\begin{aligned}
d_{f,h}^{(0)*} \nabla_{\boldsymbol{x}} u &= -e^{f/h}(h\nabla_{\boldsymbol{x}}\cdot)e^{-f/h}\nabla_{\boldsymbol{x}} u \\
&= -e^{f/h}h(\nabla_{\boldsymbol{x}}e^{-f/h}\cdot\nabla_{\boldsymbol{x}} u + e^{-f/h}\nabla_{\boldsymbol{x}}\cdot\nabla_{\boldsymbol{x}} u) \\
&= -e^{f/h}h\left(-\frac{1}{h}e^{-f/h}\nabla_{\boldsymbol{x}}f\cdot\nabla_{\boldsymbol{x}} u + e^{-f/h}\Delta_{\boldsymbol{x}} u\right) \\
&= (\nabla_{\boldsymbol{x}}f\cdot\nabla_{\boldsymbol{x}} - h\Delta_{\boldsymbol{x}})u
\end{aligned}
$$

Thus, we obtain the operator pair:

$$
\begin{aligned}
d_{f,h}^{(0)}u &= (\nabla_{\boldsymbol{x}}f + h\nabla_{\boldsymbol{x}})u & \because d_{f,h}^{(0)} : \mathbb{R} \to \mathbb{R}^d \\
d_{f,h}^{(0)*}\nabla_{\boldsymbol{x}} u &= (\nabla_{\boldsymbol{x}}f - h\nabla_{\boldsymbol{x}})\cdot\nabla_{\boldsymbol{x}} u & \because d_{f,h}^{(0)*} : \mathbb{R}^d \to \mathbb{R}
\end{aligned}
\tag{a26}
$$

Applying these operators sequentially, we derive

$$
\begin{aligned}
d_{f,h}^{(0)*}d_{f,h}^{(0)}u &= (\nabla_{\boldsymbol{x}}f - h\nabla_{\boldsymbol{x}})\cdot(\nabla_{\boldsymbol{x}}f + h\nabla_{\boldsymbol{x}})u \\
&= (\nabla_{\boldsymbol{x}}f - h\nabla_{\boldsymbol{x}})\cdot(u\nabla_{\boldsymbol{x}}f + h\nabla_{\boldsymbol{x}} u) \\
&= u\nabla_{\boldsymbol{x}}f\cdot\nabla_{\boldsymbol{x}}f + h\nabla_{\boldsymbol{x}}f\cdot\nabla_{\boldsymbol{x}} u - h\nabla_{\boldsymbol{x}}\cdot(u\nabla_{\boldsymbol{x}}f) - h^2\nabla_{\boldsymbol{x}}\cdot\nabla_{\boldsymbol{x}} u \\
&= \|\nabla_{\boldsymbol{x}}f\|^2 u + h\nabla_{\boldsymbol{x}}f\cdot\nabla_{\boldsymbol{x}} u - h\nabla_{\boldsymbol{x}}u\cdot\nabla_{\boldsymbol{x}}f - hu\nabla_{\boldsymbol{x}}\cdot\nabla_{\boldsymbol{x}}f - h^2\Delta_{\boldsymbol{x}} u \\
&= \|\nabla_{\boldsymbol{x}}f\|^2 u + h(\nabla_{\boldsymbol{x}}f\cdot\nabla_{\boldsymbol{x}} u - \nabla_{\boldsymbol{x}}u\cdot\nabla_{\boldsymbol{x}}f) - h\Delta_{\boldsymbol{x}}fu - h^2\Delta_{\boldsymbol{x}} u \\
&= -h^2\Delta_{\boldsymbol{x}}u + (\|\nabla_{\boldsymbol{x}}f\|^2 - h\Delta_{\boldsymbol{x}}f)u
\end{aligned}
$$

This establishes the zero-order Witten-Laplacian as

$$
\Delta_{f,h}^{(0)}u = -h^2\Delta_{\boldsymbol{x}}u + (\|\nabla_{\boldsymbol{x}}f\|^2 - h\Delta_{\boldsymbol{x}}f)u
\tag{a27}
$$

$\square$

Before presenting the proof of Theorem 3.3, we derive fundamental formulations related to the Witten Laplacian and the Fokker–Planck equation. Based on these formulations, we establish the correspondence between the Schrödinger equation and the Fokker–Planck equation through the following lemmas.

**Lemma C.6.** For a potential energy $\overline{V} : \mathbb{R}^d \to \mathbb{R}$ defined as $\overline{V}(\boldsymbol{x}) \triangleq \frac{m}{2}(\|\nabla_{\boldsymbol{x}}f\|^2 - h\Delta_{\boldsymbol{x}}f)$, the Schrödinger equation

$$
i\hbar\frac{\partial\psi_t}{\partial t} = -\frac{\hbar^2}{2m}\Delta_{\boldsymbol{x}}\psi_t + \bar{V}\psi_t
\tag{a28}
$$

can be rewritten equivalently as

$$
i\hbar\frac{\partial\psi_t}{\partial t} = \frac{m}{2}\Delta_{f,h}^{(0)}\psi_t = \hat{H}\psi_t,
\tag{a29}
$$

in a complex Hilbert space), $\hat{H}$ denotes the Hamiltonian operator, and $m$ is a particle mass.

**Proof of Lemma** We rewrite the Schrödinger equation as:

$$
\frac{\partial\psi_t}{\partial t} = \frac{i\hbar}{2m}\Delta_{\boldsymbol{x}}\psi_t - \frac{i}{\hbar}\overline{V}(\boldsymbol{x}) = \frac{i\hbar}{2m}\Delta_{\boldsymbol{x}}\psi_t + \frac{m}{i\hbar}\frac{1}{m}\overline{V}(\boldsymbol{x})
\tag{a30}
$$

Let $h = i\hbar/m$, where $\hbar \in \mathbb{R}$ denotes the reduced Planck's constant and $m$ denotes the mass of a particle. By substituting the potential $\overline{V}$ and the expression for $h$ into equation a30, we obtain

$$
\frac{\partial\psi_t}{\partial t} = \frac{h}{2}\Delta_{\boldsymbol{x}}\psi_t + \frac{1}{2h}(\|\nabla_{\boldsymbol{x}}f\|^2 - h\Delta_{\boldsymbol{x}}f)\psi_t
\tag{a31}
$$

By definition, $h$ is a real number. However, substituting $h$ with $i\hbar/m$ introduces ambiguity due to its complex nature. Although the square of a real number $h$ should satisfy $h^2 > 0$, directly substituting $h = i\hbar/m$ yields $h^2 = (i)^2\hbar^2/m^2 = -(\hbar/m)^2$, which is negative. This result contradicts the fundamental property that the square of a real number is always positive. Therefore, as per elementary

properties of complex numbers, replacing $h$ by a complex parameter $-ia$ (with $a \in \mathbb{R}$) leads to $h^2 = (ia)(-ia) = a^2 > 0$, since $(i)(-i) = 1$. Consequently, when computing $\frac{1}{h}h^2$ for $h = ia$, the result is $-h^2$. Applying this result to equation a31, we get

$$
\begin{aligned}
\frac{\partial \psi_t}{\partial t} &= \frac{1}{2h}\left(-h^2 \Delta_{\boldsymbol{x}} \psi_t + (\|\nabla_{\boldsymbol{x}} f\|^2 - h\Delta_{\boldsymbol{x}} f)\right) \\
&= \frac{m}{2i\hbar} \Delta_{f,h}^{(0)} \psi_t
\end{aligned}
\tag{a32}
$$

Therefore,

$$
i\hbar \frac{\partial \psi_t}{\partial t} = \frac{m}{2} \Delta_{f,h}^{(0)} \psi_t.
\tag{a33}
$$

By the definition of the Schrödinger equation, we have

$$
\Delta_{f,h}^{(0)} = \frac{2}{m} \hat{H}.
\tag{a34}
$$

$\square$

Equation equation a32 represents the fundamental form of the Schrödinger equation expressed using the Witten Laplacian. Based on Lemma C.6, we establish the connection between the standard Fokker–Planck equation and the Schrödinger equation in the following lemma.

**Lemma C.7.** The Schrödinger equation a28 induces the following standard Fokker-Planck equation through the Witten-Laplacian:

$$
\partial_t \rho_t = \nabla_{\boldsymbol{x}} \cdot (\rho_t \nabla_{\boldsymbol{x}} f) + \frac{h}{2} \Delta_{\boldsymbol{x}} \rho_t
\tag{a35}
$$

**Proof of Lemma** According to equation 13 in Assumption 4, we can immediately derive the partial derivative of $\rho_t$ with respect to $t$ such that

$$
\partial_t \rho_t = g \, \partial_t \psi_t.
\tag{a36}
$$

To elaborate on equation 16 in more detail, we compute the gradient of $\rho_t$ as

$$
\nabla_{\boldsymbol{x}} \rho_t = \nabla_{\boldsymbol{x}}(\psi_t \cdot g) = \nabla_{\boldsymbol{x}} \psi_t \cdot g - \psi_t \cdot \frac{\nabla_{\boldsymbol{x}} f}{h} g = \left(\nabla_{\boldsymbol{x}} \psi_t - \frac{\psi_t}{h} \nabla_{\boldsymbol{x}} f\right) g.
\tag{a37}
$$

The Laplacian of $\rho_t$ is then derived as

$$
\begin{aligned}
\Delta_{\boldsymbol{x}} \rho_t &= \nabla_{\boldsymbol{x}} \cdot \left(\nabla_{\boldsymbol{x}} \psi_t - \frac{\psi_t}{h} \nabla_{\boldsymbol{x}} f\right) g \\
&= \left(\Delta_{\boldsymbol{x}} \psi_t - \frac{1}{h}(\nabla_{\boldsymbol{x}} \psi_t \cdot \nabla_{\boldsymbol{x}} f + \psi_t \Delta_{\boldsymbol{x}} f)\right) + \left(\nabla_{\boldsymbol{x}} \psi_t - \frac{\psi_t}{h} \nabla_{\boldsymbol{x}} f\right) \frac{-\nabla_{\boldsymbol{x}} f}{h}\right) g \\
&= \left(\Delta_{\boldsymbol{x}} \psi_t - \frac{2}{h} \nabla_{\boldsymbol{x}} \psi_t \cdot \nabla_{\boldsymbol{x}} f - \frac{1}{h} \psi_t \Delta_{\boldsymbol{x}} f + \frac{\psi_t}{h^2} \|\nabla_{\boldsymbol{x}} f\|^2\right) g \\
&= \left(\Delta_{\boldsymbol{x}} \psi_t - \frac{2}{h} \nabla_{\boldsymbol{x}} \psi_t \cdot \nabla_{\boldsymbol{x}} f\right) g + \frac{1}{h^2}(\|\nabla_{\boldsymbol{x}} f\|^2 - h\Delta_{\boldsymbol{x}} f)\psi_t \, g.
\end{aligned}
\tag{a38}
$$

Using equation a37, we compute $\nabla_{\boldsymbol{x}}(\rho_t \cdot \nabla_{\boldsymbol{x}} f)$ as follows:

$$
\begin{aligned}
\nabla_{\boldsymbol{x}}(\rho_t \cdot \nabla_{\boldsymbol{x}} f) &= \nabla_{\boldsymbol{x}} \rho_t \cdot \nabla_{\boldsymbol{x}} f + \rho_t \Delta_{\boldsymbol{x}} f \\
&= \left(\left(\nabla_{\boldsymbol{x}} \psi_t - \frac{\psi_t}{h} \nabla_{\boldsymbol{x}} f\right) \cdot \nabla_{\boldsymbol{x}} f + \psi_t \Delta_{\boldsymbol{x}} f\right) g \\
&= \left(\nabla_{\boldsymbol{x}} \psi_t \cdot \nabla_{\boldsymbol{x}} f - \frac{1}{h}(\|\nabla_{\boldsymbol{x}} f\|^2 - h\Delta_{\boldsymbol{x}} f)\psi_t\right) g.
\end{aligned}
\tag{a39}
$$

We express the final term of equation a38 as

$$
\Delta_{\boldsymbol{x}} \rho_t = \left(\Delta_{\boldsymbol{x}} \psi_t - \frac{2}{h} \nabla_{\boldsymbol{x}} \psi_t \cdot \nabla_{\boldsymbol{x}} f\right) g + \frac{1}{h^2}(\|\nabla_{\boldsymbol{x}} f\|^2 - h\Delta_{\boldsymbol{x}} f)\psi_t \, g.
\tag{a40}
$$

Multiplying both sides of equation a40 by $-h^2$, we obtain

$$
\begin{aligned}
-h^2 \Delta_{\boldsymbol{x}} \rho_t &= -h^2 g \Delta_{\boldsymbol{x}} \psi_t + 2gh \nabla_{\boldsymbol{x}} \psi_t \cdot \nabla_{\boldsymbol{x}} f - (\|\nabla_{\boldsymbol{x}} f\|^2 - h\Delta_{\boldsymbol{x}} f)\psi_t\, g \\
&= -h^2 g \Delta_{\boldsymbol{x}} \psi_t + (\|\nabla_{\boldsymbol{x}} f\|^2 - h\Delta_{\boldsymbol{x}} f)\psi_t\, g + 2gh \nabla_{\boldsymbol{x}} \psi_t \cdot \nabla_{\boldsymbol{x}} f - 2(\|\nabla_{\boldsymbol{x}} f\|^2 - h\Delta_{\boldsymbol{x}} f)\psi_t\, g \\
&= -g \Delta_{f,h}^{(0)} \psi_t + 2h \left( \nabla_{\boldsymbol{x}} \psi_t \cdot \nabla_{\boldsymbol{x}} f - h^{-1}(\|\nabla_{\boldsymbol{x}} f\|^2 - h\Delta_{\boldsymbol{x}} f)\psi_t \right) g.
\end{aligned}
\tag{a41}
$$

Substituting equation a39 into this result yields

$$
-h^2 \Delta_{\boldsymbol{x}} \rho_t = -g \Delta_{f,h}^{(0)} \psi_t + 2h \nabla_{\boldsymbol{x}} \cdot (\rho_t \nabla_{\boldsymbol{x}} f).
\tag{a42}
$$

Dividing both sides by $2h$ and rearranging terms, we have:

$$
\frac{g}{2h} \Delta_{f,h}^{(0)} \psi_t = \nabla_{\boldsymbol{x}} \cdot (\rho_t \nabla_{\boldsymbol{x}} f) + \frac{h}{2} \Delta_{\boldsymbol{x}} \rho_t.
\tag{a43}
$$

From equation a32, the Schrödinger equation formulated via the Witten-Laplacian is

$$
\partial_t \psi_t = \frac{1}{2h} \Delta_{f,h}^{(0)} \psi_t.
\tag{a44}
$$

The relation between $\partial_t \rho_t$ and $\partial_t \psi_t$ implies

$$
\frac{g}{2h} \Delta_{f,h}^{(0)} \psi_t = g\, \partial_t \psi_t = \partial_t \rho_t.
\tag{a45}
$$

Finally, substituting equation a45 into equation a43 completes the proof. $\square$

**Theorem 3.3.** Given the thermodynamic evolution described in equation 16, replacing $h$ with $i\hbar/2m$ yields the following Schrödinger equation for $\psi_t$:

$$
i\hbar\, \partial_t \psi_t = -\frac{\hbar^2}{2m} \Delta_{\boldsymbol{x}} \psi_t + \frac{m}{2} \left( \|\nabla_{\boldsymbol{x}} f\|^2 - \frac{i\hbar}{m} \Delta_{\boldsymbol{x}} f \right) \psi_t,
\tag{17}
$$

where $\hbar$ denotes the reduced Planck constant and $m$ is the mass of a quantum particle.

**Proof of Theorem** By Lemma C.6 and Lemma C.7, the result follows directly. Consider the standard Fokker–Planck equation:

$$
\partial_t \rho_t = \nabla_{\boldsymbol{x}} \cdot (\rho_t \nabla_{\boldsymbol{x}} f) + \frac{1}{2} \tilde{h} \Delta_{\boldsymbol{x}} \rho_t.
\tag{a46}
$$

For the stationary solution $\rho_t \propto \exp(-2f/\tilde{h})$, we compute the Laplacian:

$$
\Delta_{\boldsymbol{x}} \rho_t = \nabla_{\boldsymbol{x}} \cdot \left( -\frac{2}{\tilde{h}} \nabla_{\boldsymbol{x}} f\, \rho_t \right) = \frac{4}{\tilde{h}^2} \left( \|\nabla_{\boldsymbol{x}} f\|^2 - \frac{\tilde{h}}{2} \Delta_{\boldsymbol{x}} f \right).
\tag{a47}
$$

Multiplying both sides by $\tilde{h}/2$, we obtain:

$$
\frac{\tilde{h}}{2} \Delta_{\boldsymbol{x}} \rho_t = \frac{2}{\tilde{h}} \left( \|\nabla_{\boldsymbol{x}} f\|^2 - \frac{\tilde{h}}{2} \Delta_{\boldsymbol{x}} f \right).
\tag{a48}
$$

Substituting $\tilde{h} = 2h$, we get:

$$
h \Delta_{\boldsymbol{x}} \rho_t = h^{-1} \left( \|\nabla_{\boldsymbol{x}} f\|^2 - h\Delta_{\boldsymbol{x}} f \right).
\tag{a49}
$$

This confirms that equation 16 corresponds to a standard Fokker–Planck equation. By identifying $2h = i\hbar/m$, we recover the Schrödinger equation in the form of equation 17. $\square$

**Further Discussion** Theorem 3.3 and Corollary C.7 demonstrate that the dynamics in Algorithm 1 encompass both thermodynamic and quantum mechanical processes. From a quantum mechanical perspective (Theorem 3.3), the quantization-based search is formally equivalent to the quantum-inspired annealing (QIA) process. From a thermodynamic perspective (Corollary C.7), the algorithm can be analyzed using the standard Fokker–Planck equation.

Although the quantum mechanical perspective provides valuable insights, the convergence analysis relies solely on the thermodynamic framework. As discussed in the paragraph titled "Tunneling Effect and Adiabatic Evolution," quantum mechanical analysis offers a framework for escaping local minima via quantum tunneling through energy barriers. However, this analysis describes the dynamics at a specific moment in time and does not capture the global consistency of the algorithm. It can provide bounded estimates for convergence (e.g., in terms of supremum or infimum), but it does not guarantee overall convergence behavior. Since convergence analysis depends on scalar quantities such as energy, objective function values, or vector norms, only thermodynamic analysis or spectral analysis of the Hamiltonian are effective frameworks for this purpose.

### C.4 Auxiliary Description for the paragraph "Tunneling Effect and Adiabatic Evolution"

In this section, we discuss adiabatic evolution, an intrinsic property of quantization-based optimization, as described in equation 19. Adiabatic evolution, which underpins quantum computational optimization algorithms such as QAOA, relies on the quantum tunneling effect. To explore this, we analyze the quantum tunneling effect using equation 17, derived from Algorithm 1. This analysis suggests that quantizing the objective function may serve as an alternative framework for quantum computational optimization.

We begin by proving that Algorithm 1 formalizes adiabatic evolution through equation 19.

**Theorem C.8.** Suppose that there exists the eigenvalue of Hamiltonian $\bar{H}$ in QIA:

$$\bar{H}(\boldsymbol{x}, t) = (1 - \beta(t)) H_P(\boldsymbol{x}, t) + \beta(t) H_B(\boldsymbol{x}, t), \quad \beta(t) \in \mathbb{R}[0, 1], \ t \in [0, T], \tag{19}$$

where $\bar{H} : \mathbb{R}^d \times \mathbb{R} \to \mathbb{R}^+$ represents the real-valued eigenvalue of the Hamiltonian, $H_P : \mathbb{R}^d \times \mathbb{R} \to \mathbb{R}^+$ is the problem Hamiltonian, and $H_B : \mathbb{R}^d \times \mathbb{R} \to \mathbb{R}^+$ is the mixing Hamiltonian.

The quantized objective function $f^Q(\boldsymbol{x}_t)$ derived by the quantization-based optimization is equivalent to equation 18 such that

$$f_t^Q = (1 - b^{-t}) f(\boldsymbol{x}_t) + b^{-t} f_b, \tag{20}$$

where $f_b$ denotes the ground state, i.e., the value of the objective function at the lowest quantization resolution as determined by quantization.

**Proof of Theorem** We rewrite the quantization of the objective function $f$ using the base $b$ for the quantization parameter $Q_p$ as defined in Definition 2:

$$f = f_b + \sum_{k=1}^{\infty} f_k b^{-k}, \quad f_k \in \mathbb{Z}^+[0, b) \tag{a50}$$

Based on equation a50, we decompose the objective function as:

$$f = f_b + \sum_{k=1}^{t-1} f_k b^{-k} + \sum_{k=t}^{\infty} f_k b^{-k} = f^Q - \varepsilon^q Q_p^{-1}(t). \tag{a51}$$

From the simplified form of $Q_p(k)$, we define $f_{t-1}^Q$ as the quantized objective function at the k-th resolution, i.e., $f_{t-1}^Q = f_b + \sum_{k=1}^{t-1} f_k b^{-k}$. Substituting $f_{t-1}^Q$ into equation a51, we express the quantization error as the following power series:

$$f - f_{t-1}^Q = \sum_{k=t}^{\infty} f_k b^{-k} = \text{sign}(f - f_{t-1}^Q) \cdot b^{-(t-1)} \sum_{k=1}^{\infty} \epsilon_k b^{-k}, \tag{a52}$$

where $\epsilon_k \in \mathbb{Z}[0, b)$ is a remaining coefficient for numerical representation. Furthermore, although the sequence $\{\epsilon_k\}_{k=1}^{\infty}$ and $\{f_k\}_{k=t}^{\infty}$, the quantized value $f_b$ are not identical, he quantized value $f_b$

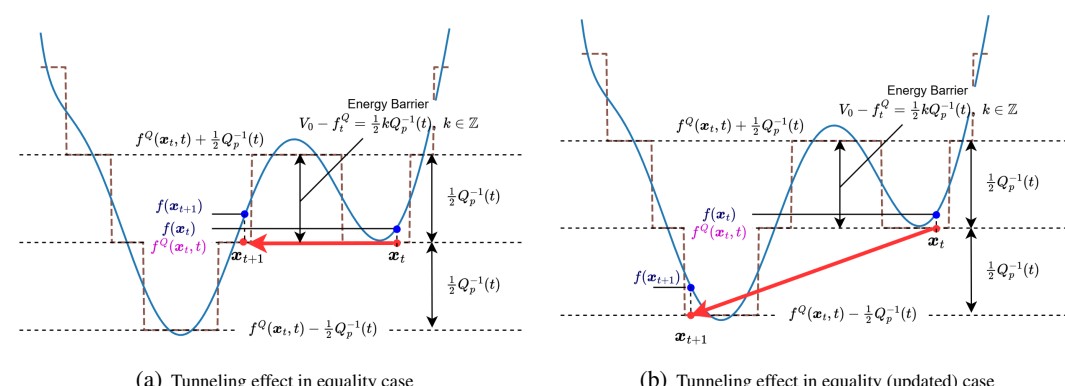

(a) Tunneling effect in equality case    (b) Tunneling effect in equality (updated) case

Figure 4: Conceptual diagram of the tunneling effect. Left: in the equality case, Algorithm 1 tunnels the state vector through the energy barrier at the same quantized objective value. Right: in the inequality case, Algorithm 1 tunnels the state vector through the barrier at a lower quantized objective value.

coincides for both sequences. Therefore, by the above auxiliary equations, we rewrite $f_{t-1}^Q$ such that

$$
\begin{aligned}
f_{t-1}^Q &= f + (f_{t-1}^Q - f) \\
&= f - \mathrm{sign}(f - f_{t-1}^Q) \cdot b^{-(t-1)} \sum_{k=1}^{\infty} \epsilon_k b^{-k} \\
&= f - b^{-(t-1)} \mathrm{sign}(f - f_0^Q) \sum_{k=1}^{\infty} \epsilon_k b^{-k} \\
&= f - b^{-(t-1)}(f - f_b) \\
&= f + b^{-(t-1)}(f_b - f) = b^{-(t-1)} f_b + (1 - b^{-(t-1)}) f.
\end{aligned}
$$

Therefore, we obtain

$$
f_t^Q = b^{-t} f_b + (1 - b^{-t}) f \tag{a53}
$$

Since $b^{-1} < 1$, we designate $(1 - b^{-t})f$ as the eigenvalue of $(1 - \lambda(t))H_0$ and $b^{-t} f_b$ as $\lambda(t) H_k$ such that $f_b = \langle \psi | H_0 | \psi \rangle / \langle \psi | \psi \rangle$. Finally, we set $f_t^Q$ as the eigenvalue of the Hamiltonian $\hat{H}(t)$ for the Schrödinger equation. $\square$

**Time Independent Analysis of Schrödinger Equation for Tunneling Effect**  To focus on the essential aspects, let $\Gamma$ be a one-dimensional eigenspace of the Hamiltonian $\bar{H}(\boldsymbol{x}_t, t)$, which is homeomorphic to $\mathbb{R}$ at a fixed time $t$. Accordingly, the domain of the wave function $\psi(\boldsymbol{x}_t, t)$ is restricted to $\Gamma \cong \mathbb{R}$. To analyze the tunneling effect on $\Gamma$, we consider the following time-independent Schrödinger equation:

$$
\left( -\frac{\hbar^2}{2m} \nabla^2 + V_0 \right) \psi(\boldsymbol{x}_t, t) = f_t^Q \psi(\boldsymbol{x}_t, t), \tag{a54}
$$

where $f_t^Q : \mathbb{R}^n \times \mathbb{R} \to \mathbb{R}$ denotes the eigenvalue function of $\bar{H}$ as defined in Theorem C.8, and $V_0 \in \mathbb{R}$ is a constant potential representing an energy barrier of width $D$, such that $\Gamma[0, D) \cong \mathbb{R}[0, D)$. We assume that $V_0 > f_t^Q$.

Here, $x \in \mathbb{R}[0, D)$ indicates a position inside the barrier, while $x \in (\mathbb{R}[0, D))^c$ refers to a position outside the barrier. Restricting the domain to $\Gamma$ yields the simplified form:

$$
\frac{d^2 \psi}{dx^2}(x, t) = \frac{2m}{\hbar^2} \left( V_0 - f_t^Q \right) \psi(x, t), \tag{a55}
$$

where $x \in \Gamma$ denotes the state vector $\boldsymbol{x}_t$ projected onto $\Gamma$.

From the classical mechanics perspective, when the particle mass is sufficiently large, the right-hand side of equation a55 becomes negligible, and the solution $\psi$ tends toward zero. This implies that the particle cannot penetrate the energy barrier to reach the other side. In contrast, from the quantum

mechanics perspective, a particle with sufficiently small mass can tunnel through the barrier. The transmission probability $T \in [0, 1]$ is defined as $T = |\psi_{\{x|x\in\mathbb{R}[0,D)\}}|^2/|\psi_{\{x|x\in(\mathbb{R}[0,D))^2\}}|^2$, and decays exponentially with respect to the barrier width $D$, as given by

$$T \propto \cdot \exp\left(-\frac{2}{\hbar}\int_x^{x+D} \sqrt{2m(V(x) - f_t^Q)}\,dx\right). \tag{a56}$$

Equation equation a56 describes the typical tunneling effect in a scalar domain (Hall, 2013). For a one-dimensional eigenspace and constant potential $V(x) = V_0$, the transmission probability simplifies to

$$T \propto \cdot \exp\left(-\frac{2}{\hbar}\sqrt{2m(V_0 - f_t^Q)}\,D\right). \tag{a57}$$

We interpret the eigenspace $\Gamma$ as being induced by the gradient direction generated from a search algorithm. This foundational concept is extended to a general analysis of tunneling behavior under adiabatic evolution.

**Time-dependent Analysis of Schrödinger Equation for Tunneling Effect**  In this section, we show that the primary process in Algorithm 1, namely $f^Q \leq f_{\text{opt}}^Q$, corresponds to a tunneling effect within the framework of adiabatic evolution.

To analyze this, we consider a state vector $|\psi(t)\rangle \in L^2(\mathbb{R}^n)$ associated with a wave function $\psi : \mathbb{R}^n \times \mathbb{R} \to \mathbb{C}$. The energy is computed as the expectation value $E = \frac{\langle\psi|H|\psi\rangle}{\langle\psi|\psi\rangle}$. If the state vector is normalized, i.e., $\langle\psi|\psi\rangle = 1$, which simplifies to $E = \langle\psi|H|\psi\rangle$ when the state is normalized, i.e., $\langle\psi|\psi\rangle = 1$. Throughout this section, we assume that each distinct state vector $\psi_i$, indexed by $i \in \mathbb{Z}^+$, is orthonormal. Given the time-dependent Schrödinger equation

$$i\hbar\frac{\partial|\psi(t)\rangle}{\partial t} = H(t)|\psi(t)\rangle, \tag{a58}$$

we model the Hamiltonian as a two-level quantum system representing transitions between local minima:

$$H_{2\text{-level}}(s) = \begin{pmatrix} E_1(s) & \Delta(s)/2 \\ \Delta(s)/2 & E_2(s) \end{pmatrix}, \tag{a59}$$

where $s$ is a time-like parameter, $E_k$ denotes the energy (i.e., the eigenvalue of $\bar{H}$ corresponding to the state $|\psi_k\rangle$ at $x_k$), and $\Delta(s)$ is the tunneling matrix element. According to Wentzel–Kramers–Brillouin (WKB) theory Hall (2013), $\Delta(s)$ is proportional to the square root of the transmission probability:

$$\Delta(s) = \sqrt{T} \propto \exp\left(-\frac{1}{\hbar}\int_{x_1}^{x_2} \sqrt{2m(V(x) - \tilde{E}(s))}\,dx\right), \tag{a60}$$

where $\tilde{E}(s)$ is the eigenvalue of $H_{2\text{-level}}(s)$.

By computing the eigenvalues of $H_{2\text{-level}}(s)$, we obtain

$$\tilde{E}(s) = \frac{1}{2}\left[(E_1 + E_2) \pm \sqrt{(E_1 - E_2)^2 + \Delta^2(s)}\right]. \tag{a61}$$

In the context of Algorithm 1, we set $E_1 = f_t^Q(x_1)$ and $E_2 = f_t^Q(x_2)$ for distinct points $x_1, x_2 \in \mathbb{R}^d$ with distance $D = \|x_2 - x_1\|$. To illustrate the case where the quantized optimum $f_{\text{opt}}^Q$ equals a candidate value $f_\tau^Q$, we assume $E_1 = E_2 = f_s^Q$, which simplifies the eigenvalues to

$$\tilde{E}(s) = f_s^Q \pm \frac{\Delta(s)}{2}. \tag{a62}$$

Additionally, we assume that the potential $V(x)$ is constant $V_0$ for the domain $\{x|x \in \|x - \frac{1}{2}(x_1 + x_2)\| < \frac{D}{2}\}$. Under this assumption, the integral in equation a60 simplifies to

$$\int_{x_1}^{x_2} \sqrt{2m\left(V(x) - \tilde{E}(s)\right)}\,dx = \sqrt{2m\left(V_0 - \tilde{E}(s)\right)}\,D, \tag{a63}$$

Table 4: Experimental Environment

| PC Name | OS | GPU | CPU |
|---|---|---|---|
| PC-1 | Linux Ubuntu 24.0 | NVIDIA GeForce GTX 1080Ti | Intel i9 7900 |
| PC-2 | Windows 11 | NVIDIA GeForce RTX 3050 | Intel i9 7900 |
| PC-3 | Windows 11 | NVIDIA GeForce GTXTi | Intel i7 6700 |

which yields

$$\Delta(s) \approx \exp\left[ -\frac{1}{\hbar}\sqrt{2m\left(V_0 - \tilde{E}(s)\right)}\,D \right].$$
(a64)

Since all parameters in equation a64 are finite, the transmission probability is strictly positive, whereas in classical mechanics it would vanish. This phenomenon reflects the quantum tunneling effect in adiabatic evolution. Even when $f_{\mathrm{opt}}^Q = f_\tau^Q$, quantum tunneling enables feasible transitions in Algorithm 1. As a result, the tunneling effect induces a non-strictly decreasing sequence of $f_{\mathrm{opt}}^Q$ without requiring smoothness or convexity, thereby supporting the global convergence of Algorithm 1.

Moreover, the energy given by the eigenvalue in equation a62 corresponds to the quantized objective function, indicating that the dynamics of quantization-based optimization can be interpreted within the formalism of quantum mechanics.

# D   DETAILED INFORMATION OF EXPERIMENTS

All experiments were implemented in Python 3.11.4 using PyTorch 1.18.1, with environment management handled by Anaconda 24.50. The experiments were conducted on three distinct workstations, and the detailed hardware specifications are provided in Table 4.

## D.1   EXPERIMENTS FOR TRAVELING SALESMAN PROBLEM

We conducted optimization tests using identical city coordinates across all experimental trials to ensure consistency. Figure 5 compares the initial route generated by the nearest neighbor algorithm with final solutions from three methods: simulated annealing (SA), quantum-inspired annealing (QIA), and the quantization-based optimization algorithm (QTZ).

Figure 7 shows the minimum cost trajectories for each algorithm tested. Because simulated annealing and quantum-inspired annealing employ acceptance probabilities, these algorithms exhibit significant fluctuations in the early stages of optimization. In contrast, the quantization-based algorithm does not incorporate an acceptance probability, resulting in a decrease in the minimum cost with considerably less fluctuation than that observed in simulated annealing and quantum-inspired annealing.

In addition, this reduced fluctuation leads to faster convergence to a feasible or even global solution compared to the other algorithms. The quantization mechanism in the algorithm induces a hill-climbing effect or a tunneling effect similar to that of conventional methods. However, the proposed algorithm effectively regulates this effect, preventing candidate solutions from diverging to infeasible regions during the optimization process. In contrast, other algorithms allow candidate solutions to diverge with a certain acceptance probability, which often results in longer convergence times, even when the global minimum can eventually be found.

Theoretical analysis (as detailed in the manuscript) demonstrates that quantization confines divergence during hill-climbing within each quantization step $Q_p^{-1}(t)$. However, when the measure of the level set corresponding to higher energy states becomes sufficiently small, suboptimal states may still transition between local minima with non-zero probability. This effective regulation of the hill-climbing effect contributes to the robust optimization performance of the proposed algorithm, especially when compared to algorithms that rely on acceptance probabilities.

**Hyperparameters for Each Algorithm**   The quantization-based optimization requires several hyperparameters associated with the quantization parameter $Q_p(t)$. In the TSP experiments, we set

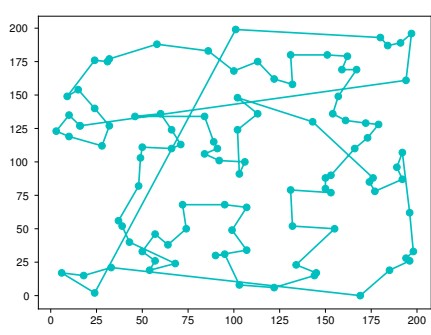

(a) Initial path generated by the nearest neighbor algorithm (cost: 2159).

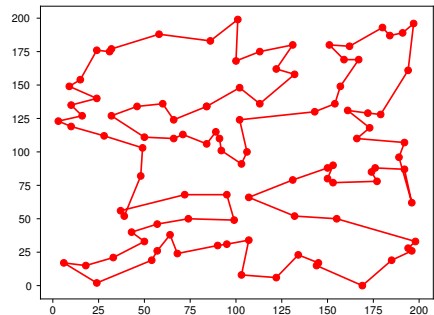

(b) Final path obtained by the simulated annealing algorithm (minimum cost: 1731)

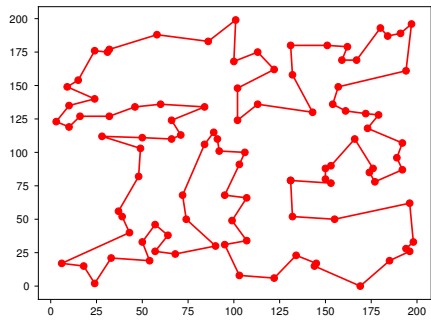

(c) Final path obtained by the quantum-inspired annealing algorithm (minimum cost is 1706)

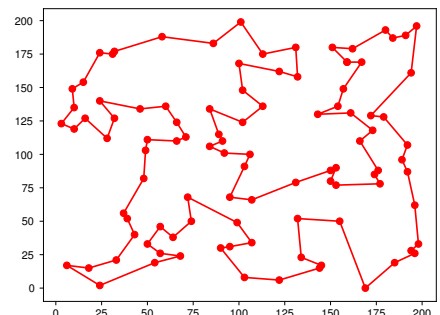

(d) Final path obtained by the quantization-based algorithm (the minimum cost is 1636)

Figure 5: Comparison of 100-city TSP routes provided by each optimization algorithm

---

**Algorithm 2:** Simulated annealing algorithm for TSP

---

**Input:** Objective function $f(x) \in \mathbf{R}^+$
**Output:** $x_{opt}$, $f(x_{opt})$
**Data:** $x \in \mathbf{R}^n$
**Initialization**
Set $\tau \leftarrow 0$ and $T = T_0$
Set initial candidate $x_0$ and $x_{opt} \leftarrow x_0$
Compute the initial objective function $f(x_0)$
$f_{opt} \leftarrow f(x_0)$

**while** *Stopping condition is not satisfied* **do**
    Set $\tau \leftarrow \tau + 1$
    Select $x_\tau$ randomly and compute $f(x_\tau)$
    Set $R = rand() \in \mathbb{R}[0, 1)$
    **if** $f < f_{opt}$ *or* $R < \exp\left(-\frac{|f(x_\tau) - f_{opt}|}{T}\right)$ **then**
        $x_{opt} \leftarrow x_\tau,\ f_{opt} \leftarrow f(x_\tau)$
    **end**
    Set $T \leftarrow T \cdot \alpha$
**end**

---

one of these hyperparameters, $\eta$, relative to an arbitrary initial objective value $f(\boldsymbol{x}_0)$. The hyperparameter values used for $Q_p(t) = \eta\, b^{\bar{h}(t)}$ in the TSP experiments are

$$b = 2, \qquad \eta = b^{-\left\lfloor \log_b\left(\eta_0 \cdot f(\boldsymbol{x}_0)\right)+1 \right\rfloor}, \qquad \bar{h}(t) = \bar{h}(t-1) + 1,\ \bar{h}(0) = 0, \qquad \text{(a65)}$$

where $\eta_0 \in \mathbb{Z}^{++}$ is a positive integer scaling coefficient that controls the initial quantization level. Choosing $\eta_0 > 0$ makes the initial $Q_p(0)$ proportional to the initial objective value.

Rather than the common logarithmic temperature schedule, we employ an exponential schedule. The temperature schedule used for SA and QIA mirrors the quantization step for QTZ:

$$T(t) = T_0 \cdot \alpha^t, \qquad T_0 = 1000,\ \alpha = 0.9995. \qquad \text{(a66)}$$

---

**Algorithm 3:** Quantum Inspired annealing algorithm for TSP

| | |
|---|---|
| **Input:** Objective function $f(x) \in \mathbf{R}^+$ | **while** *Stopping condition is not satisfied* **do** |
| **Output:** $x_{opt}$, $f(x_{opt})$ | $\quad$ Set $\tau \leftarrow \tau + 1$ and Select $x_\tau$ randomly |
| **Data:** $x \in \mathbf{R}^n$ | $\quad$ Compute $H_p(x_\tau) = f(x_\tau), H_B(x_\tau)$ |
| **Initialization** | $\quad$ Compute $\beta \leftarrow 1 - \sqrt{\tau/T_f}$ |
| Set $\tau \leftarrow 0$ and Stopping Time $T_f$ | $\quad$ Compute $H \leftarrow (1 - \beta)H_P + \beta H_B$ |
| Set initial candidate $x_0$ and $x_{opt} \leftarrow x_0$ | $\quad$ Set $R = rand() \in \mathbb{R}[0, 1)$ |
| Compute the mixing Hamiltonian $H_B(x_0)$ | $\quad$ **if** $H < H_{opt}$ *or* $R < \exp\left(-\frac{|H - H_{opt}|}{T}\right)$ **then** |
| Set $H_{opt} \leftarrow H_B$ | $\quad\quad$ $x_{opt} \leftarrow x_\tau, H_{opt} \leftarrow H$ |
| | $\quad$ **end** |
| | $\quad$ Set $T \leftarrow T \cdot \alpha$ |
| | **end** |

---

When we tested SA and QIA with a logarithmic temperature schedule, the search required many more iterations, making fair performance comparison difficult. Using the exponential schedule above, the acceptance probability for SA and QIA is

$$P_{acc}(\boldsymbol{x}_t, t) = \exp\left(-\frac{|f(\boldsymbol{x}_t) - f(\boldsymbol{x}_{\mathrm{opt}})|}{T(t)}\right). \tag{a67}$$

For QIA, we set additional hyperparameters to satisfy the TSP constraints. The adiabatic schedule $\beta(t)$ in equation 22 is defined as

$$\beta(t) = 1 - \sqrt{t/T_f}, \tag{a68}$$

where $T_f = 10{,}000$ is the maximum number of iterations (the same value is used for SA and QTZ). The mixing Hamiltonian $H_B(\boldsymbol{x}_\tau)$ for TSP is implemented as the operator that performs random swaps of two cities in the initial solution produced by the Nearest-Neighbor heuristic.

**Experimental Result for TSP**  For the experiments, we randomly generated city locations, varying the number of cities from 100 to 200, and conducted each experiment using these fixed sets of locations.

The cost trajectories indicate that the quantization-based algorithm effectively constrains the cost at each iteration while exploring suboptimal states. In contrast, SA and QIA exhibit higher costs during the early stages of the search and only minor cost fluctuations in later stages as they approach suboptimal solutions.

Figure 7 illustrates the stochastic behavior inherent in SA and QIA, which arises from the acceptance probability governed by the Gibbs distribution during the early phase of the search. Although QTZ theoretically shares similar stochastic characteristics, Figure 7(c) does not visibly reflect this property in its convergence graph. Even in the enlarged visualization provided in Figure 8, which highlights the stochastic behavior across all algorithms, the trajectory of QTZ exhibits relatively weaker stochasticity. This behavior suggests that QTZ offers more robust optimization capability. The smaller standard deviation observed in the TSP experiments supports this claim.

Furthermore, Table 1 in the main manuscript shows that quantum mechanics-based algorithms outperform thermodynamics-based algorithms in relatively difficult problem instances. For TSP instances with more than 125 cities, both QIA and QTZ demonstrate superior search performance, achieving lower final costs and reduced variance. This trend is similarly observed in other nonlinear optimization problems. In other words, thermodynamics-based algorithms may perform better on relatively simple optimization tasks, whereas quantum tunneling-based algorithms are more effective for complex problems.

These experimental results support the validity of the analysis presented in the main manuscript.

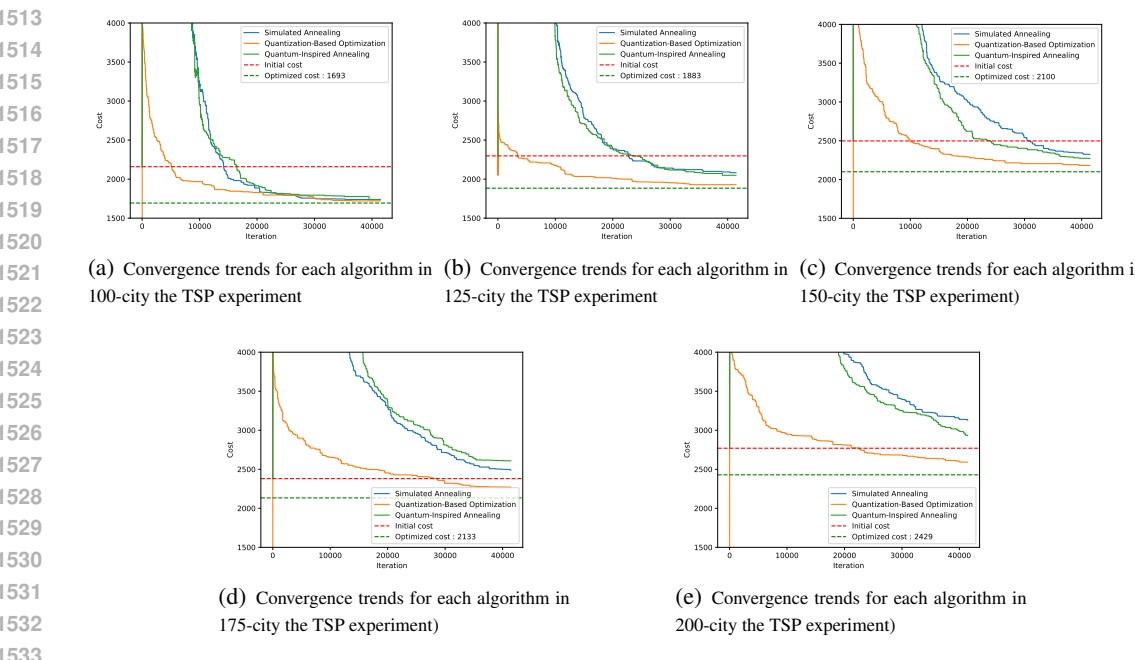

(a) Convergence trends for each algorithm in 100-city the TSP experiment

(b) Convergence trends for each algorithm in 125-city the TSP experiment

(c) Convergence trends for each algorithm in 150-city the TSP experiment)

(d) Convergence trends for each algorithm in 175-city the TSP experiment)

(e) Convergence trends for each algorithm in 200-city the TSP experiment)

Figure 6: Convergence trends for each algorithm in the TSP experiment. The quantization-based optimization scheme exhibits a faster convergence property compared to other algorithms. As the number of cities increases, QTZ represents better convergence performance compared to SA and QIA.

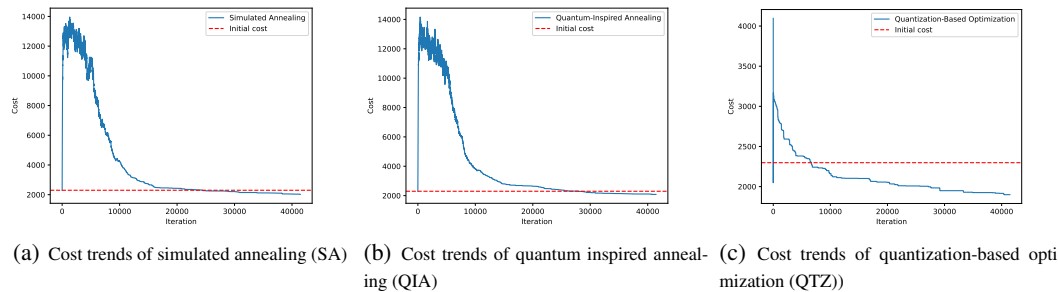

(a) Cost trends of simulated annealing (SA)

(b) Cost trends of quantum inspired annealing (QIA)

(c) Cost trends of quantization-based optimization (QTZ))

Figure 7: Minimum cost trajectories for each algorithm as a function of iteration in the TSP optimization test. The minimum cost for QTZ appears much lower than that of the other algorithms because the overall cost scale for QTZ is significantly lower.

## D.2 EXPERIMENTS FOR NONLINEAR OPTIMIZATION PROBLEM

### D.2.1 TEST FUNCTIONS WITH BINARY SEARCH

To confirm the similarities between QIA and the proposed quantization scheme, we evaluated the optimization performance of the algorithms using the parabolic washboard potential function:

$$f(x) = 0.125x^2 + 2\sin(\alpha x) + 2, \quad \forall x \in \mathbb{R}, \tag{a69}$$

where $\alpha \in \mathbf{R}$ is a tunneling band parameter. A small $\alpha$ corresponds to a wider tunneling band, while a larger $\alpha$ indicates a narrower band. For QIA, we defined $H_0 = 0.125x^2$ as the primitive Hamiltonian at $s = 0$ and $H_1 = f(x)$ at $s = 1$.

Performance was tested using a traditional linear scheduling function $s = t/t_f$, where $t_f$ denotes the final time.

The function in equation a69, studied in Stella et al. (2005); de Falco & Tamascelli (2011), served as a benchmark to validate QIA's optimization superiority. For simulations, $t_f$ was set to 1,000.

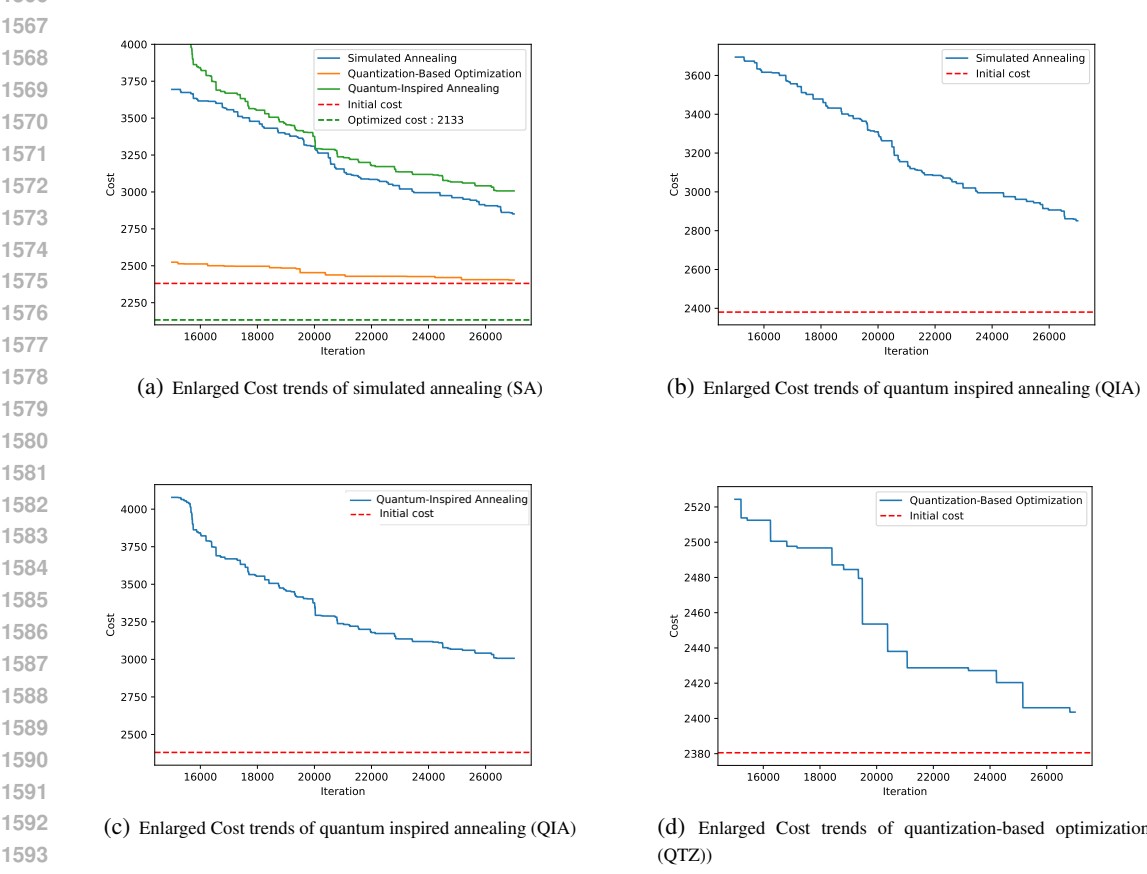

(a) Enlarged Cost trends of simulated annealing (SA)

(b) Enlarged Cost trends of quantum inspired annealing (QIA)

(c) Enlarged Cost trends of quantum inspired annealing (QIA)

(d) Enlarged Cost trends of quantization-based optimization (QTZ))

Figure 8: Enlarged cost trajectories from 15000 to 30000 iterations.

Table 5: Simulation results of the parabolic washboard potential function

| Criterion | Narrow band $\alpha = 10.0$ | | | Wide band $\alpha = 3.0$ | | |
|---|---|---|---|---|---|---|
| | SA | QIA | QTZ | SA | QIA | QTZ |
| Average minimum cost | 0.0031 | 0.0031 | 0.0036 | 0.034 | 0.216 | 0.034 |
| Improvement ratio to the initial setting | 99.93 | 99.93 | 99.92 | 97.75 | 85.73 | 97.75 |

The stopping criterion required the optimization error $f(x) - f(x^*)$ to be less than or equal to the quantization error $Q_p^{-1} = 1/4,096 = 2^{-12}$, which was uniformly applied to all three algorithms.

Contrary to expectations, all algorithms exhibited adequate performance for the narrowband case ($\alpha = 10.0$). However, QIA's improvement ratio decreased significantly for the wideband case ($\alpha = 3.0$), indicating its inability to reliably locate the global minimum in broader tunneling regimes, whereas SA and QTZ succeeded.

Table 5 summarizes the simulation results for the parabolic washboard function with narrow ($\alpha = 10.0$) and wide ($\alpha = 3.0$) bandwidths.

Additional experiments were conducted on continuous benchmark functions from CEC 2017 and CEC 2022, including the Xin-She Yang N4, Salomon, Drop-Wave, and Shaffel N2 functions, which are standard for low-dimensional search algorithm testing.

All tested functions are continuous, enabling SA, QA, and QTZ to converge to global minima within finite iterations, except for the Xin-She Yang N4 function under QA. As shown in Table 6, QTZ outperformed SA and QA, achieving comparable or superior accuracy with fewer iterations.

Table 6: Experimental results for low-dimensional benchmark functions. Left: Number of iterations to convergence. Right: Final result as a percentage of the global optimum (100 = exact optimum, Unit is %).

| Function | QTZ | SA | QIA | Function | QTZ | SA | QIA |
|---|---|---|---|---|---|---|---|
| Xin-She Yang N4 | 3144 | 6420 | 17* | Xin-She Yang N4 | 54.57 | 54.57 | 35.22 |
| Salomon | 1727 | 1312 | 7092 | Salomon | 100.00 | 99.99 | 99.99 |
| Drop-Wave | 254 | 907 | 3311 | Drop-Wave | 100.00 | 99.99 | 99.99 |
| Shaffer N2 | 2073 | 7609 | 9657 | Shaffer N2 | 100.00 | 99.99 | 99.99 |

Table 7: Benchmark functions and their optimization difficulty scores for locating global optima.

| Benchmark Function | Equation | Difficulty |
|---|---|---|
| Ackley | $f(\mathbf{x}) = -a \cdot \exp(-b\sqrt{\frac{1}{d}\sum_{i=1}^{d} x_i^2}) - \exp(\frac{1}{d}\sum_{i=1}^{d} \cos(c \cdot x_i)) + a + \exp(1)$ | 48.25 |
| Whitley | $f(\mathbf{x}) = 1 + \frac{1}{4000}\sum_{i=1}^{n} x_i^2 - \prod_{i=1}^{n} \cos(\frac{x_i}{\sqrt{i}})$ | 4.92 |
| Rosenbrock 2D | $f(\mathbf{x}) = \sum_{i=1}^{d-1}[b(x_{i+1} - x_i^2)^2 + (a - x_i)^2], \quad \mathbf{x} \in \mathbf{R}^2$ | 44.17 |
| Rosenbrock 100D | $f(\mathbf{x}) = \sum_{i=1}^{d-1}[b(x_{i+1} - x_i^2)^2 + (a - x_i)^2], \quad \mathbf{x} \in \mathbf{R}^{100}$ | None |
| EggHolder | $f(x,y) = 977 - (y+47)\sin\left(\sqrt{|y + 0.5y + 47|}\right) - x\sin\left(\sqrt{|x - (y+47)|}\right)$ | 18.92 |
| Xin-She Yang N.4 | $f(\mathbf{x}) = 2.0 + \left(\sum_{i=1}^{d}\sin^2(x_i) - \exp(-\sum_{i=1}^{d} x_i^2)\right)\exp(-\sum_{i=1}^{d}\sin^2\sqrt{|x_i|})$ | 26.33 |
| Rosenbrock Modification | $f(\mathbf{x}) = 74 + 100(x_2 - x_1^2)^2 + (1 - x_1)^2 - 400e^{-\frac{(x_1+1)^2 + (x_2+1)^2}{0.1}}$ | 8.42 |
| Salomon | $f(\mathbf{x}) = 1 - \cos(2\pi\sqrt{\sum_{i=1}^{d} x_i^2}) + 0.1\sqrt{\sum_{i=1}^{d} x_i^2}$ | 10.33 |
| Drop-Wave | $f(x,y) = 1 - \frac{1 + \cos(12\sqrt{x^2 + y^2})}{(0.5(x^2 + y^2) + 2)}$ | 21.25 |
| Powell D4 | $f(\mathbf{x}) = \sum_{i=1}^{d} |x_i|^{i+1}$ | 32.58 |
| Schaffel N. 2 | $f(x,y) = 0.5 + \frac{\sin^2(x^2 - y^2) - 0.5}{(1 + 0.001(x^2 + y^2))^2}$ | 39.58 |

### D.2.2 Standard Continuous Test Functions

We evaluated the performance of the proposed algorithm using ten benchmark functions listed in Table 7. These include the Rosenbrock, Ackley, Whitley, and Powell functions (traditional optimization benchmarks), along with six recently developed functions rated as difficult for locating global optima. To assess performance on high-dimensional problems, we tested the 100-dimensional Rosenbrock function and 4-dimensional Xin-She Yang and Powell functions. With the exception of the Whitley and EggHolder functions, the remaining eight functions (including alternatives like the 2D Rosenbrock) are standardized in the CEC 2017 and CEC 2022 benchmark suites.

Table 7 provides difficulty scores for identifying the global optimum of each function. The quantization process begins at a 5-bit resolution and terminates at a maximum of 17-bit resolution. We implemented a modified QTZ algorithm as the difference equation equation 25 from the manuscript:

$$\boldsymbol{X}\tau t + 1^q = \boldsymbol{X}_{\tau_t}^q + \overline{Q}_p^{-1}(\tau_t)\left\lfloor \overline{Q}p(\tau_t)\left(\eta h(\boldsymbol{X}\tau_t^q, \tau_t) + \frac{1}{2}\right)\right\rfloor, \quad \overline{Q}_p^{-1} \in \mathbb{R}^+, ; \overline{Q}_p^{-1} \triangleq Q_p^{-\frac{1}{2}}. \tag{a70}$$

We compared the proposed algorithm's search speed and accuracy against three conventional gradient-based methods using the Armijo-Wolfe line search: standard gradient descent, conjugate gradient, and quasi-Newton (BFGS). The proposed quantization scheme was applied to each gradient method.

The line search algorithm efficiently identifies minima along the gradient-defined search direction at each iteration. Consequently, the proposed method achieves an optimal line search compared to conventional approaches, accelerating convergence by approximately 30% without compromising accuracy despite quantization errors. As shown in Table 8, the quantization-based algorithm slightly outperforms conventional methods in global optimization tasks while maintaining significantly faster search speeds.

These results suggest that further refinement of quantization-based difference search algorithms could yield even greater improvements over the current proposal.

| Name | Benchmark Function plot | Gradient Descent | Conjugate Gradient | Quasi Newton (BFGS) |
|------|------------------------|------------------|--------------------|--------------------|

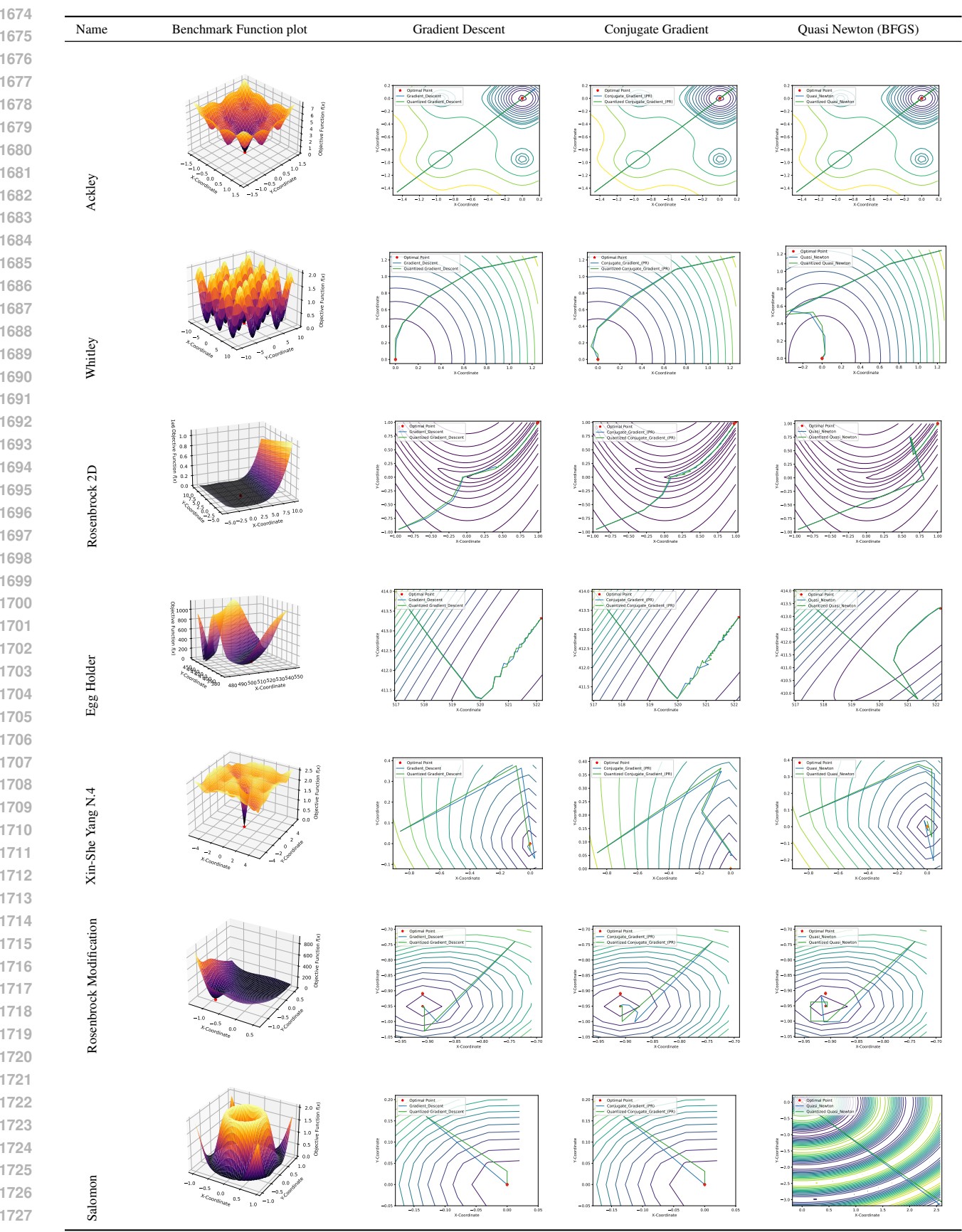

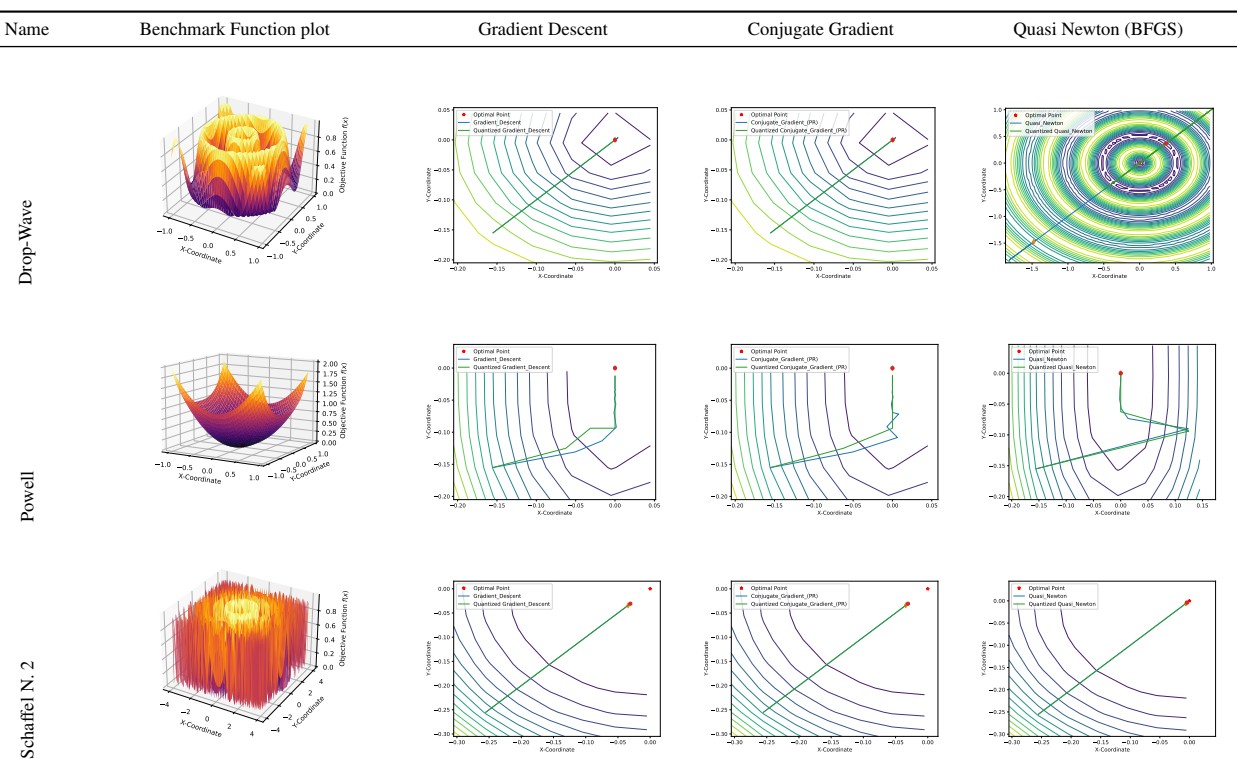

Figure 9: Results of successful search traces for each algorithm on benchmark functions. For a fair comparison, we selected search results in which both the conventional and quantization-based algorithms successfully found an optimal point. The blue line indicates the search trajectories of conventional algorithms, while the green line shows those of the quantization-based algorithm. In some cases, the conventional search trajectory appears as a straight line, which may look like the result of a single iteration; however, these actually required multiple iterations along the straight path. In contrast, the quantization-based algorithm, which produces bent trajectories, required fewer iterations compared to the conventional algorithms.

Table 8: Experimental results of benchmark functions with 3 gradient-based search algorithms using line search

| Benchmark Function | Algorithm | Conventional Algorithm | | QTZ Algorithm | | Imp. ratio of Steps | Imp. ratio of Succ. |
|---|---|---|---|---|---|---|---|
| | | Final step | Success | Final step | Success | | |
| Ackley | Gradient Descent | 8 | 43.0 | 3 | 49.0 | 62.50 | 13.95 |
| | Conjugate Gradient | 16 | 35.0 | 4 | 32.0 | 75.00 | −8.57 |
| | Quasi Newton(BFGS) | 23 | 34.0 | 6 | 23.0 | 73.91 | −32.35 |
| Whitley | Gradient Descent | 13 | 54.0 | 12 | 54.0 | 7.69 | 0.00 |
| | Conjugate Gradient | 9 | 53.0 | 7 | 53.0 | 22.22 | 0.00 |
| | Quasi Newton(BFGS) | 6 | 26.0 | 6 | 30.0 | 0.0 | 15.38 |
| Rosenbrock 2D | Gradient Descent | 3182 | 100.0 | 2427 | 95.0 | 23.73 | −5.00 |
| | Conjugate Gradient | 1601 | 83.0 | 1220 | 81.0 | 23.80 | −2.41 |
| | Quasi Newton(BFGS) | 47 | 87.0 | 49 | 89.0 | -4.26 | 2.30 |
| Rosenbrock 100D | Gradient Descent | 6845 | 82.0 | 2685 | 80.0 | 60.77 | −2.44 |
| | Conjugate Gradient | 4144 | 76.0 | 1262 | 76.0 | 69.55 | 0.0 |
| | Quasi Newton(BFGS) | 839 | 76.0 | 77 | 80.0 | 90.82 | 5.26 |
| EggHolder | Gradient Descent | 85 | 48.0 | 78 | 48.0 | 8.24 | 0.0 |
| | Conjugate Gradient | 113 | 32.0 | 111 | 33.0 | 1.77 | 3.13 |
| | Quasi Newton(BFGS) | 9 | 34.0 | 9 | 37.0 | 0.0 | 8.82 |
| Xin-She Yang N.4 | Gradient Descent | 17 | 3.0 | 10 | 32.0 | 41.18 | 966.67 |
| | Conjugate Gradient | 0 | 0.0 | 8 | 41.0 | inf | inf |
| | Quasi Newton(BFGS) | 17 | 7.0 | 4 | 26.0 | 76.47 | 271.43 |
| Rosenbrock Modification | Gradient Descent | 7 | 11.0 | 8 | 12.0 | −14.29 | 9.09 |
| | Conjugate Gradient | 40 | 23.0 | 46 | 30.0 | −15.00 | 30.43 |
| | Quasi Newton(BFGS) | 7 | 8.0 | 10 | 8.0 | −42.86 | 0.00 |
| Salomon | Gradient Descent | 6 | 18.0 | 1 | 17.0 | 83.33 | −5.56 |
| | Conjugate Gradient | 5 | 18.0 | 1 | 18.0 | 80.00 | 0.00 |
| | Quasi Newton(BFGS) | 6 | 5.0 | 2 | 4.0 | 60.00 | −20.00 |
| Drop-Wave | Gradient Descent | 5 | 5.0 | 2 | 4.0 | 60.00 | −20.00 |
| | Conjugate Gradient | 5 | 4.0 | 1 | 4.0 | 80.00 | 0.00 |
| | Quasi Newton(BFGS) | 4 | 9.0 | 1 | 7.0 | 75.00 | −22.22 |
| Powell D4 | Gradient Descent | 70 | 100.0 | 69 | 100.0 | 1.43 | 0.00 |
| | Conjugate Gradient | 68 | 100.0 | 65 | 100.0 | 4.41 | 0.00 |
| | Quasi Newton(BFGS) | 15 | 100.0 | 14 | 100.0 | 6.67 | 0.00 |
| Schaffel N. 2 | Gradient Descent | 9 | 66.0 | 10 | 63.0 | -11.11 | −4.55 |
| | Conjugate Gradient | 10 | 58.0 | 11 | 58.0 | -10.00 | 0.00 |
| | Quasi Newton(BFGS) | 6 | 64.0 | 7 | 58.0 | -16.67 | −9.37 |
| Average | | | 44.30 | | 46.73 | 32.48 | 0.18 |

Table 9: Input domain (represented with min and max per component) and optimal point corresponding to benchmark functions

| Benchmark Function | Input Domain | Optimal Point |
|---|---|---|
| Ackley | $[-32, 32]$ | $[0, 0]$ |
| Whitley | $[-512, 512]$ | $[0, 0]$ |
| Rosenbrock 2D | $[-5, 10]$ | $[1, 1]$ |
| Rosenbrock 100D | $[-5, 10]$ | $[1, 1, \cdots, 1] \in \mathbf{R}^{100}$ |
| EggHolder | $[400, 600], [300, 500]$ | $[522.16, 413.31]$ |
| Xin-She Yang N.4 | $[-5, 5]$ | $[0, \cdots 0] \in \mathbf{R}^4$ |
| Rosenbrock Modification | $[-1.3, 0.6]$ | $[-0.91, -0.95]$ |
| Salomon | $[-1, 1]$ | $[0, 0]$ |
| Drop-Wave | $[-1.0, 1.0]$ | $[0, 0]$ |
| Powell D4 | $[-1, 1]$ | $[0, \cdots, 0] \in \mathbf{R}^4$ |
| Schaffel N. 2 | $[-4, 4]$ | $[0, 1.25]$ |

## D.3 EXPERIMENTS FOR IMAGE DATASETS

We conducted experiments on the FashionMNIST dataset using a vanilla CNN with three layers. For the CIFAR-10 and CIFAR-100 datasets, we employed the ResNet-50 model, which consists of 50 layers. Representative experimental results are presented in Table 12 and Table 13. A fixed learning rate of 0.01 was used in all experiments, and each model was trained for 200 epochs on every dataset. The batch sizes for FashionMNIST, CIFAR-10, CIFAR-100, and STL-10 were set to 100, 128, 100, and 100 samples, respectively.

We present the learning equations and auxiliary equations for all optimizers used in the experiments in Table 10. The hyperparameters for each learning algorithm are summarized in Table 11. We searched for an appropriate fixed learning rate for each dataset in the range from 0.01 to 0.25 and selected 0.01.

Unlike conventional approaches that employ stepwise learning rate schedules—which are commonly used as default methods in many machine learning frameworks and typically cause error metrics to decrease in a stepwise manner over epochs—we fixed the learning rate at 0.01 for all learning algorithms to isolate and verify the effect of quantization. Although quantization could theoretically diminish the benefits of adaptive learning rates, our analysis suggests that quantization-based optimization can outperform traditional methods in general machine learning applications due to global optimization properties such as thermodynamic equilibrium effects and quantum tunneling dynamics.

**QSGLD and QSLD-ADAM** The fundamental learning equation of the quantization-based optimization is given by

$$\boldsymbol{X}_{\tau+1}^{Q} = \boldsymbol{X}_{\tau}^{Q} + Q_p^{-1}(\tau)\left[Q_p(\tau) \cdot \lambda h(\boldsymbol{X}_{\tau}^{Q})\right]^{Q}, \tag{a71}$$

where $h(\boldsymbol{X}_\tau) = -\nabla f(\boldsymbol{X}_\tau)$ for QSGLD, and $h(\boldsymbol{X}_\tau) = -\frac{\hat{\boldsymbol{m}}_\tau}{\sqrt{\hat{\boldsymbol{v}}_\tau}+\epsilon}$ for Adam-based QSLD.

The quantization parameter $Q_p$ is defined as

$$Q_p = \eta b^{\bar{p}(t_e)}, \tag{a72}$$

where $t_e = \tau/N_B$ denotes the epoch-scaled time index, with $N_B$ representing the number of mini-batches per epoch. The power function $\bar{p}(t_e)$ is bounded by the following inequalities:

$$Q_p = \eta b^{\bar{p}(t_e)}\big|_{t_e=\tau/N_B} \le \frac{1}{C}\log(\tau+2)$$

$$b^{\bar{p}(t_e)}\big|_{t_e=\tau/N_B} \le \frac{1}{\eta\,C}\log(\tau+2) \tag{a73}$$

$$\bar{p}(t_e)\big|_{t_e=\tau/N_B} \le \log_b\left(\frac{1}{\eta\,C}\log(\tau+2)\right).$$

For convenience, we define the constant $C$ in equation a73 as the reciprocal of $\eta$. To ensure the quantization parameter remains bounded and rational, we apply the floor function to its upper bound. The power function for the quantization parameter is thus given by:

$$\bar{p}(t_e)\big|_{t_e=\tau/B} \triangleq \lfloor \log_b \log(\tau+2) \rfloor. \tag{a74}$$

To prevent premature convergence, we introduce an enforcement function:

$$r(\tau, \boldsymbol{X}_\tau) = \left\lfloor \lambda \cdot \frac{\exp(-\varkappa(\tau-\tau_0))}{1+\exp(-\varkappa(\tau-\tau_0))} \cdot \frac{h(\boldsymbol{X}_\tau)}{\|h(\boldsymbol{X}_\tau)\|} \right\rfloor, \quad \tau_0 \in \mathbb{Z}^d, \tag{a75}$$

where $\tau_0$ (measured in mini-batches) determines the activation interval of the enforcement function, and $\varkappa$ controls its decay rate (larger values induce faster decay).

The following summarizes all equations for QSLD and QSLGD:

$$\begin{aligned}
t_e &= \tau/B \\
\bar{p}(t_e) &= \lfloor \log_b \log(\tau+2) \rfloor \\
Q_p &= \eta b^{\bar{p}(t_e)} \\
r(\tau) &= \left\lfloor \lambda \cdot \frac{\exp(-\varkappa(\tau-\tau_0))}{1+\exp(-\varkappa(\tau-\tau_0))} \cdot \frac{h(\boldsymbol{X}_\tau^Q)}{\|h(\boldsymbol{X}_\tau^Q)\|} \right\rfloor \\
\boldsymbol{X}_{\tau+1}^Q &= \boldsymbol{X}_\tau^Q + Q_p^{-1}(\tau)\left[Q_p(\tau) \cdot \left(\lambda h(\boldsymbol{X}_\tau^Q) + r(\tau, \boldsymbol{X}_\tau^Q)\right)\right]^Q.
\end{aligned} \tag{a76}$$

### D.3.1 EXPERIMENTAL RESULTS FOR EACH DATASET

**FashionMNIST** The FashionMNIST dataset is a drop-in replacement for the original MNIST dataset, which consists of handwritten digits. FashionMNIST contains 60,000 grayscale images, each representing one of ten different fashion categories, with each image having a resolution of 28×28 pixels. The ten categories include T-shirts/tops, trousers, pullovers, dresses, coats, sandals,

Figure 10: Error trends of test algorithms to the dataset and neural models

(a) Training error trends of the CNN model on FashionMNIST dataset (Enlarged Training error trends)

(b) Training error trends of ResNet-50 on CIFAR-10 dataset (Enlarged Training error trends)

(c) Training error trends of ResNet-50 on CIFAR-100 dataset (Enlarged Training error trends)

shirts, sneakers, bags, and ankle boots. Each image is labeled according to the category of the depicted fashion item.

Simple vanilla multilayer networks, when equipped with well-tuned optimizers and moderately wide hidden layers, can achieve high accuracy in classifying each category of the MNIST dataset, resulting in minimal classification errors. This makes it difficult to meaningfully evaluate the relative performance of different optimizers on MNIST.

However, although the classification accuracy for FashionMNIST is higher than for CIFAR-10 and CIFAR-100, standard SGD outperforms the ADAM optimizer family on the FashionMNIST dataset. This result suggests that the objective function for FashionMNIST is more convex around the optimum, and previous research (Xie et al. (2021)) has shown that the ADAM optimizer may struggle to select flat minima.

**Experiments for FashionMNIST** The experiments on the FashionMNIST dataset yielded several notable findings. First, similar to the MNIST dataset, the training classification accuracy exceeded 90% for all models tested. However, a clear performance difference was observed between the SGD and ADAM optimizers, with SGD achieving higher classification accuracy than ADAM. As shown in Figure 10, the error curves indicate that ADAM converges more rapidly during the initial epochs, but as training progresses, SGD gradually reduces the error more effectively than ADAM. In general, once the number of epochs exceeds 400, SGD consistently attains 100% training classification accuracy regardless of the learning rate, while the test classification accuracy plateaus at approximately 91.25%.

The proposed QSGLD and ADAM-based QSLD algorithms outperform conventional SGD and ADAM optimizers, but the improvements on the FashionMNIST dataset are not substantial. QSGLD achieves a modest improvement of about 0.2% in both training and test accuracy, while ADAM-based QSLD shows an improvement of around 2%. As illustrated in Figure 10, QSGLD converges slightly faster than SGD due to the similar learning rate, although the overall convergence speeds are comparable. In contrast, ADAM-based QSLD exhibits a convergence pattern similar to that of the ADAM optimizer, but consistently achieves lower error rates than conventional ADAM.

**CIFAR-10, CIFAR-100, and STL-10** We omit detailed explanations for the CIFAR-10, CIFAR-100, and STL-10 datasets, as they are sufficiently well-established in the literature. For the CIFAR and STL datasets, experiments were conducted across ResNet architectures of varying depths to validate the robustness of the proposed algorithm to different network complexities. The results demonstrate that the proposed method consistently outperforms conventional optimizers across all ResNet configurations.

As shown in Table 12, when classifying the CIFAR-10 dataset using ResNet-50, QSGD achieves an 8% improvement in test classification accuracy over SGD. Figure 10 further illustrates that QSGLD

Figure 11: Error trends in the performance of Adam-based QSLD with varying application periods of the enforcement function

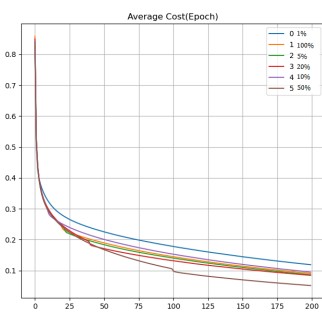 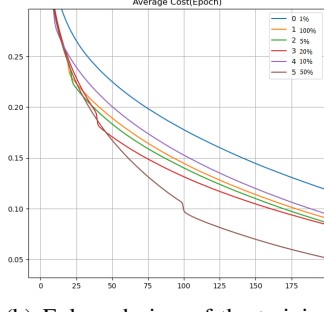 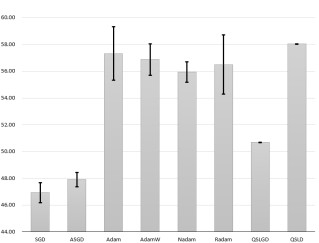

(a) Training error trends of a 3-Layered CNN on the FashionM-NIST dataset

(b) Enlarged view of the training error trends.

(c) Accuracy performances and standard deviations for each optimizer

significantly enhances convergence speed and error reduction compared to SGD, while Adam-based QSLD achieves a 1.5% higher test accuracy than conventional Adam optimizers.

Similar trends hold for CIFAR-100 and STL-10. QSGD outperforms SGD by 11% in test accuracy, whereas Adam-based QSLD shows a 1.0% improvement over standard Adam-based methods.

Notably, experiments on STL-10 reveal distinct characteristics of quantization-based methods. All conventional optimizers exhibit low standard deviations in test accuracy: SGD-based methods show 2% deviation, Adam-based methods 0.5–1.0%, while quantization-based optimizers demonstrate dramatically lower variability (0.005–0.007%). This stability mirrors results from the TSP experiments, suggesting broad applicability of the quantization framework.

## E    ETHICS STATEMENT

This work follows the ethical principles reflected in recent community checklists and the ICLR 2026 submission requirements. We describe data and methods used, potential impacts, and steps taken to maximize societal benefit, scientific integrity, fairness, and transparency.

We aim to advance methods that can improve accessibility and productivity in education and research. The datasets employed are publicly available or licensed for research use; no personally identifiable information or sensitive biometric data were collected or used. Expected positive impacts include improved language understanding and tools for non-native speakers; possible negative impacts (e.g., misuse to produce misleading polished text) are discussed and mitigated below.

All experiments follow reproducible protocols: we provide dataset identifiers, hyperparameters, training procedures, and evaluation scripts where allowed. Results are reported with clear metrics and confidence intervals. We performed ablation studies and include failure cases to avoid overstating contributions.

We explicitly describe all dataset sources, preprocessing steps, model architectures, and training procedures in the paper and supplementary material. Any use of external tools is disclosed. **Notably, an independent large language model was used solely for English grammar correction and sentence polishing of non-technical portions of this manuscript; the LLM was not used to generate technical content, experimental claims, results, or interpretations. All text edits performed by the LLM were reviewed and approved by the authors prior to submission**.

We evaluated model behavior across demographic and linguistic subgroups relevant to our tasks. Where disparities were identified, we report them and describe mitigation attempts (data balancing, calibration, or post-processing filters). We avoid claims that the model is universally applicable and recommend domain-specific validation before deployment.

Table 10: Learning Equations for Each Optimizer

| Optimizer | Learning Equation | Auxiliary Equation |
|---|---|---|
| QSLD | $\boldsymbol{X}_{\tau+1}^{Q} = \boldsymbol{X}_{\tau}^{Q} + Q_p^{-1}(\tau)\left\lfloor Q_p(\tau)\cdot\left(\lambda h(\boldsymbol{X}_{\tau}^{Q}) + r(\tau,\boldsymbol{X}_{\tau}^{Q})\right) + \frac{1}{2}\right\rfloor$ | $Q_p = \eta b^{\bar{p}(t_e)}$ |
| QSLGD | $\boldsymbol{X}_{\tau+1}^{Q} = \boldsymbol{X}_{\tau}^{Q} - Q_p^{-1}(\tau)\left\lfloor Q_p(\tau)\cdot\left(\lambda\nabla_{\boldsymbol{x}}f(\boldsymbol{X}_{\tau}^{Q}) + r(\tau,\boldsymbol{X}_{\tau}^{Q})\right) + \frac{1}{2}\right\rfloor$ | $r(\tau,\boldsymbol{X}_{\tau}) = \left\lfloor \lambda\cdot\frac{\exp(-\varkappa(\tau-\tau_0))}{1+\exp(-\varkappa(\tau-\tau_0))}\cdot\frac{h(\boldsymbol{X}_{\tau})}{\|h(\boldsymbol{X}_{\tau})\|}\right\rfloor$ |
| SGD | $\boldsymbol{X}_{\tau+1} = \boldsymbol{X}_{\tau} - \lambda\nabla f(\boldsymbol{X}_{\tau})$ | $\lambda = 0.01$ |
| ASGD | $\boldsymbol{X}_{\tau+1} = \frac{1}{t}\sum_{i=0}^{t-1}\nabla f(\boldsymbol{X}_{\tau-i})$ | $\lambda = 0.01$ |
| | | $\boldsymbol{m}_{\tau} = \beta_1\cdot\boldsymbol{m}_{\tau-1} + (1-\beta_1)\cdot\boldsymbol{g}_{\tau}$ |
| | | $\boldsymbol{v}_{\tau} = \beta_2\cdot\boldsymbol{v}_{\tau-1} + (1-\beta_2)\cdot\boldsymbol{g}_{\tau}^2$ |
| ADAM | $\boldsymbol{X}_{\tau} = \boldsymbol{X}_{\tau-1} - \frac{\eta}{\sqrt{\hat{v}_{\tau}}+\epsilon}\cdot\hat{\boldsymbol{m}}_{\tau}$ | $\hat{\boldsymbol{m}}_{\tau} = \frac{\boldsymbol{m}_{\tau}}{1-\beta_1^{\tau}},$ |
| | | $\hat{\boldsymbol{v}}_{\tau} = \frac{\boldsymbol{v}_{\tau}}{1-\beta_2^{\tau}}$ |
| | | $\mu_{\tau} = \beta_1\left(1-\frac{1}{2}0.96^{\tau\psi_t}\right)$ |
| | | $\mu_{\tau+1} = \beta_1\left(1-\frac{1}{2}0.96^{(\tau+1)\psi_t}\right)$ |
| NADAM | $\boldsymbol{X}_{\tau} = \boldsymbol{X}_{\tau-1} - \frac{\eta}{\sqrt{\hat{v}_{\tau}}+\epsilon}\cdot\hat{\boldsymbol{m}}_{\tau}$ | $\boldsymbol{m}_{\tau} = \beta_1\cdot\boldsymbol{m}_{\tau-1} + (1-\beta_1)\cdot\boldsymbol{g}_{\tau},$ |
| | | $\boldsymbol{v}_{\tau} = \beta_2\cdot\boldsymbol{v}_{\tau-1} + (1-\beta_2)\cdot\boldsymbol{g}_{\tau}^2,$ |
| | | $\hat{\boldsymbol{m}}_{\tau} = \frac{\mu_{\tau+1}}{1-\prod_{i=1}^{t+1}}\boldsymbol{m}_{\tau} + \frac{1-\mu_{\tau}}{1-\prod_{i=1}^{t}}\boldsymbol{g}_{\tau},$ |
| | | $\hat{\boldsymbol{v}}_{\tau} = \frac{\boldsymbol{v}_{\tau}}{1-\beta_2^{\tau}},$ |
| | | $\boldsymbol{m}_{\tau} = \beta_1\cdot\boldsymbol{m}_{\tau-1} + (1-\beta_1)\cdot\boldsymbol{g}_{\tau},$ |
| | | $\boldsymbol{v}_{\tau} = \beta_2\cdot\boldsymbol{v}_{\tau-1} + (1-\beta_2)\cdot\boldsymbol{g}_{\tau}^2,$ |
| | | $\hat{\boldsymbol{m}}_{\tau} = \frac{\boldsymbol{m}_{\tau}}{1-\beta_1^{\tau}},$ |
| RADAM | $\boldsymbol{X}_{\tau} = \boldsymbol{X}_{\tau-1} - \eta\,\boldsymbol{m}_{\tau}\cdot\begin{cases}\boldsymbol{r}_{\tau}\frac{\sqrt{1-\beta_2^{\tau}}}{\sqrt{\boldsymbol{v}_{\tau}}+\epsilon} & \rho_t > 5\\ 1 & \text{else}\end{cases}$ | $\hat{\boldsymbol{v}}_{\tau} = \frac{\boldsymbol{v}_{\tau}}{1-\beta_2^{\tau}},$ |
| | | $\rho_{\tau} = \rho_{\infty} - \frac{2t\beta_2^t}{1-\beta_2^t}$ |
| | | $\boldsymbol{r}_{\tau} = \sqrt{\frac{(\rho_{\tau}-4)(\rho_{\tau}-2)\rho_{\infty}}{(\rho_{\infty}-4)(\rho_{\infty}-2)\rho_{\tau}}}$ |
| | | $\boldsymbol{m}_{\tau} = \beta_1\cdot\boldsymbol{m}_{\tau-1} + (1-\beta_1)\cdot\boldsymbol{g}_{\tau},$ |
| | | $\boldsymbol{v}_{\tau} = \beta_2\cdot\boldsymbol{v}_{\tau-1} + (1-\beta_2)\cdot\boldsymbol{g}_{\tau}^2,$ |
| ADAMW | $\boldsymbol{X}_{\tau} = \boldsymbol{X}_{\tau-1} - \frac{\eta}{\sqrt{\hat{v}_{\tau}}+\epsilon}\cdot(\hat{\boldsymbol{m}}_{\tau} + \lambda\boldsymbol{X}_{\tau-1})$ | $\hat{\boldsymbol{m}}_{\tau} = \frac{\boldsymbol{m}_{\tau}}{1-\beta_1^{\tau}},$ |
| | | $\hat{\boldsymbol{v}}_{\tau} = \frac{\boldsymbol{v}_{\tau}}{1-\beta_2^{\tau}},$ |

Table 11: Hyperparameters for Each Optimizer

| Learning Algorithm | Hyperparameters |
|---|---|
| QSLD | $\eta \in 2^{19} \approx 0.5\times10^6,\ \bar{p}(t_e)|_{t_e=\tau/B} \triangleq \lfloor\log_b\log(\tau+2)\rfloor$ |
| QSLGD | Equivalent to QSLD |
| SGD | $\lambda = 0.01$ |
| ASGD | Not Defined |
| ADAM | $\beta_1 = 0.9,\ \beta_2 = 0.999,\ \epsilon = 10^{-8}.$ |
| NADAM | Equivalent to ADAM |
| RADAM | $\rho_{\infty} = \frac{2}{1-\beta_2} - 1$ and Others are equivalent to ADAM |
| ADAMW | Equivalent to ADAM |

| Data Set Model | FashionMNIST CNN 3 Layers | CIFAR10 | CIFAR100 | STL10 |
|---|---|---|---|---|
| | | ResNet-50 (56 Layer Blocks) | | |
| QSLGD | 89.29 | 73.8 | 37.77 | 50.68 |
| SGD | 91.47 | 63.31 | 25.90 | 46.92 |
| ASGD | 91.42 | 63.46 | 26.43 | 47.90 |
| QSLD | 91.59 | 85.09 | 49.60 | 58.04 |
| ADAM | 87.12 | 82.08 | 46.32 | 57.32 |
| ADAMW | 86.81 | 82.20 | 47.01 | 56.87 |
| NADAM | 87.55 | 82.46 | 48.56 | 55.93 |
| RADAM | 87.75 | 82.26 | 48.61 | 56.5 |

Table 12: Experimental Results for Image Datasets: Test Accuracy for QSLGD (SGD-based) and QSLD (Adam-based).

| Data Set Model | FashionMNIST CNN 3 Layers | CIFAR10 | CIFAR100 | STL10 |
|---|---|---|---|---|
| | | ResNet-50 (56 Layer Blocks) | | |
| QSLGD | 0.085426 | 0.009253 | 0.030104 | 0.007205 |
| SGD | 0.132747 | 0.001042 | 0.005478 | 2.214468 |
| ASGD | 0.130992 | 0.001166 | 0.004981 | 2.001648 |
| QSLD | 0.059952 | 0.011456 | 0.037855 | 0.005939 |
| ADAM | 0.176379 | 0.012421 | 0.038741 | 0.53936 |
| ADAMW | 0.182867 | 0.012551 | 0.038022 | 0.74659 |
| NADAM | 0.140066 | 0.014377 | 0.037409 | 1.17814 |
| RADAM | 0.146404 | 0.010526 | 0.044913 | 0.763353 |

Table 13: Experimental Results for Image Datasets: Training Errors for QSLGD (SGD-based) and QSLD (Adam-based).

Respect the Work Required to Produce New Ideas and Artefacts We acknowledge prior work and data contributors; all sources are cited. When using third-party code, we follow the original licenses and indicate any modifications. We do not present others' proprietary artifacts as our own and provide appropriate credit for datasets, toolchains, and inspirations.

If accepted, we will include a clear limitations section and a statement of intended use cases. We invite community feedback and commit to updating documentation and mitigation strategies in response to valid concerns.

