# OpenReview forum: "Quantum mechanical framework for quantization-based optimization: from Gradient flow to Schrödinger equation"
_ICLR.cc/2026/Conference — ICLR 2026 Conference Withdrawn Submission_

### Official Review · Reviewer_K2p8 · 2025-10-31

**Soundness:** 3
**Presentation:** 2
**Contribution:** 3
**Rating:** 4
**Confidence:** 3

**Summary:**

This paper proposes a theoretical framework for analyzing optimization algorithms based on objective function "quantization." The authors present an ambitious mathematical narrative that aims to model this quantization-based search process as a gradient-flow dissipative system. This model is then linked to the Hamilton-Jacobi-Bellman (HJB) equation and, through a series of transformations, is claimed to yield the Schrödinger equation. The central thesis is that this connection reveals quantum tunneling as the fundamental mechanism for escaping local minima. The framework also draws parallels to the Fokker-Planck equation, suggesting a unified view of thermodynamic and quantum dynamics in optimization. The authors derive a stochastic gradient update rule from this theory and present experiments on both combinatorial (TSP) and continuous (machine learning) problems.

**Strengths:**

1. Extensive Theoretical Work: The authors have clearly invested a significant amount of effort in constructing an elaborate mathematical framework (Section 3). The attempt to connect concepts from optimal control (HJB), quantum mechanics (Schrödinger), and statistical physics (Fokker-Planck) is mathematically ambitious and, on the surface, intellectually stimulating.

2. Good Performance on Combinatorial Tasks: The gradient-free QTZ algorithm (Algorithm 1) demonstrates strong and robust performance on the TSP instances presented (Table 1), outperforming both SA and QIA, which is a non-trivial experimental result.

**Weaknesses:**

1. Leap of faith in assumption 4: The main problem of this paper's theoretical claim lies in assumption 4, where the authors posits the existence of a "virtual wave function" $\overline{\psi}$ and then defines a new density $\rho \triangleq \psi g$ (Eq 13) by arbitrarily multiplying this new "quantum" probability with the classical Gibbs distribution. This assumption is introduced ad hoc and is not derived from any preceding principles of optimization or quantization. The entire "quantum mechanical" part of the paper (Theorem 3.3) and the "thermodynamic" part (Theorem 3.2) are direct consequences of this single, unsubstantiated assumption. The paper fails to provide a rigorous justification for why signal quantization (a numerical rounding operation) should induce such a complex-valued wave function.
2. Lack of Novelty in the Final Algorithm: After this extensive theoretical journey, the final SDE (Eq 23) and its discrete update rule (Eq 24) are formally identical to Annealed Langevin Dynamics (SGLD). The paper's framework serves as an extremely complex re-derivation of this well-known algorithm, where the "quantization step size" $Q_p^{-1}(t)$ is simply a re-labeling of the temperature $T(t)$ in a cooling schedule. The connection between Langevin dynamics, Fokker-Planck, and escaping potential barriers (which can be interpreted as tunneling via path integrals) is standard, decades-old knowledge in statistical physics and is not a novel contribution of this work.
3. Unconvincing ML Experimental Support: The ML experiments (Sec 4.2) test the SGLD-like algorithm (Eq 25). The results in Table 3 are not compelling; the proposed QSLD/QSLGD methods are often outperformed by standard Adam or SGD (e.g., on CIFAR-100), which contradicts the paper's claims of superior optimization.

**Questions:**

1. How can the gradient-based theory (Section 3) be used to analyze the gradient-free Algorithm 1? If it cannot, why are these two parts presented together as a unified framework?
2. Can the authors provide a rigorous, first-principles derivation (not just an assertion) for how quantization (Definition 1) induces the "virtual wave function" $\overline{\psi}$?
3. For a given objective function $f(x)$, how would one find or write down the specific $\overline{\psi}$ (and thus $\rho$ via Eq 13) that this assumption claims must exist? Without a constructive proof, this assumption remains arbitrary.
4. Given that the final update rule (Eq 24) is formally identical to SGLD, what is the practical, algorithmic-level contribution of this framework beyond re-labeling temperature $T(t)$ as a quantization step $Q_p^{-1}(t)$?

---

> ### Author Response · Authors · 2025-11-17
> **Response to Reviewer's Comments in the Weakness Section**
>
> We would like to thank the reviewer for their detailed and helpful feedback on our manuscript.
> On this page, we first address the reviewer's concerns in the Weaknesses section and provide our perspective.
> We then respond to the reviewer's specific questions on the following page.
> Furthermore, we have revised and uploaded the manuscript to the ICLR open review page.  The line numbers and the pages are the same as in the previous version.
>
> ### **Response to the Reviewer's First Comments**
> As the Reviewer correctly noted, the assumption of a wave function as a virtual function does not derive directly from existing optimization or quantization principles.
> However, as explained in our paper, this assumption arises naturally from the following argument: the objective function value that vanishes due to numerical quantization by the quantization stepsize $Q_p^{-1}$   is arbitrarily represented as the objective function in that region multiplied by a Gibbs distribution.
> Consequently, instead of a wave function, $\phi(\boldsymbol{x})$ can be assumed to be any arbitrary function, such as a quadratic function.
>
> On pages 6, lines 291–298 of our paper, we demonstrate that when the quantized objective function $f_t^Q$ is mapped to the kernel of a Gibbs distribution, the rightmost term in equation (4) of Assumption 4 emerges naturally from the Gibbs distribution kernel of the true objective function. Therefore, although this assumption does not derive directly from existing optimization or quantization principles, it emerges naturally through the mapping of the quantized objective function to the Gibbs distribution kernel. Thus, it constitutes a valid and well-justified assumption.
>
> An additional reason for assuming a wave function in our paper is that when the quantization-based optimization algorithm is analyzed numerically, as shown in equation (19) on page 7, it reduces to adiabatic evolution. This phenomenon is understandable through the lens of quantum mechanics rather than thermodynamic characteristics, as adiabatic evolution is fundamentally explained by quantum tunneling effects.
>
> Furthermore, considering elementary quantum mechanics, the energy levels of quantum particles are inherently discrete. The concept of a quantized objective function is analogous to this principle. Given this fundamental similarity, is it unreasonable to assume a wave function when interpreting our optimization algorithm through the framework of quantum mechanics?
>
> ### **Response to the Reviewer's Second Comment**
> It appears the Reviewer may have interpreted this section as presenting the final algorithm of the paper, likely because it appears near the end of the manuscript.
>
> However, the primary objective of this paper is to present a quantum mechanical framework for analyzing quantization-based optimization. Specifically, through the framework developed up to page 7, we establish a standard Fokker-Planck equation for quantization-based optimization algorithms employed in combinatorial optimization. This framework further enables us to derive continuous-variable optimization algorithms based on Langevin dynamics.
>
> Therefore, equations (23) and (24), as pointed out by the Reviewer, are not the final conclusions of our paper. Rather, they serve to demonstrate that combinatorial optimization and continuous-variable optimization can be connected through the methodology presented in this work. The primary purpose of this discussion is to validate the utility and generalizability of the framework we propose.
>
> Equation (25) represents the fundamental form of the state equations for continuous-variable and machine learning optimization, which is developed on the basis of this framework. The specific implementation methodologies are detailed in Appendices D2.2 and D2.3.
>
> ### **Response to the Reviewer's Third Comment**
> We believe that the table presentation may have caused some misunderstanding. QSLD is a quantization-based machine learning optimizer built upon the ADAM framework, whereas QSGLD is a quantization-based optimizer built upon the SGD framework. Consequently, to ensure fair and meaningful performance comparisons, QSGLD should be compared with SGD-based optimizers, and QSLD should be compared with ADAM-based optimizers.
>
> On CIFAR-10, CIFAR-100, and STL-10 datasets, QSLD demonstrates superior classification performance compared to ADAM-based optimizers when applied to deep networks such as ResNet-50. Similarly, QSGLD exhibits improved performance relative to SGD-based methods on the same benchmarks.
>
> Additionally, the quantization-based optimizer shows dramatically lower variability compared to conventional optimizers.: SGD-based methods show 2% deviation, Adam-based methods 0.5 \~1.0%, while the quantization-based methods demonstrate 0.005 \~ 0.007%.
>
> Due to the word limit, we will address the experimental results on the Fashion-MNIST dataset in the next page.

---

> ### Author Response · Authors · 2025-11-17
> **Response to Reviewer’s Questions (for 1 and 2)**
>
> ### **Question 1**
> Even for gradient‑free algorithms, if the state update occurs only when the update condition for finding a minimum, i.e. $f_{\tau} < f_{\text{opt}}$, is satisfied, we may model the algorithmic dynamics by treating the directional derivative as a negative gradient and writing $dx_t = -\nabla_x f_t\,dt$ (Assumption 3).
>
> In other words, when a randomly chosen candidate satisfies the update condition, the intended interpretation is not a discrete optimization step on a continuous function but a motion from the current state to the next state that satisfies the update condition; this motion may follow a curved trajectory (by a classical least‑action principle) rather than a straight line.
>
>  In fact, continuous‑time analyses of simulated annealing (which do not explicitly use gradients) commonly model the state $x_t$ by an SDE of the form
> $$
> dx_t = -\nabla_x f_t\,dt + \sqrt{2T(t)}\,dW_t.
> $$
> Here $-\nabla_x f_t$ models the process that seeks the minimum, while the Wiener process $dW_t$ represents the Gibbs‑distribution‑based probabilistic decision of whether to accept an update to $x_t$ [1][2][3].
>
> Because our algorithm does not include a probabilistic acceptance step for state updates, we analyze the gradient‑free Algorithm 1 under Assumption 3.
>
> Additionally, if the algorithm’s random selection of candidate states is replaced by selecting an arbitrary state along a gradient or search directional vector, and the update rule is changed from a direct objective comparison to an Armijo‑Wolfe style acceptance rule, the method becomes a classical nonlinear optimization algorithm on continuous functions.
>
> Therefore, whether the algorithm explicitly defines an eigen‑space for search (e.g., gradient directions) and selects candidates on that space, or selects candidate states at random, as long as the algorithm intrinsically implements a process that seeks the minimum, we can represent the state‑update dynamics by a differential equation driven by the negative gradient.
>
> ### **Question 2**
> Fundamentally, the perspective presented in this paper is that the numerically quantized objective and the true objective can be treated as indistinguishable within a range smaller than the quantization stepsize, i.e. $|f_t^Q - f_t| \leq \frac{1}{2} Q_p^{-1}(t)$ (for convenience we denote $f(x_t)$ by $f_t$, which is consistent with the notation in Section 3.2). Within this range, we ignore the exact objective values and use only the gradient of the objective; we assume that the algorithm behaves as if it had some arbitrarily assigned surrogate function value.
>
> Even if we assume an arbitrary surrogate function value within that range, this assumption is merely an analysis tool for interpreting the algorithmic dynamics and is not used at all in the actual algorithm implementation.
>
> If a fully rigorous derivation is required, one would start from the axiom that a quantized objective corresponds to a discrete energy spectrum of a quantum. In quantum mechanics, the discreteness of energy levels arises because the eigenvalues of the Hamiltonian become discrete when the Schrödinger equation’s wavefunction satisfies boundary conditions; therefore, assuming a wavefunction as an axiom is logically coherent.
>
> However, because the logical structure of this paper interprets the algorithm’s tunneling effect via a derivation based on the Schrödinger equation, adopting an orthodox formulation would invert the order of the arguments. Thus, to give a mathematically rigorous proof would effectively require retracing the historical and conceptual development that led to the birth of quantum mechanics.
>
> ### **Observations on the FashionMNIST Dataset Experiments**
> On simpler image classification tasks such as FashionMNIST, classical SGD exhibits superior classification performance compared to our method when applied to shallow networks (e.g., CNN with 3 layers). When the objective function possesses strong convexity (as may be the case for FashionMNIST), pure gradient-based learning methods such as SGD may be more suitable for classification tasks than momentum-based machine learning optimizers with fixed learning rates. This highlights that the effectiveness of quantization-based optimization  reveals better performance for the objective function with lots of local minima.
>
> ### **Reference**
> [1] Geman, Stuart and Geman, Donald, "Stochastic Relaxation, Gibbs Distributions, and the Bayesian Restoration of Images", IEEE Transactions on Pattern Analysis and Machine Intelligence, vol. PAMI-6, no. 6, pp. 721-741, 1984.
>
> [2] Chiang, T.-S. and Hwang, C.-R. and Sheu, S. J., "Diffusion for Global Optimization in $\mathbf{R}^n $, SIAM Journal on Control and Optimization, vol.25, no.3, pp.737-753, 1987.
>
> [3] Locatelli_1996, "Convergence Properties of Simulated Annealing for Continuous Global Optimization", Journal of Applied Probability, vol33. no.4, pp.1127-1140, 1996

---

> ### Author Response · Authors · 2025-11-17
> **Response to Reviewer’s Questions (for 3 and 4)**
>
> ### **Question 3**
> Please refer to the section below Assumption 4, specifically lines 291–298. A quantization error $\varepsilon^q Q_p^{-1}$ exists between the quantized objective function and the true objective function. In Definition 1, we define the fraction for quantization as $\varepsilon^q : \Omega \mapsto \mathbb{R}$.  Since the algorithm compares quantized objective functions, it is not necessary to describe this quantization error rigorously. Therefore, we can represent this quantization error in equation (14) as a function $\phi : \mathbb{R}^d \mapsto \mathbb{R}[-1, 1]$.
>
> Rewriting the kernel of the Gibbs distribution of the quantized objective function using $\phi$, we derive the expression on line 297 as follows:
>
> $$
> \exp(-\textstyle{\frac{1}{h}} f_t^Q)
> = \exp(-\textstyle{\frac{1}{h}} (f(\boldsymbol{x}_t) + \textstyle{\frac{1}{2}} Q_p^{-1}(t) \phi(\boldsymbol{x}_t))) \\
> = \exp(-\textstyle{\frac{1}{h}}f(\boldsymbol{x}_t)) \exp(-\textstyle{\frac{1}{2}} Q_p^{-1}(t) \phi(\boldsymbol{x}_t))
> = g(\boldsymbol{x}_t) \psi(\boldsymbol{x}_t, t).
> $$
>
> In Assumption 4, we set $\psi(\boldsymbol{x}_t, t) = \vert \bar{\psi}(\boldsymbol{x}_t, t) \vert^2$. By introducing an arbitrary real phase function $S(\mathbf{x},t) \in \mathbb{R}$, we can find $\bar{\psi} \in \mathbb{C}$ as follows:
>
> $$
> \bar{\psi}(\mathbf{x},t)
> = \exp\left( \frac{1}{4} Q_p^{-1}(t) (\phi(\boldsymbol{x}_t) + jS(\boldsymbol{x}_t, t)) \right)
> $$
>
> As demonstrated in our paper, by setting $\phi(\boldsymbol{x}_t)$ to a sinusoidal function and $S(\mathbf{x},t)$ to represent the phase variation of a quantum wave, we can simultaneously model both the magnitude of the quantization error and the quantum mechanical wave phenomena.
>
> ### **Question 4**
> One of the most significant contributions is demonstrated in equation (19) and Appendix C.4: for Adiabatic Evolution-based Quantum-Inspired Annealing (QIA, not Quantum Annealing), there is **no need to consider how to define the base Hamiltonian**.
>
> In quantum-inspired algorithms, the base Hamiltonian is typically modeled as a quadratic function. However, in most practical problems, constructing a feasible quadratic function is challenging. Particularly for combinatorial optimization problems, the base Hamiltonian must be designed based on problem-specific approximate solutions. While benchmark functions used in algorithm performance testing have known function equations and optimal solutions, allowing the assumption of a quadratic function as a feasible base Hamiltonian, the absence of such information makes the algorithm prone to erroneous search trajectories.
>
> In quantization-based optimization algorithms, the quantized objective function with the coarsest quantization resolution—corresponding to the largest quantization stepsize—automatically serves as the base Hamiltonian. As the quantization step size is reduced to increase quantization resolution, the search process naturally becomes equivalent to Quantum-Inspired Annealing based on adiabatic evolution.
>
> As mentioned in the Weaknesses section, the formal equivalence between quantization-based optimization and SGLD established through this framework implies that the algorithm is universally applicable with guaranteed convergence and consistency. Since algorithm convergence must be analyzed based on a measure—in this work, a probability measure—we necessarily transform quantum mechanical properties into thermodynamic dynamics for analysis. Consequently, the framework yields the key conclusion that the temperature $T(t)$ can be reinterpreted in terms of the quantization stepsize $Q_p^{-1}(t)$.
>
> However, the framework enables us to feasibly approximate quantum mechanical information of the algorithm (e.g., tunneling probability, escape dynamics from potential wells) through the quantized objective function. This capability will allow us to develop optimization algorithms with enhanced search performance. For instance, in adaptive temperature scheduling, where we predict the required temperature increase to escape a potential well using a general objective function, conventional transmission probability calculations require knowledge of the ground energy $V_0$ and energy level $E$, along with integration according to geometric properties such as objective function curvature. Quantization of the objective function substantially simplifies this: according to equation (21), the transmission probability becomes $T = \exp(-2\sqrt{k Q_p(t)}) \cdot D$, where $k \in \mathbb{Z}^+$ denotes a positive integer. Through this simplification, we can develop efficient and practical optimization algorithms.

---

### Official Review · Reviewer_6Heq · 2025-11-01

**Soundness:** 2
**Presentation:** 1
**Contribution:** 2
**Rating:** 2
**Confidence:** 3

**Summary:**

This submission presents a framework for analyzing optimization algorithms from a quantum mechanical point of view. This framework starts with an optimization problem for the function $f$, and then proposes a general quantization depending on the step size. The authors showed that the level set search process can be modeled as thermodynamic evolution, which can further be modeled as a Schrödinger equation. The ground state of the Schrödinger dynamics can be found using adiabatic evolution, and because of the quantum mechanical tunneling effect, the quantum dynamics leads to global minima.

**Strengths:**

This submission examines the optimization problem from a physics perspective and provides numerical results to support its observations.

**Weaknesses:**

I found the manuscript very hard to read. The whole text is written in a physicsy language without a comprehensive computer science intuition. I am worrying that this submission might not find a suitable audience at ICLR.

In particular, I was trying to grasp the main algorithmic results of this work, but I don't think Theorems 3.1, 3.2, or 3.3 can serve in this role. They just say how to map one type of dynamics to another, but they didn't say anything about why such maps work. For example, does the ground state of equation (17) solve the optimization problem? If so, in what quality? In many places, the authors make vague claims without quantitatively justifying them. For example, in Lines 334-336, "when the energy gap is sufficiently small, the quantum tunneling effect enables the system's state to transition to a lower energy..." This vague statement lacks precise characterization, such as in what probability, what the transition rate is, and how the rate depends on the energy gap. In addition, the general computer science audience might not know what the energy gap is.

This work might contain an important contribution; however, based on the current presentation, it is very hard to justify its value. For example, there's no convergence analysis, so it's hard to see whether this framework has any advantage compared with classical analysis. I think the ultimate goal, as pointed out by the authors, is to propose quantum algorithms based on this framework. Maybe this work will find its value once faster quantum algorithms are developed based on it.

Besides, there is existing work on using quantum mechanical frameworks to speed up optimization problems, for example, arXiv:2303.01471, arXiv:2503.15878, and arXiv:2505.14670. This submission cited the last one, but didn't capture their contribution.

Minor comment: Line 364, redundant word "equation"

**Questions:**

Please address my concerns in the Weaknesses section.

---

> ### Author Response · Authors · 2025-11-17
> **Response to Reviewer's Comments in the Weakness Section (Question 1 and 2):**
>
> We sincerely appreciate the reviewer's thoughtful comments. Regarding certain aspects of the Weaknesses section, we required additional time to fully understand the reviewer's intent and formulate our response accordingly. We now provide our detailed response to the reviewer's concerns below.
>
> ### **Question-1**
> IIn recent years, research papers on quantum mechanics-based optimization algorithms have continued to appear at major AI conferences such as NeurIPS and ICML. Moreover, with the rapid advancement of large language models (LLMs), there remains sustained interest in faster, more efficient optimization algorithms to address diverse subcomponent optimization challenges within LLM architectures.
>
> Therefore, I believe that optimization algorithms grounded in quantum mechanics are, at least for now, receiving meaningful attention from experts across various fields of artificial intelligence, including computer science.
>
> ### **Question-2**
> In Section 2.2, we first presented the operational characteristics of quantization-based optimization (Algorithm 1) through the concept of shrinking level sets. To explain state transitions within a given quantization level, we derived Theorems 3.1, 3.2, and 3.3 in Sections 3.1 and 3.2, demonstrating that Algorithm 1 operates under quantum mechanical dynamics. Theorem 3.1 establishes the fundamental convergence condition, Theorem 3.2 reveals the thermodynamic properties, and Theorem 3.3 demonstrates quantum mechanical properties (as detailed through Assumption 4 up to line 299).
>
> Particularly, through equations (14) and those below, we demonstrate the validity of Assumption 4 by leveraging the stochastic nature of the quantization fraction to represent it as a wave function. Building on Assumption 4, we derive Theorems 3.2 and 3.3 to establish that state transitions within the same quantization level can be modeled quantum mechanically. Therefore, we do not consider this exposition to be an unsupported claim.
>
> While the quantization error $Q_p^{-1}(t) \varepsilon_t$ trivially exists in the algorithm, it is neither computed nor used explicitly. The fact that we model this quantization error as a wave function with equivalent range and subsequently derive the Schrödinger equation demonstrates rigorous grounding, not unfounded assertion.
>
> In contrast, regarding the example cited by the reviewer in lines 334–336, upon reconsideration, we acknowledge that potential ambiguity may exist. However, that passage was added solely to conceptually explain the dynamics of optimization algorithms using adiabatic evolution; we do not believe it necessary to provide detailed derivations of transition probabilities or transition rates for adiabatic evolution in this paper.
>
> Given that we included references to Quantum-Inspired Annealing (QIA) based on adiabatic evolution in the Related Works and lines 359–360 rather than in Section 3, the reviewer may have interpreted lines 326–335 as our original claims. Our primary original contribution concerns equation (19) proven in the Appendix. If needed, we can replace the passage the reviewer identified with an appropriate reference and clarifying statement.
>
> "When the energy gap becomes small during adiabatic evolution, non-adiabatic transitions can occur, allowing the system to temporarily access excited states and tunnel through energy barriers, facilitating global optimization."
>
> **Reference**
> [1] Bettina Heim  and Troels F. Rønnow  and Sergei V. Isakov  and Matthias Troyer, "Quantum versus classical annealing of Ising spin glasses", Science, vol. 348, no.6231, pp.215-217, 2015, .
> [2] Lorenzo Stella and Giuseppe E. Santoro and Erio Tosatti, "Optimization by quantum annealing: Lessons from simple cases", Physical Review B, vol. 72, no.1, 2005.
> [3 ] Diego de Falco and Dario Tamascelli, "An introduction to quantum annealing", {RAIRO} - Theoretical Informatics and Applications, vol.45, no. 1, pp. 99-116, 2011
>
> ### **Appendix**
> Furthermore, we have revised and uploaded the manuscript to the ICLR open review page. The line numbers and the pages are the same as in the previous version.

---

> ### Author Response · Authors · 2025-11-17
> **Response to Reviewer's Comments in the Weakness Section (Question 3, 4,  and 5):**
>
> ### **Question-3**
> Regarding the claim that this paper lacks convergence analysis, we provided a sketch of the convergence proof in Section 3.3 on page 8, lines 400–409. Because the result is essentially the same as the convergence proof used in the referenced work, we presented it in the form of a citation.
>
> Concerning convergence analysis more broadly, in my current research, it is difficult to directly use quantum-mechanical effects in convergence proofs. The reason is that the Schrödinger equation describes the motion of a wavefunction as a vector, while convergence analysis must be formulated using a measure. In quantum-mechanical dynamics, one must analyze convergence with a probability measure, and in that case, the difference from thermodynamics-based convergence analysis disappears. As shown in Theorems 3.2 and 3.3 of this paper (with detailed proofs provided via Appendix Lemmas C.6 and C.7), the probability measure induced by the standard form of the Schrödinger equation is equivalent to the standard form of the Fokker–Planck equation.
>
> However, existing convergence analyses have been carried out for the regime where $T(t)$ is sufficiently small regardless of any convexity assumptions. Because those proofs do not include convexity assumptions (for example, positive definiteness of the Hessian), they only establish convergence in principle; determining convergence rates would require deeper study beyond the scope of this paper.
>
> For example, if one analyzes the geometric properties of the objective function via quantum electrodynamics and then uses Morse theory to design dynamics that escape local minima, such dynamics are likely to be interpreted as divergent in a convergence proof, so appropriate constraints would be needed to prove convergence. Those constraints, however, could make it much harder to identify meaningful convergence rates, so further in-depth research is required.
>
> ### **Question-4**
> I have read the first and third papers you suggested and will refer to them when pursuing more efficient methods for non‑convex, non‑smooth continuous optimization. The first paper exists on arXiv; although I read it, I did not include it in the references of the current draft. If you prefer, I will add that arXiv preprint to the bibliography in the revised manuscript.
>
> I agree with the reviewer’s point that citing the third paper does not amount to a direct contribution from that work to our paper. The reason is that our approach does not develop an optimizer by explicitly constructing a quantum Hamiltonian; rather, we design an optimization algorithm by applying numerical quantization to the objective function and embedding the resulting structure into the algorithmic process.
>
> Both your suggested papers and our work share the same underlying motivation: leveraging Hamiltonian‑like dynamics and tunneling‑type effects to escape local minima. The key difference is methodological. The papers you cite realize this by physically or explicitly constructing Hamiltonians (and studying their quantum evolution), while our work attains a similar functional outcome by embedding a numerically quantized representation of the objective into the algorithmic update rules. As a consequence, the Hamiltonian in our framework does not appear as an explicit physical operator in the algorithmic description.
>
> Accordingly, while I have cited the related works, their technical constructions could not be incorporated wholesale into our method.
>
> ### **Question-5**
> That was my mistake. When compiling LaTeX with the ICLR style, I made an error when using \eqref\eqref. I have corrected that part and uploaded a new version of the manuscript.

---

### Official Review · Reviewer_VFFg · 2025-11-02

**Soundness:** 2
**Presentation:** 3
**Contribution:** 2
**Rating:** 2
**Confidence:** 5

**Summary:**

The authors study the stochastic optimzation problem using the quantum-based framework.

**Strengths:**

The experimental results look good. If you are willing to further strengthen the experimental section, I would be open to reconsidering my score.

**Weaknesses:**

However, theoretical part seems to have already been established in the literature, with more comprehensive and in-depth analyses available.  Have the authors read the following papers？

* Bin Shi, Weijie Su and Michael I. Jordan; On Learning Rates and Schrödinger Operators, Journal of Machine Learning Research, 24(379):1-53, 2023
* B. Shi; On the Hyperparameters in Stochastic Gradient Descent with Momentum, Journal of Machine Learning Research, 25(236):1-40, 2024.

**Questions:**

Could you please explain Algorithm 1: Blind Random Search (BRS) with quantization-based optimization in detail?

---

> ### Author Response · Authors · 2025-11-17
> **Response to Reviewer's Comments in the Weakness Section**
>
> Thank you for your insightful comment.
>
> We carefully read the first paper, "On Learning Rates and Schrödinger Operators".
> In this paper, the authors transform the general gradient-based learning equation or Langevin dynamics-based learning equation into a Schrödinger equation by representing the learning rate as the transition probability of the approximated SDE solution from the standard Fokker-Planck equation, and by dividing the solution of the FPK equation $\rho_s(t, \cdot)$ by the square root of the Gibbs distribution $\mu_s$ to define the wave function.
> (This transform is almost similar to Assumption 4 in this manuscript.)
>
> From the derived Schrödinger equation, after defining the Schrödinger operator, the eigenvalue of the Schrödinger operator and its relationship to the learning rate are calculated.
> The authors of the first paper then analyze the spectrum of the Witten-Laplacian defined through the Schrödinger operator and use this spectral analysis to characterize the properties of the learning rate and derive relationships for the rate of convergence.
>
> Finally, the authors of the first paper define the objective function as a Morse function and, following Morse theory, analyze the determinant of the objective function’s Hessian using Kramer's rule. This analysis reveals that the learning rate is determined by the product of the Gaussian curvature information of the objective function and the exponential of the Morse saddle barrier.
>
> The first paper thus demonstrates that the learning rate in a learning equation should be determined so as to allow the learning process to overcome the Morse saddle barrier, by reflecting the geometric properties of the objective function through the intermediary of the Schrödinger equation. Furthermore, it shows that an exponential decay of the learning rate determines the rate of convergence.
>
> We have not read the second paper. Based on the reviewer's comments, the second paper appears to be an extension of the first, analyzing the properties of the learning rate and other hyperparameters for a learning equation based not on momentum or gradient but on directional derivative, in a manner similar to the first paper.
>
> Personally, I have read another related paper by the same author:
>
> Jiaqi Leng and Bin Shi, "Quantum Optimization via Gradient-Based Hamiltonian Descent", ICML, 2025.
>
> This paper defines the Hamiltonian from quantum mechanics and applies it to learning, proposing a quantum mechanical approach for continuous optimization. This shares the fundamental ideas underlying our work.
>
> We appreciate the reviewer's suggested reference and we hope to have a detailed discussion of how our work relates to and extends this research.
>
> ### **Response to the question**
> The core operation of Algorithm 1 is that we compare **the quantized value of the objective function $f^Q \in \mathbb{Q}$**  instead of the original objective function $f$ to minimize $f$.
>
> As shown in Algorithm 1, if the current quantized value of the objective function $f^Q$ is less or equal to the temporal optimization of the quantized objective function $f_{opt}^Q$, the temporal solution of the state $x_{opt}$, and $f_{opt}^Q$ are updated by the current state $x_{\tau}$ and $f^Q$.   Additionally, when  $f^Q \leq f_{opt}$, the quantization parameter $Q_p$ is updated as represented in Algorithm 1.
>
> The core mechanism of Algorithm 1 can be understood as follows:
> In elementary quantum mechanics, energy levels of quantum particles are inherently discrete. From an optimization perspective, where the objective function represents the energy magnitude, a relatively large quantization step size $Q_p^{-1}(t)$ implies that quantum-mechanical dynamics govern the optimization algorithm.
> Conversely, as the quantization stepsize $Q_p^{-1}(t)$ decreases, this is equivalent to a reduction in $h$ in the relation $h = i\hbar/2m = Q_p^{-1}(t)$.
> Since the variable component in the definition of $h$ is the particle mass $m$, as multiple quantum particles aggregate into molecular and macroscopic systems, the physics transitions from quantum mechanical characteristics to thermodynamic characteristics.
> This transition naturally explains why large-scale systems exhibit thermodynamic rather than quantum behavior; the effective Planck constant decreases as the composite mass increases, pushing the system into the classical regime.

---

### Official Review · Reviewer_UiYk · 2025-11-05

**Soundness:** 1
**Presentation:** 1
**Contribution:** 1
**Rating:** 0
**Confidence:** 4

**Summary:**

This paper develops a theoretical framework for quantization-based optimization that links its sampling stage to gradient-flow dynamics, expressed through a Hamilton–Jacobi–Bellman (HJB) representation. The analysis leverages tools from Schrödinger operator theory and exploits a correspondence between Fokker–Planck dynamics and Schrödinger equations, aiming to provide a unified methodology across combinatorial and continuous optimization. The authors complement the theory with numerical studies on traveling-salesman problems, low-dimensional nonconvex tasks, and standard machine learning datasets (e.g., FashionMNIST, CIFAR), where the approach exhibits promising results.

**Strengths:**

- This work tries to propose a theoretical framework that unifies the analysis of quantization-based optimization for combinatorial and continuous problems.
- An algorithm named "Blind Random Search" (BRS) with quantization-based optimization is proposed. Despite its simple implementation, the authors propose a mechanism (under a set of assumptions) that this algorithm may work in practice.
- Numerical experiments are provided to illustrate the practical performance of quantization-based optimization. The results appear to be promising, as the proposed method often outperforms existing quantum and classical optimization methods across a wide spectrum of application domains.

**Weaknesses:**

**The exposition lacks cohesion**: although the manuscript is dense with terminology and equations, central notions (quantization-based optimization, Schrödinger operators, adiabatic algorithms, HJB) are named but neither defined nor carried through, and sections do not build on one another **semantically** or **technically**.

1. The central concept is not adequately defined. The manuscript attempted to unify combinatorial and continuous optimization under a “quantization-based” umbrella, but it remains unclear what “quantization-based” precisely means in each setting and whether the same mathematical machinery applies across them. Please provide a formal definition (including assumptions), a minimal working example in each domain, and an explicit mapping that shows which results/tools transfer (and which do not) between the discrete and continuous cases.

2. The “Related Works” section omits several recent threads on nonconvex optimization grounded in quantum mechanics and quantum-inspired methods. For example:
- Leng, Jiaqi, Ethan Hickman, Joseph Li, and Xiaodi Wu. "Quantum Hamiltonian Descent." arXiv preprint arXiv:2303.01471 (2023).
- Chen, Zherui, Yuchen Lu, Hao Wang, Yizhou Liu, and Tongyang Li. "Quantum Langevin dynamics for optimization." Communications in Mathematical Physics 406, no. 3 (2025): 52.
- Goto, Hayato, Kosuke Tatsumura, and Alexander R. Dixon. "Combinatorial optimization by simulating adiabatic bifurcations in nonlinear Hamiltonian systems." Science advances 5, no. 4 (2019): eaav2372.

3. This paper would benefit from clearer notation and stronger narrative flow. Some symbols are non-standard or insufficiently defined, and transitions between paragraphs (even within a section) are difficult to follow. New concepts are repeatedly introduced without tying them back to earlier definitions or results; see some examples in the "Questions" section.

**Questions:**

- What does $f \in \mathbb{R}$ or $f \in \mathbb{Q}$ mean? $f$ represents a function, and it is ambiguous to claim that it belongs to the set of real/rational numbers.
- Is the "quantization of f" (Definition 1) a new function? How should I interpret $f \in \mathbb{Q}$? What if the function takes irrational values?
- There should be a period at the end of Definition 4.
- Please justify the rationale behind Assumption 3: why "blind random search" (Algorithm 1) can be interpreted as a gradient flow? How should we interpret "gradient" in this process?
- Near Line 205: why "the set has a non-zero measure" implies "the spectrum of $f(x_{t+1})$ ....", and how does this "coincide with the eigenvalues of the 2-level Hamiltonian"? What is the spectrum of a function (standard definition of "spectrum" means the set of eigenvalues of a matrix, if finite-dimensional)? And which 2-level Hamiltonian?
- How is the "exponential kernel $\Phi$" related to any algorithms/procedures discussed in Section 2? Why the function $V$ (Eq. (8)) is defined over $[0,1]$, while $f(x)$ is apparently defined over $\mathbb{R}^d$?
- Thermodynamical evolutions (such as those in the Langevin dynamics) are fundamentally different from adiabatic quantum evolution. I do not follow the discussion on page 7.
- In the numerical experiment (Figure 2), how is the "iterations" defined for quantum annealing? Isn't quantum annealing a continuous-time quantum evolution by default? And is it a fair comparison using "iterations"? Quantum clock rate (i.e., the frequency of implementing elementary quantum computation on quantum computers) can be much higher than classical clock rate, leading to very different wall-clock time scales.

---

> ### Author Response · Authors · 2025-11-19
> **Response to Reviewer's Comments in the Weakness Section**
>
> ### **Explicit mapping of each section**
> The sections are mapped as follows:
> - Section 2, 2.1: Definition and Assumptions for our research
> - Section 2, 2.2: Convergence analysis based on the measure of the level set in Algorithm 1
>   - The case $f_{t+1}^Q < f_t^Q$ is trivial.
>   - For the equality case, $f_{t+1}^Q = f_t^Q$, if there exists a search process that can find a feasible solution, then at least the sequence of the measure of the level set forms a non-strictly monotone decreasing sequence, thus guaranteeing convergence.
> - Section 3, 3.1: To explain the search process for the equality case, we derive the Hamilton-Jacobi-Bellman (HJB) equation based on gradient flow.
> - Section 3, 3.2: Since the quantized objective function satisfies $\\| \nabla_x f_t^Q \\|= 0$, we explain the additional transformation for interpreting this and the validity of the transformation. Moreover, through this additional transformation, Algorithm 1 is interpreted quantum mechanically via the Fokker-Planck equation and the Schrödinger equation based on it.
>   - The evolution process of Algorithm 1 is explained as a quantum-inspired annealing based on adiabatic evolution through the Schrödinger equation.
>   - For continuous function optimization, a basic discrete optimization model is derived through the Fokker-Planck equation (Overdamped Langevin Dynamics).
>   - We present an example of a learning equation applicable to machine learning applications based on the contents so far.
>
> ### **Response to the requirement of mapping on definitions, assumptions, and mathematical mechanisms**
> First, it is necessary to briefly explain the main idea of this paper, as this will provide a reasonable response to the Reviewer’s comments.
>
> To briefly explain the idea of this paper, by regarding the state $x_t$ as a quantum entity, quantum mechanics informs us that its energy levels are discrete. Therefore, we treat the quantized  objective function as discrete energy levels while performing the search algorithm. This idea requires a quantum mechanical interpretation. For this purpose, we first model the most basic form of the search algorithm as a negative gradient. As a result, the movement of the state is modeled as the gradient flow.
>
> We formulate the gradient flow as the Hamilton-Jacobi-Bellman (HJB) equation, as presented in the paper. From the perspective of classical mechanics, we establish a transformation that derives the Fokker-Planck equation(FPE) from the HJB equation. Using the Witten-Laplacian, we derive the Schrödinger equation from the FPE. Through this procedure, we can analyze the quantum-mechanical properties of the algorithm.
>
> "**quantization-based**" throughout this paper refers to the quantization of the objective function. This is clearly defined within the paper via Definition 1 and Algorithm 1. Section 2.1 summarizes the definitions and assumptions used in this paper, and Assumption 4 is explicitly stated in Section 3.2, since the Fokker-Planck and Schrödinger equations need to be derived from the HJB equation.
>
> We describe each equation’s derivation method in the Appendix. Regarding the HJB equation in Equation (11), it is an equation presented in many references; therefore, we omitted its derivation.
> The Reviewer mentioned uncertainty about whether the same mathematical mechanism applies. Please provide more specific cases. The examples in the Questions appear to be misunderstandings or misinterpretations by the Reviewer.
> We have described the mathematical mechanisms in the main text in a unified and consistent manner, and we provide detailed proofs in the Appendix through Lemmas and additional propositions.
>
> ### **Response to the related work**
> The first paper was read; however, it was not cited in the manuscript. Instead, we read and cited an extended version of that work in our paper[1].
>
> The methodology of the second paper relies on a convexity assumption to prove convergence; for this reason, we believe its practical efficiency is limited relative to its methodological potential. The third paper proposes a novel optimization method based on adiabatic evolution.
> However, the main objective of the paper is to demonstrate that numerical quantization is an alternative implementation of the quantum‑mechanical tunneling effect for adiabatic evolution.
> Accordingly,  we cited prior literature that explains the essential adiabatic dynamics.
>
> The reviewer noted that some notation in our manuscript is nonstandard or unclear. However, we have clarified all notation used in our paper in a dedicated appendix.
>
> We respectfully suggest that what matters most is the ideas and the theoretical/algorithmic foundations on which the results are built, rather than strict adherence to a notational convention.
>
> ### **Reference**
> [1]  Jiaqi Leng and Bin Shi, "Quantum Optimization via Gradient-Based Hamiltonian Descent", ICML, 2025.

---

> ### Author Response · Authors · 2025-11-19
> **Response to Reviewer’s Questions (Question 1~ Question 5)**
>
> ### **Question 1, 2**
> The Reviewer does not distinguish between $f \in \mathbb{R}$ and $f^Q \in \mathbb{R}^Q$ as defined in Definition 1, whether intentionally or by mistake.
> In our manuscript, Definition 1 defines the (numerical) quantization that underlies the entire paper.
> The “quantization” we use is emphatically not the quantization of quantum mechanics.
> We explicitly state in both the Introduction and the Conclusion that “quantization” in this paper denotes numerical quantization; therefore, the quantization in Definition 1 should be interpreted in the sense of number theory and signal processing.
>
> Even when $f$ takes an irrational value, the quantized value $f^Q$ given by Definition 1 (equation (3)) is rational. For example, let $f=\sqrt{3}$ and choose $Q_p=4$ according to the definition. Then by equation (3),
>
> $$
> f^Q = \frac{1}{4}\left\lfloor 4\big(\sqrt{3}+0.5\big)\right\rfloor
> = \frac{1}{4}\left\lfloor 4\sqrt{3}+2\right\rfloor
> = \frac{1}{4}\left\lfloor 4\times 1.73205\ldots + 2\right\rfloor
> = \frac{1}{4}\lfloor 8.9282\ldots\rfloor
> = \frac{1}{4}\times 8 = 2.
> $$
>
> Thus, in this example $f^Q \in \mathbb{Q}$. In Appendix B (Notation), we explicitly declare that the result of the $\lfloor\cdot\rfloor$ operation is an integer; therefore, the statement $f^Q \in \mathbb{Q}$ in Definition 1 is immediate and without ambiguity.
>
> ### **Question 3**
> This is a typo. We uploaded the modified version to OpenReview.
>
> ### **Question 4**
> The answer to this question is the same as our response to Reviewer K2p8’s first question, so we summarize briefly.
>
> In short, it is not that selecting candidates randomly is interpreted as gradient flow; rather, the update process that accepts a candidate when it satisfies the acceptance condition and thereby decreases the objective can be modeled by a gradient flow.
>
> Let the objective at $x_t$ be $f: \mathbb{R}^d \to \mathbb{R}^+$ with value $f_t$, and suppose an accepted update $x_{t+1}$ satisfying, for example, $f_{t+1}\le f_t$ is obtained after multiple trials so that the intermediate samples are $\\{x_\tau \\}_{\tau=t}^{\tau=t+1}$. There is no need to model the dynamics over $t$ by explicitly including the random sampling index $\tau$. Therefore, by the update condition, we may write.
>
> $$
> 0 \ge f\_{t+1} - f\_t = \int\_{x\_t}^{x\_{t+1}} df\_s = \int\_t^{t+1} \nabla\_x f\_s \cdot dx\_s,
> $$
>
> and, by the least action principle, setting $dx_s = -\nabla_x f_s\,ds$ is consistent. Indeed,
>
> $$
> \int_t^{t+1} \nabla_x f_s \cdot dx_s = -\int_t^{t+1} \|\nabla_x f_s\|^2\,ds < 0,
> $$
> which matches the required decrease.
>
> ### **Question 5**
> The manuscript does not use the phrase “the **spectrum** of $f(x_{t+1})$.” Nowhere in the paper do we assert a spectrum for the function. The Reviewer appears to have confused the paper’s use of **supremum** with a statement about spectrum.
>
> After reading the Reviewer’s comment again, we agree that the sentence “\~coincide with the eigenvalues of the 2‑level Hamiltonian” is too strong. It is more appropriate to revise that sentence to “\~be proportional to the eigenvalues of the 2‑level Hamiltonian.”
>
> A more detailed explanation is as follows (in Appendix C.4, page 25). We introduced a Hamiltonian because the scalar relation in the main text, equation (19), has the same structural form as the adiabatic‑evolution expression in equation (18). Equation (18) is written in terms of a (linear) Hamiltonian operator, while equation (19) is a scalar relation among $f_t, f_t^Q,$ and $f_0$, so it can be viewed as an expression for eigenvalues of a Hamiltonian.
>
> The algorithm models state transitions between two quantized objective values, so the appropriate quantum analogue is a 2‑level Hamiltonian. The 2‑level Hamiltonian describes transitions between two energy levels $E_1$ and $E_2$, and in Algorithm 1, the transition from $f_{opt}^Q = f_{t_0}^Q$ to $f_t^Q$ is the corresponding process.
>
> The nontrivial case is $f_t^Q = f_{t_0}^Q$, for which we must explain the state transition; accordingly we identify $f_t^Q$ and $f_{t_0}^Q$ with $E_1$ and $E_2$ respectively and derive equation a62 via the steps around lines 1340–1342:
>
> $$
> \tilde{E}(t) = f_t^Q \pm \frac{\Delta(t)}{2}\quad \because s=t.
> $$
>
> The Hamiltonian eigenvalue $\tilde{E}(\tau)$ corresponds to the (non‑quantized) objective value, so
>
> $$
> f_t = f_t^Q \pm \frac{\Delta(t)}{2} \Rightarrow f_t^Q = f_t \mp \frac{\Delta(t)}{2}.
> $$
>
> By Definition 1,
>
> $$
> \mp \frac{\Delta(t)}{2} = \varepsilon_t\, Q_p^{-1}(t).
> $$
>
> This relation follows from the definition of the quantized objective and from the equality case condition:
>
> $$
> |f(\boldsymbol{x}\_{t+1}) - f(\boldsymbol{x}\_t)| < \tfrac{1}{2} Q\_p^{-1}(t)
> \Rightarrow
> \sup\_{\boldsymbol{x}\_{t+1}\in\check S(f\_t^Q)} f(\boldsymbol{x}\_{t+1}) = f(\boldsymbol{x}\_t) + \tfrac{1}{2} Q\_p^{-1}(t).
> $$

---

> ### Author Response · Authors · 2025-11-19
> **Response to Reviewer’s Questions (Question 6, 7, and 8)**
>
> ### **Question 6**
> This was a typo on our part in the definition of the function $V$. Part of the sentence about $g:\mathbb{R}^d\times\mathbb{R}^+\mapsto\mathbb{R}[0,1]$ was introduced by a typing/copying error. We have uploaded a corrected version to OpenReview.
>
> The “exponential kernel $\Phi$” is used as a transformation tool for the interpretation of the algorithm.
>
> ### **Question 7**
> I agree with the Reviewer that thermodynamic evolution and adiabatic quantum evolution are fundamentally different. From the perspective of total evolution time and on whether quantum fluctuations are included, the dynamics of the two algorithms are indeed distinct.
>
> However, when the adiabatic quantum evolution is applied to combinatorial optimization problems that can be readily and formally mapped to an Ising model, Thermodynamical Evolution such as Simulated Annealing and Quantum (Inspired) Annealing can be mathematically equivalent in certain senses[1].
>
> To analyze the convergence of adiabatic quantum evolution based on an Ising model, one must study the spectrum (eigenvalues) of the Hamiltonian. In that analysis, the wave‑nature of quantum mechanics does not explicitly appear; instead, the mathematical structure reduces to the spectrum of a probability measure that is the basis of thermodynamic evolution. This equivalence is explained in the reference[1, 2].
>
> Distinguishing adiabatic quantum evolution from thermodynamic evolution, therefore, requires examining local‑time optimization dynamics (i.e., whether the mechanism that enables escape from local minima is quantum tunneling or a thermodynamic fluctuation), rather than only global convergence properties[3].
> Accordingly, in Section 3.2, we explicitly analyze the tunneling effect and adiabatic evolution and show that, in the equality case, Algorithm 1 produces optimization behavior that can be understood as quantum‑mechanical tunneling.
>
> Additionally, references [4, 5, 6] report experimental results showing no consistent performance advantage of QIA over SA; their findings indicate that performance differences depend on the specific optimization problem.
>
> #### **Reference**
> [1] Nishimori, Hidetoshi and Tsuda, Junichi and Knysh, Sergey, "Comparative study of the performance of quantum annealing and simulated annealing", Phys. Rev. E, vol.91, issue 1, pp.012104, 2015.
> [2] Somma, R. D., Batista, C. D. and Ortiz, G., "Quantum Approach to Classical Statistical Mechanics", Phys. Rev. Lett., vol. 99, no. 3, pp.030603, 2007.
> [3]  Nishimori, H., "Comparison of quantum annealing and simulated annealing",  *Eur. Phys. J. Spec. Top. 224, pp. 15–16,  2015.
> [4] Marlon Azinović, Daniel Herr, Bettina Heim, Ethan Brown, Matthias Troyer, "Assessment of Quantum Annealing for the Construction of Satisfiability Filters", SciPost Physics, vol. 2, issue 2, id. 013, 2017.
> [5] Dennis Willsch, Madita Willsch, Fengping Jin, Kristel Michielsen, Hans De Raedt, "GPU-accelerated simulations of quantum annealing and the quantum approximate optimization algorithm", Computer Physics Communications, Volume 278, 2022.
> [6] Bettina Heim, Troels F. Rønnow, Sergei V. Isakov, Matthias Troyer, "Quantum versus classical annealing of Ising spin glasses", vol. 348, no.6231, pp.215-217, 2015.
>
> ### **Question 8**
> I was surprised by this question. In the main text we consistently refer to quantum‑inspired annealing, so I did not understand why the Reviewer mentioned quantum annealing. Upon checking, I found that Figure 2 mistakenly labels the method as “quantum annealing” instead of “quantum‑inspired annealing.” This is a typo on our part. We will correct the figure to read “quantum‑inspired annealing” to address the Reviewer’s concern.
>
> Quantum‑inspired annealing is a discrete‑time simulation of continuous‑time quantum evolution implemented in software on conventional computers. Therefore, all algorithms used in our experiments simulate quantum dynamics in discrete time on conventional hardware, and the numerical comparisons are fair.
>
> As the Reviewer knows, before quantum computing‑based optimization research matured, many methods now called quantum‑inspired annealing were historically referred to as quantum annealing up to around 2020. This naming history is reflected in the references cited in our paper.

---

### Note · Authors · 2026-01-26

**Comment:**

Dear ICLR Program Chairs,

We hereby request the withdrawal of our manuscript entitled "Quantum mechanical framework for quantization-based optimization: from Gradient flow to Schrödinger equation" (Submission No.: 4222), which was submitted to ICLR 2026 and has received a rejection decision.

Accordingly, we would like to officially withdraw this paper from the submission system.

Due to unforeseen circumstances, we were unable to engage in sufficient discussion with the reviewers prior to receiving this decision, which has been a source of disappointment. Nevertheless, we sincerely appreciate the efforts of the Program Committee (PC) and Area Chairs (AC), who dedicated their time under considerable pressure to evaluate our work.

We would like to once again express our gratitude to the reviewers and the program committee for their evaluation. Based on the feedback, we plan to revise the manuscript and consider resubmission to another venue in the future.

Thank you very much.

Sincerely,

**Withdrawal Confirmation:**

I have read and agree with the venue's withdrawal policy on behalf of myself and my co-authors.

---

### Meta-Review · Area_Chair_CF3g · 2025-12-31

**Summary:**

All four reviewers agree that the paper presents a review of standard techniques within the applications of spectral operator theory to the analysis of optimization algorithms, in the tradition of:

- Marco Locatelli: Simulated Annealing Algorithms for Continuous Global Optimization: Convergence Conditions Published. Journal of Optimization Theory, 104: 121–133, 2000

- Bin Shi, Weijie Su and Michael I. Jordan; On Learning Rates and Schrödinger Operators, Journal of Machine Learning Research, 24(379):1-53, 2023

and then, in Section 3, utilizes an unsubstantiated Assumption 4 to obtain the main results.

**Reviewer Concerns:**

The concerns of the reviewers are rather hard to address -- and have most definitely not been addressed.

**Reviewer Scores:**

None of the reviewers indicated any willingness to improve their scores.

---

### Decision · Program_Chairs · 2026-01-26

Reject